# OSCS: Online Selection with Provable FAR Control for LLM Safety

**Zirui Hu** [1]   **Zheng Zhang** [1]   **Yingjie Wang** [1]   **Dacheng Tao** [1]

## Abstract

Large language models (LLMs) are vulnerable to malicious inputs, posing serious risks in high-stakes applications. Although existing detection-based defenses achieve strong empirical performance, they generally lack explicit control over the false acceptance rate (FAR), a critical safety requirement in sensitive deployment scenarios. This challenge is further complicated by two practical constraints: the lack of malicious calibration samples and the streaming nature of real-world inputs. To address these challenges, we propose *OSCS*, a novel framework for online FAR control without requiring malicious calibration data. OSCS leverages detection scores produced by existing defenses and employs recursive density estimation to estimate benign probability from the test stream. Based on these estimates, OSCS performs real-time accept/reject decisions while provably satisfying a user-specified FAR target. Theoretically, we show that OSCS controls the FAR up to a vanishing excess term under mild assumptions. Extensive experiments on backdoor and jailbreak attack tasks further demonstrate the effectiveness of OSCS, showing that it consistently achieves robust FAR control across diverse attack settings while outperforming existing baselines.

## 1. Introduction

Large language models (LLMs) have achieved remarkable success across a wide range of natural language processing (NLP) tasks (OpenAI, 2023; Minaee et al., 2025; Chang et al., 2024; Zhang et al., 2025a). Despite these advances, LLMs remain vulnerable to serious safety threats, including backdoor attacks (Yao et al., 2024; Qi et al., 2021b), prompt injection (Pathade, 2025), and jailbreaking (Zou et al., 2023;

Wei et al., 2023). These attacks carefully craft malicious inputs to manipulate model behavior, potentially inducing harmful or misleading outputs. As LLMs are increasingly deployed in high-stakes domains such as healthcare (Maity & Saikia, 2025), finance (Chen et al., 2024), and education (Wang et al., 2024; Ross et al., 2025), the consequences of such failures become particularly severe, motivating the development of a line of effective defense mechanisms.

Among existing defenses, detection-based methods (Yi et al., 2024; Gao et al., 2019; Qi et al., 2021a; Jain et al., 2023) have attracted substantial attention due to their practicality: they can be deployed at inference time without modifying or retraining the underlying LLM. At inference, these approaches assign each test input $X_t$ a detection score $W_t = W(X_t)$ that quantifies its deviation from benign behavior, for example through representation inconsistency (Chen et al., 2022) or robustness-based signals (Gao et al., 2019). The score is then compared against a predefined threshold to make a binary accept/reject decision, accepting only inputs deemed benign to reach the model.

Despite achieving strong empirical detection accuracy, existing detection-based defenses generally lack explicit control over the quality of accepted inputs—most notably the false acceptance rate (FAR), defined as the proportion of malicious inputs among all accepted inputs. Explicit FAR control is critical in safety-sensitive applications, where even a small number of false acceptances can lead to disproportionate harm. For example, when LLMs have access to sensitive or private information, a handful of successful jailbreak attacks may suffice to extract confidential data, resulting in severe privacy violations and substantial legal or financial risks. The absence of rigorous FAR guarantees therefore constitutes a fundamental barrier to deploying current detection-based defenses in high-stakes settings.

Motivated by this limitation, we study the problem of malicious input detection for LLMs under explicit FAR control in a realistic deployment setting. This setting introduces two key challenges. First, **the lack of malicious calibration samples**: in practice, defenders rarely have access to labeled malicious data in advance, as adversarial strategies continually evolve. As a result, classical false discovery rate (FDR) and local FDR (lFDR) control techniques from multiple hypothesis testing (Benjamini & Hochberg, 1995; Jin &

[1]Generative AI Lab, College of Computing and Data Science, Nanyang Technological University. Correspondence to: Yingjie Wang <yingjiewang1201@gmail.com>, Dacheng Tao <dacheng.tao@gmail.com>.

*Proceedings of the 43rd International Conference on Machine Learning*, Seoul, South Korea. PMLR 306, 2026. Copyright 2026 by the author(s).

Candes) are not directly applicable, since they rely on contaminated calibration data. Second, **the streaming nature of inputs**: test samples $\{X_t\}_{t \geq 1}$ arrive sequentially, and decisions must be made in real time without access to future data or the full test set. This makes method requiring offline density estimation unapplicable (Hu et al., 2026). Together, these challenges leave a critical gap: existing methods either require malicious calibration samples or require full access to future data, and thus fail to provide FAR guarantees in realistic, streaming LLM deployment scenarios.

To address this gap, we propose **OSCS** (**O**nline **S**election with **C**ontrolled **S**afety), a novel framework for online FAR control that operates without requiring access to malicious calibration samples. Given a benign calibration dataset $\{X_i\}_{i=1}^{n_0}$ and a stream of test inputs $\{X_t\}_{t \geq 1}$, OSCS tackles the challenges holistically by continuously learning malicious-related patterns from the test stream and adapting its decisions in real time. Specifically, OSCS leverages detection scores $\{W_t\}_{t \geq 1}$ produced by an existing scoring function $W(\cdot)$ to capture discrepancies between benign and malicious inputs. It then performs online density estimation over the test stream to estimate the posterior probability that each input is benign. Based on these estimates, OSCS makes real-time accept/reject decisions while provably controlling the FAR at a user-specified level.

We theoretically justify OSCS by showing that it controls the FAR up to an excess term that vanishes as the number of calibration samples, the score resolution, the time horizon, and the discriminative power of the detection score increase.

Empirically, we evaluate OSCS against two representative LLM safety threats—backdoor attacks and jailbreaking—across multiple datasets and attack scenarios. The results demonstrate that OSCS consistently achieves effective FAR control while maintaining strong detection performance, highlighting its suitability for controllable safety in realistic deployment settings.

In summary, our main contributions are:

- **Problem formulation:** We formalize the problem of online malicious detection for LLMs under explicit FAR control, addressing a key limitation of existing defenses.

- **Methodology and theory:** We propose OSCS, an online selection framework with controlled safety, and establish rigorous theoretical guarantees for FAR control with asymptotically vanishing error terms.

- **Empirical evaluation:** We validate OSCS on representative LLM safety threats, including backdoor attacks and jailbreaking, demonstrating its effectiveness and robustness across a wide range of settings.

## 2. Related Work

**Malicious Attacks against LLMs** Despite their impressive performance, LLMs remain vulnerable to malicious attacks that exploit weaknesses in their training or inference processes. Two prominent classes of such attacks are *backdoor attacks* and *jailbreak attacks*. Backdoor attacks embed hidden behaviors into models, causing them to produce attacker-specified outputs when inputs contain particular triggers, typically introduced through data poisoning. In NLP, these triggers can range from rare words (Chen et al., 2017; Yang et al., 2021a) and seemingly innocuous phrases (Dai et al., 2019; Yang et al., 2021c) to higher-level patterns such as stylistic features (Qi et al., 2021b) or syntactic structures (Qi et al., 2021c). Jailbreak attacks, on the other hand, do not modify model parameters but instead leverage adversarial prompts to bypass safety mechanisms. These attacks often rely on prompt engineering techniques, including token-level manipulations (Zou et al., 2023) and prompt-level optimization strategies (Chao et al., 2024). Even straightforward methods, such as suppressing refusal behaviors, have been shown to effectively compromise models (Wei et al., 2023). Both attack paradigms represent significant challenges for ensuring the safe and reliable deployment of LLMs in real-world applications.

**Malicious Detection for LLMs** Detection-based defenses aim to identify malicious inputs by performing classification at inference time. Existing methods construct discrepancy scores based on abnormal model activations (Yi et al., 2024), representation statistics (Chen et al., 2022; Xian et al., 2023), or robustness to perturbations (Gao et al., 2019; Yang et al., 2021b). For jailbreak detection, prior work has explored heuristic filters and auxiliary models to assess prompt harmfulness (Jain et al., 2023; Alon & Kamfonas, 2023; Phute et al., 2024; Dong et al., 2025). These approaches can achieve high classification accuracy but typically lack explicit guarantees on the quality of accepted inputs.

**Controllable Sample Selection** Our work is also related to sample selection methods with explicit error-rate control. Classical procedures such as Benjamini–Hochberg and its variants achieve FDR control via calibrated thresholds (Benjamini & Hochberg, 1995; Benjamini & Yekutieli, 2001; Storey et al., 2004), with recent extensions based on conformal prediction or local FDR estimates (Jin & Candes; Marandon et al., 2024; Bates et al., 2023; Gang et al., 2023; Huo et al., 2024; Wu et al., 2024). However, these methods generally require access to malicious samples for calibration, making them unsuitable for malicious detection of LLMs. Although Hu et al. (2026) eliminates the need for known malicious calibration data, it still depends heavily on offline access to the entire test set, rendering it unsuitable for online streaming scenarios where inputs arrive sequentially and storing all past data is impractical.

# 3. Preliminaries

In this section, we formalize the problem of online malicious input detection for LLMs under explicit FAR control. We first give the necessary notation used throughout the paper. Next, we introduce a unified attacker framework and show how common threats such as backdoor and jailbreak attacks fit within this abstraction. Finally, we describe the defender's capabilities and objectives in this setting.

## 3.1. Notation

We use uppercase letters (e.g., $X, Y$) to denote random variables and lowercase letters (e.g., $x, y$) for their realizations. For $n \in \mathbb{N}$, let $[n] = \{1, \ldots, n\}$. For a function $f : \mathbb{R} \to \mathbb{R}$, we denote its supremum norm by $\|f\|_\infty = \sup_{x \in \mathbb{R}} |f(x)|$ and its $L^2$ norm by $\|f\|_2 = (\int f^2(x) \, dx)^{1/2}$. Without other specifications, we use $i$ to index calibration samples and $t$ to index test-time samples.

## 3.2. Attacker Framework

**General Framework** We consider a general attacker model involving both *model side* and *sample side* attacks.

- **Model side.** Let $\mathcal{M} : \mathcal{X} \to \mathcal{Y}$ denote a benign LLM mapping inputs in $\mathcal{X}$ to outputs in $\mathcal{Y}$. On the model side, the adversary modifies the training process to embed malicious behaviors into the model, resulting in a compromised model $\mathcal{M}_{\mathrm{mal}}$.

- **Sample side.** At inference time, the attacker injects a *trigger* into an otherwise benign input to activate the malicious behavior. Formally, given a benign input $X \in \mathcal{X}$, the attacker constructs a malicious input

$$X_{\mathrm{mal}} = \mathcal{T}(X),$$

where $\mathcal{T} : \mathcal{X} \to \mathcal{X}_{\mathrm{mal}}$ is a transformation function. The transformed input induces the attacker's intended output $y_{\mathrm{mal}}$, i.e.,

$$\mathcal{M}_{\mathrm{mal}}(X_{\mathrm{mal}}) = y_{\mathrm{mal}}.$$

During inference, the attacker injects malicious inputs $X_{\mathrm{mal}} \in \mathcal{X}_{\mathrm{mal}}$ into the original test stream of the compromised model $\mathcal{M}_{\mathrm{mal}}$, forming a mixed test stream $\{X_t\}_{t \geq 1}$ that contains both benign inputs from $\mathcal{X}$ and malicious inputs from $\mathcal{X}_{\mathrm{mal}}$.

**Instantiations** Under this framework, both backdoor and jailbreak attacks can be viewed as specific instantiations:

- **Backdoor attacks.** Backdoor attacks involve both model-side and sample-side manipulation. On the model side, the attacker constructs $\mathcal{M}_{\mathrm{mal}}$ through techniques such as data

poisoning (Gu et al., 2019), weight manipulation (Yang et al., 2021c), or model editing (Li et al., 2023b). On the sample side, the attacker applies $\mathcal{T}$ by inserting predefined trigger tokens (Gu et al., 2019; Dai et al., 2019) or even stylistic elements (Qi et al., 2021b;c) into the input.

- **Jailbreak attacks.** In contrast, jailbreak attacks aim to exploit existing vulnerabilities in the model rather than modifying it, so they operate solely on the sample side. In this case, $\mathcal{T}$ generates adversarial prompts by appending optimized suffixes produced via gradient-based methods (Zou et al., 2023) or genetic algorithms (Liu et al., 2023).

## 3.3. Defender's Capacity and Objective

We consider a setting where the defender has access to a benign calibration dataset, denoted as $\mathcal{D}_{\mathrm{cal}} = \{X_i\}_{i=1}^{n_0}$, and observes a sequence of test inputs $\{X_t\}_{t \geq 1}$ in an online manner. The test stream may consist of both benign and malicious inputs. At each time step $t$, the goal of defender is to make an real-time accept/reject decision for the current input $X_t$, without access to future data, while ensuring that the FAR is controlled at a predefined level $\alpha$.

Formally, the FAR is defined as:

$$\mathrm{FAR}(\hat{\boldsymbol{\delta}}^T) = \mathbb{E}\left[ \frac{\sum_{t=1}^T \delta_t (1 - \hat{\delta}_t)}{\sum_{t=1}^T (1 - \hat{\delta}_t) \vee 1} \right], \qquad (1)$$

where $\delta_t \in \{0, 1\}$ represents the true malicious label of $X_t$, with $\delta_t = 1$ indicating that $X_t$ is malicious ($X_t \in \mathcal{X}_{\mathrm{mal}}$). Similarly, $\hat{\delta}_t \in \{0, 1\}$ denotes the defender's prediction, where $\hat{\delta}_t = 1$ indicates rejection (i.e., the sample is predicted to be malicious). $\hat{\boldsymbol{\delta}}^T = (\hat{\delta}_1, \ldots, \hat{\delta}_T)$ denotes the sequence of decisions up to time $T$ and $a \vee b$ represents the maximum of $a$ and $b$.

To rule out trivial strategies that reject all inputs, we also consider the *power* of the detector, defined as the proportion of benign inputs that are correctly accepted:

$$\mathrm{Power}(\hat{\boldsymbol{\delta}}^T) = \mathbb{E}\left[ \frac{\sum_{t=1}^T (1 - \delta_t)(1 - \hat{\delta}_t)}{\sum_{t=1}^T (1 - \delta_t) \vee 1} \right]. \qquad (2)$$

# 4. Methodology

In this section, we present the proposed OSCS framework in detail. We first derive probabilistic benignity estimates that enable decision-making with controllable FAR, and then establish their connection to online FAR control. Next, we discuss practical implementation considerations. Finally, we introduce an adaptive extension of OSCS that handles time-varying poisoning levels in the test stream.

## 4.1. Estimating Benign Probabilities

To make FAR-controllable decisions, we have to find a suitable statistic that quantifies "the benignity" of test inputs as the selection evidence. Here, we use the *posterior benign probability*, defined as the conditional probability that an input is benign given its observed score. However, directly calculate the probability using $X_t$ is intractable due to the discrete nature of text data and the curse of dimensionality.

To address this challenge, OSCS builds upon existing detection-based defenses that provide a real-valued scoring function $W(\cdot)$, which serves as a weak but informative signal for distinguishing benign from malicious inputs. Empirically, such scores tend to exhibit systematic distributional differences between benign and malicious samples, a property exploited by many prior works (Xian et al., 2023; Yi et al., 2024).

We begin by modeling the distribution of detection scores. Let $f_c$ and $f_p$ denote the density functions of the scores for benign and malicious inputs respectively, and let $f_{\mathrm{mix}}$ denote the overall distribution of scores in the test stream. Since the test stream contains both benign and malicious inputs, we model the distribution of detection scores in the test stream as a mixture:

$$f_{\mathrm{mix}}(w) = \pi_c f_c(w) + (1 - \pi_c) f_p(w), \qquad (3)$$

where $\pi_c$ denotes the (unknown) proportion of benign inputs in the test stream. Accordingly, we have the following generative model for the scores of calibration and test samples:

$$W_i \sim f_c, \quad W_t \sim f_{\mathrm{mix}}, \qquad (4)$$

where $W_i = W(X_i)$ are the scores of calibration samples and $W_t = W(X_t)$ are the scores of test inputs.

Under this mixture model, the probability that an input with score $w_t$ is benign can be computed via Bayes' rule:

$$L_t = L(w_t) = \Pr(\delta_t = 0 \mid w_t) = \frac{\pi_c f_c(w_t)}{f_{\mathrm{mix}}(w_t)}. \qquad (5)$$

This quantity coincides with the lFDR used in multiple hypothesis testing literature (Gang et al., 2023).

Intuitively, $L_t$ quantifies how likely the observed score $w_t$ is to originate from the benign distribution rather than the malicious one. Larger values of $L_t$ indicate higher confidence in benignity and thus favor acceptance, whereas smaller values suggest a higher probability of maliciousness and motivate rejection. Next, we formalize this intuition by connecting $L_t$ to FAR control.

## 4.2. Link to FAR Control

The following proposition provides the link between FAR and the posterior benign probabilities $L_t$, enabling explicit FAR control through appropriate selection of decisions $\hat{\boldsymbol{\delta}}^T$ based on $L_t$.

**Proposition 4.1** (The equivalence of FAR definitions)**.** *The definition of FAR in equation* (1) *is equivalent to*

$$\mathrm{FAR}(\hat{\boldsymbol{\delta}}^T) = \mathbb{E}\left[ \frac{\sum_{t \leq T}(1 - L_t)(1 - \hat{\delta}_t)}{\sum_{t \leq T}(1 - \hat{\delta}_t)} \right]. \qquad (6)$$

The proof is provided in Appendix B.6.

Based on this proposition, we control the FAR by selecting decisions $\hat{\boldsymbol{\delta}}^T = (\hat{\delta}_1, \dots, \hat{\delta}_T)$ such that

$$\frac{\sum_{t=1}^{T}(1 - \hat{\delta}_t)(1 - L_t)}{\sum_{t=1}^{T}(1 - \hat{\delta}_t) \vee 1} \leq \alpha, \qquad (7)$$

which enforces that the average probability of accepting a malicious input among all accepted inputs is bounded by the target level $\alpha$, thereby achieving explicit FAR control.

## 4.3. Practical Implementation

After linking $L_t$ to FAR control, the central challenge reduces to estimating $L_t$ accurately in an online manner. From the definition of $L_t$, this requires knowledge of $\pi_c$, $f_c$, and $f_{\mathrm{mix}}$, all of which are unknown in practice and must therefore be estimated from data.

**Estimating the benign distribution** $f_c$    To estimate $f_c$, we leverage the calibration dataset $\mathcal{D}_{\mathrm{cal}}$, which contains only benign samples. Given the scores of calibration samples $\{W_i\}_{i=1}^{n_0}$, a natural choice is to use the standard kernel density estimation (KDE):

$$\hat{f}_c(w) = \frac{1}{n_0} \sum_{i=1}^{n_0} K_{h_0}(w - W_i), \qquad (8)$$

where $K_{h_0}(\cdot) = \frac{1}{h_0} K\left(\frac{\cdot}{h_0}\right)$ is the kernel function with bandwidth $h_0$.

**Estimating the mix distribution** $f_{\mathrm{mix}}$    Estimating $f_{\mathrm{mix}}$ is more challenging, as the inputs arrive sequentially in a streaming manner. To accommodate this setting, we adopt recursive kernel density estimation (RKDE) to update the estimate of $f_{\mathrm{mix}}$ online. The key idea behind RKDE is to maintain kernel density estimates on a fixed grid and update them recursively as new observations arrive, thereby avoiding the need to store the entire data history. Specifically, we partition the scoring range into $B$ equal-width bins, forming grid points $G = \{g_1, g_2, \dots, g_B\}$. Without loss of generality, we assume the scores lie in $[-R, R]$. At time step $t$, after observing the score $W_t$ of the incoming input $X_t$, we update the estimate of $f_{\mathrm{mix}}$ on a grid $g$ as follows:

$$\hat{f}_{\mathrm{mix}}^{(t)}(g) = \lambda_t \hat{f}_{\mathrm{mix}}^{(t-1)}(g) + (1 - \lambda_t) K_{h_t}(g - W_t), \qquad (9)$$

where $\lambda_t = 1 - \frac{1}{t^\tau}$ with $\tau \in (0.5, 1)$ denotes the decay factor, and $h_t = t^{-\gamma}$ with $\gamma \in (0, \tau)$ denotes the bandwidth at time $t$.

To evaluate the estimated density at a score value $w$ that may not lie exactly on the grid, we apply linear interpolation between adjacent grid points. Specifically, suppose $w \in [g_j, g_{j+1}]$ for some $j \in [B - 1]$. Then,

$$\hat{f}_{\text{mix}}^{(t)}(w) = \hat{f}_{\text{mix}}^{(t)}(g_j) + \frac{\hat{f}_{\text{mix}}^{(t)}(g_{j+1}) - \hat{f}_{\text{mix}}^{(t)}(g_j)}{g_{j+1} - g_j}(w - g_j). \tag{10}$$

Unlike traditional KDE that requires storing all past observations, RKDE maintains only the current density estimate $\hat{f}_{\text{mix}}^{(t-1)}$ and updates it using the latest observation $W_t$. This results in a per-step time and space complexity of $O(B)$, making RKDE well suited for streaming scenarios with limited memory and computational resources.

**Estimating the benign proportion $\pi_c$** To estimate the benign proportion $\pi_c$, we adopt the approach of Storey et al. (2004). We first define an empirical p-value for each test input based on the calibration data:

$$\hat{p}_t = \frac{\sum_{i=1}^{n_0} \mathbf{1}\{G(W_i) \le G(W_t)\} + 1}{n_0 + 1} \tag{11}$$

where $G(\cdot)$ is a monotonic transformation of the scores that standardizes their interpretation, ensuring that higher values consistently indicate a greater probability of being benign. For example, if lower scores indicate a higher probability of being benign, we can define $G(w) = -w$.

Then, at time $t$, we estimate $\pi_c$ as:

$$\hat{\pi}_c^{(t)} = \frac{\sum_{t'=1}^{t} \mathbf{1}\{\hat{p}_{t'} > \lambda\}}{(1 - \lambda)t}, \tag{12}$$

where $\lambda \in (0, 1)$ is a predefined threshold. The intuition of this estimation is that since the $G(W_{t'})$ of benign inputs tend to be larger than those of malicious inputs, the samples with $\hat{p}_{t'} > \lambda$ are very likely to be benign. Moreover, since $\hat{p}_{t'}$ is uniformly distributed in $[0, 1]$ for benign inputs, the total number of benign samples in test stream can be approximated by $\frac{1}{1-\lambda}\sum_{t'=1}^{t} \mathbf{1}\{\hat{p}_{t'} > \lambda\}$, and thus the proportion of benign samples $\pi_c$ can be estimated accordingly.

Given the estimates $\hat{f}_c$, $\hat{f}_{\text{mix}}^{(t)}$ and $\hat{\pi}_c^{(t)}$, we estimate the posterior benign probability $L_t$ as:

$$\hat{L}_t = \Pi_{[0,1]}\left(\frac{\hat{\pi}_c^{(t)}\hat{f}_c(W_t)}{\hat{f}_{\text{mix}}^{(t)}(W_t)}\right). \tag{13}$$

where $\Pi_{[0,1]}(\cdot)$ denotes the projection operator that ensures the estimate lies within the valid probability range of $[0, 1]$.

---

**Algorithm 1** Overall Procedure of OSCS

---

**Require:** Calibration dataset $\mathcal{D}_{\text{cal}} = \{X_i\}_{i=1}^{n_0}$ (benign samples only); testing stream $\{X_t\}_{t \ge 1}$; kernel function $K$; number of grid points $B$; parameters $\tau, \lambda, R$; target FAR level $\alpha$; scoring function $W(\cdot)$.

**Ensure:** Sequential accept/reject decisions $\hat{\boldsymbol{\delta}}^T = \{\hat{\delta}_t\}_{t=1}^{T}$.

1: **Calibration Phase**
2: Compute scores $W_i = W(X_i)$ for all $X_i \in \mathcal{D}_{\text{cal}}$.
3: Estimate the benign score density $f_c$ using KDE based on $\{W_i\}_{i \in [n_0]}$, get $\hat{f}_c$.
4: **Initialization**
5: Construct a uniform grid $G = \{g_1, g_2, \ldots, g_B\}$ over $[-R, R]$.
6: Initialize the RKDE estimator:

$$\hat{f}_{\text{mix}}^{(0)}(g_j) = \frac{1}{2R}, \qquad j = 1, \ldots, B.$$

7: Set $U_0 = 0$ and $R_0 = 0$.
8: **Online Update and Decision**
9: **for** $t = 1, 2, \ldots$ **do**
10:    Observe $X_t$ and compute its score $W_t = W(X_t)$.
11:    Update the mixture density estimator $\hat{f}_{\text{mix}}^{(t)}$ using current score $W_t$ via Equation (9) and the estimated benign proportion $\hat{\pi}_c^{(t)}$ via Equation (12).
12:    Compute the estimated posterior benign probability $\hat{L}_t$ via Equation (13).
13:    **if** $\frac{U_{t-1} + (1 - \hat{L}_t)}{R_{t-1} + 1} > \alpha$ **then**
14:        **Reject** $X_t$: set $\hat{\delta}_t = 1$.
15:        Set $U_t = U_{t-1}$ and $R_t = R_{t-1}$.
16:    **else**
17:        **Accept** $X_t$: set $\hat{\delta}_t = 0$.
18:        Set $U_t = U_{t-1} + (1 - \hat{L}_t)$ and $R_t = R_{t-1} + 1$.
19:    **end if**
20: **end for**

---

Substituting $\hat{L}_t$ into (7) yields the practical selection rule. Specifically, we set $\hat{\delta}_t$ by ensuring

$$\frac{U_t}{R_t} = \frac{\sum_{t'=1}^{t}(1 - \hat{\delta}_{t'})(1 - \hat{L}_{t'})}{\sum_{t'=1}^{t}(1 - \hat{\delta}_{t'}) \vee 1} \le \alpha \tag{14}$$

The procedure of OSCS is summarized in Algorithm 1.

### 4.4. Extension to varying poison proportions

In practice, the proportion of malicious inputs in the test stream may change over time rather than remain constant. To address this challenge, Appendix D considers a time-varying poison proportion $1 - \pi_c^{(t)}$ and introduces an adaptive version of OSCS that continuously updates estimates of both $\pi_c^{(t)}$ and $f_{\text{mix}}^{(t)}$ through a sliding-window strategy. We further establish theoretical guarantees for the extension and validate its effectiveness through empirical studies.

## 5. Theoretical Analysis

In this section, we establish statistical guarantees for OSCS. We begin by introducing the assumptions required for our theoretical analysis, all of which are standard and mild in nonparametric density estimation and online decision-making. We then present our main theoretical result, showing that OSCS achieves explicit control of the FAR up to a vanishing excess term.

**Assumption 5.1.** The mixture density $f_{\mathrm{mix}}$ is uniformly bounded and admits a finite moment of order $K > 0$, i.e.,

$$\|f_{\mathrm{mix}}\|_\infty < \infty, \qquad \mathbb{E}|W|^K < \infty.$$

**Assumption 5.2** (Smoothness of target densities). The benign and mixture score densities $f_c$ and $f_{\mathrm{mix}}$ satisfy the following conditions. For any $R > 0$, there exist constants $0 < l_R \leq M < \infty$ such that

$$l_R \leq f_c(w) \leq M, \qquad l_R \leq f_{\mathrm{mix}}(w) \leq M, \qquad (15)$$

for all $|w| \leq R$.

In addition, $f_c$ and $f_{\mathrm{mix}}$ are Hölder continuous on $\mathbb{R}$: there exist exponents $0 < \beta_1, \beta_2 \leq 1$ and constants $c_{\beta_1}, c_{\beta_2} > 0$ such that

$$|f_c(x) - f_c(y)| \leq c_{\beta_1}|x - y|^{\beta_1}, \qquad (16)$$

for all $x, y \in \mathbb{R}$, and $f_{\mathrm{mix}}$ satisfies the analogous condition with parameters $\beta_2$ and $c_{\beta_2}$.

*Remark* 5.3. Assumption 5.2 is standard in nonparametric analysis. Local boundedness away from zero ensures that the posterior benign probability $L(w)$ is well-defined on compact sets and avoids degeneracy caused by vanishing denominators. The Hölder continuity condition is a classical smoothness assumption and includes Lipschitz continuous densities as a special case ($\beta_1 = \beta_2 = 1$).

**Assumption 5.4** (Kernel regularity). The kernel $K : \mathbb{R} \to \mathbb{R}$ is bounded and square-integrable, i.e., $\|K\|_\infty < \infty$ and $\|K\|_2 < \infty$, and satisfies the moment condition

$$\int K(u)\,|u|^\beta\,du < \infty \quad \text{for all } \beta > 0.$$

*Remark* 5.5. Assumption 5.4 is standard in density estimation convergence analysis (Jiang, 2017) and is satisfied by commonly used kernels, including Gaussian, Epanechnikov, and Laplace kernels.

**Assumption 5.6** (Separation of contaminated scores). There exists a threshold $\lambda \in (0, 1)$ such that

$$\Pr\big(p_j > \lambda \mid \delta_j = 1\big) \leq \epsilon_\lambda. \qquad (17)$$

*Remark* 5.7. Assumption 5.6 formalizes a mild separation condition between benign and contaminated inputs. It requires that contaminated scores do not frequently exceed

a high quantile of the benign score distribution. The parameter $\epsilon_\lambda$ quantifies the degree of overlap between the two distributions. This assumption is necessary because OSCS operates without access to malicious calibration samples and must therefore rely on weak distributional separation. Similar conditions are standard in the multiple testing and false discovery rate literature (Storey, 2002; Storey et al., 2004). In practice, the assumption holds as long as the scoring function $W(\cdot)$ exhibits nontrivial discriminative power.

**Assumption 5.8** (Selection process). Let $(\mathcal{F}_t)_{t \geq 0}$ be a filtration. Let

$$q_t := \mathbb{E}[1 - \hat{\delta}_t \mid \mathcal{F}_{t-1}], \qquad \mu_T := \sum_{t=1}^{T} q_t. \qquad (18)$$

Assume there exist constants $c > 0$ and $t_0$ such that $\mu_T \geq cT$ for all $T \geq t_0$.

*Remark* 5.9. Assumption 5.8 ensures that the expected number of accepted samples grows linearly with time, ruling out degenerate regimes in which nearly all inputs are rejected. It is mild in that it only requires the long-run acceptance rate to be bounded away from zero.

**Theorem 5.10** (FAR control). *Under the stated assumptions, the FAR of the decision sequence $\hat{\boldsymbol{\delta}}^T$ produced by Algorithm 1 satisfies*

$$\mathrm{FAR}\big(\hat{\boldsymbol{\delta}}^T\big) \leq \alpha + \Delta_{n_0,T,B,R},$$

*where the excess term $\Delta_{n_0,T,B,R}$ is given by*

$$\Delta_{n_0,T,B,R} = C_1 n_0^{-\frac{\beta_1}{2\beta_1+1}} \sqrt{\log n_0} + C_2 T^{-\frac{\tau\beta_2}{2\beta_2+1}} \sqrt{\log(TB)}$$
$$+ C_3 B^{-\beta_2} + C_4 R^{-K} + C_5 \epsilon_\lambda, \qquad (19)$$

*for constants $C_1, C_2, C_3, C_4, C_5 > 0$.*

*Remark* 5.11. Theorem 5.10 shows that OSCS controls the FAR at the target level $\alpha$ up to an excess term $\Delta_{n_0,T,B,R}$, which vanishes asymptotically. The first term in (19) arises from the estimation error of the benign density $f_c$ based on the finite calibration dataset. The second and third terms capture the error in estimating the mixture density $f_{\mathrm{mix}}$ via recursive KDE, reflecting respectively the finite-sample nature of the data stream and the discretization error induced by the grid approximation. The fourth term corresponds to the tail truncation error introduced by restricting the analysis to the compact interval $[-R, R]$. Under the finite-moment condition in Assumption 5.1, this term decays polynomially in $R$. The choice of $R$ therefore reflects a trade-off between tail control and the local lower-bound condition in Assumption 5.2. The final term accounts for imperfect separation between benign and contaminated score distributions. Although the constants may depend on $R$ through the local lower bound $l_R$, $R$ is treated as fixed and therefore does not affect the asymptotic rates.

*Table 1.* FAR control results for the BERT-base-uncased model. The numbers presented are the average of 5 independent experiments. We bold the method that controls the FAR (FAR $\leq 0.05$), or the method with the lowest FAR if no method achieves successful control.

| | Yelp | | | | | | | | AGNews | | | | | | | | HSOL | | | | | | | |
|---|---|---|---|---|---|---|---|---|---|---|---|---|---|---|---|---|---|---|---|---|---|---|---|---|
| | BadNets | | AddSent | | StyleBkd | | SynBkd | | BadNets | | AddSent | | StyleBkd | | SynBkd | | BadNets | | AddSent | | StyleBkd | | SynBkd | |
| | FAR | Power | FAR | Power | FAR | Power | FAR | Power | FAR | Power | FAR | Power | FAR | Power | FAR | Power | FAR | Power | FAR | Power | FAR | Power | FAR | Power |
| BKI | 0.158 | 1.000 | 0.220 | 0.734 | 0.296 | 0.594 | 0.080 | 1.000 | 0.156 | 1.000 | 0.087 | 1.000 | 0.157 | 0.839 | 0.188 | 0.815 | 0.157 | 1.000 | 0.208 | 0.880 | 0.215 | 0.866 | 0.213 | 0.881 |
| CUBE | 0.322 | 0.526 | 0.219 | 0.894 | 0.308 | 0.517 | 0.200 | 0.999 | 0.250 | 0.749 | 0.250 | 0.748 | 0.203 | 0.978 | 0.200 | 0.998 | **0.001** | **0.940** | 0.387 | 0.347 | 0.255 | 0.602 | 0.200 | 0.999 |
| STRIP | 0.195 | 0.886 | 0.192 | 0.885 | 0.192 | 0.894 | 0.193 | 0.923 | 0.194 | 0.868 | 0.196 | 0.883 | 0.199 | 0.924 | 0.193 | 0.828 | 0.194 | 0.889 | 0.197 | 0.885 | 0.198 | 0.919 | 0.197 | 0.901 |
| RAP | 0.208 | 0.941 | **0.000** | **0.622** | 0.187 | 0.951 | 0.229 | 0.821 | 0.203 | 0.975 | 0.186 | 0.937 | 0.203 | 0.975 | 0.178 | 0.906 | 0.205 | 0.962 | 0.141 | 0.971 | 0.199 | 0.968 | 0.221 | 0.623 |
| SCM-md | 0.090 | 0.924 | 0.206 | 0.934 | 0.190 | 0.924 | **0.036** | **0.944** | 0.208 | 0.951 | 0.208 | 0.950 | 0.202 | 0.955 | 0.210 | 0.938 | 0.141 | 0.940 | 0.209 | 0.948 | 0.196 | 0.951 | 0.201 | 0.950 |
| SCM-badacts | 0.116 | 0.995 | 0.191 | 0.994 | 0.182 | 0.996 | 0.200 | 0.996 | 0.178 | 0.999 | 0.065 | 0.999 | 0.084 | 1.000 | 0.128 | 0.999 | 0.188 | 0.998 | 0.200 | 1.000 | 0.199 | 0.999 | 0.161 | 0.999 |
| OSCS-md | **0.028** | **0.891** | **0.017** | **0.658** | **0.022** | **0.781** | **0.013** | **0.837** | **0.033** | **0.720** | **0.046** | **0.780** | **0.048** | **0.749** | **0.040** | **0.537** | **0.014** | **0.744** | 0.053 | 0.777 | 0.056 | 0.452 | **0.042** | **0.598** |
| OSCS-badacts | **0.009** | **0.853** | 0.056 | 0.966 | 0.056 | 0.918 | **0.028** | **0.916** | **0.021** | **0.936** | 0.062 | 0.996 | 0.061 | 0.984 | **0.021** | **0.910** | **0.033** | **0.956** | **0.043** | **0.868** | **0.036** | **0.835** | **0.047** | **0.947** |

*Table 2.* FAR control results for backdoor sample detection on RoBERTa-base model. The settings are the same as those in Table 1.

| | Yelp | | | | | | | | AGNews | | | | | | | | HSOL | | | | | | | |
|---|---|---|---|---|---|---|---|---|---|---|---|---|---|---|---|---|---|---|---|---|---|---|---|---|
| | BadNets | | AddSent | | StyleBkd | | SynBkd | | BadNets | | AddSent | | StyleBkd | | SynBkd | | BadNets | | AddSent | | StyleBkd | | SynBkd | |
| | FAR | Power | FAR | Power | FAR | Power | FAR | Power | FAR | Power | FAR | Power | FAR | Power | FAR | Power | FAR | Power | FAR | Power | FAR | Power | FAR | Power |
| BKI | 0.158 | 1.000 | **0.018** | **1.000** | 0.435 | 0.325 | 0.056 | 1.000 | 0.156 | 1.000 | 0.120 | 1.000 | 0.150 | 0.853 | 0.144 | 0.995 | 0.157 | 1.000 | 0.206 | 0.884 | 0.216 | 0.859 | 0.216 | 0.877 |
| CUBE | **0.001** | **0.997** | **0.008** | **0.999** | **0.004** | **0.899** | 0.309 | 0.515 | 0.250 | 0.749 | 0.250 | 0.749 | 0.238 | 0.768 | 0.200 | 0.999 | 0.069 | 0.821 | 0.372 | 0.403 | 0.256 | 0.723 | 0.224 | 0.868 |
| STRIP | 0.196 | 0.908 | 0.196 | 0.900 | 0.185 | 0.863 | 0.193 | 0.870 | 0.194 | 0.858 | 0.196 | 0.896 | 0.194 | 0.844 | 0.190 | 0.773 | 0.195 | 0.810 | 0.193 | 0.849 | 0.199 | 0.884 | 0.190 | 0.817 |
| RAP | 0.341 | 0.484 | 0.213 | 0.921 | 0.214 | 0.912 | 0.228 | 0.844 | 0.218 | 0.892 | 0.210 | 0.943 | 0.216 | 0.904 | 0.229 | 0.840 | 0.204 | 0.938 | 0.209 | 0.944 | 0.098 | 0.752 | 0.212 | 0.931 |
| SCM-md | 0.209 | 0.935 | 0.180 | 0.935 | **0.005** | **0.940** | 0.206 | 0.945 | 0.209 | 0.946 | 0.206 | 0.951 | 0.209 | 0.942 | 0.132 | 0.946 | 0.195 | 0.952 | 0.199 | 0.933 | 0.209 | 0.949 | | |
| SCM-badacts | 0.056 | 0.994 | 0.199 | 0.996 | **0.040** | **0.997** | **0.001** | **0.998** | 0.198 | 0.999 | 0.200 | 1.000 | 0.168 | 0.999 | 0.162 | 0.999 | **0.025** | **0.999** | 0.200 | 0.998 | 0.190 | 0.998 | 0.172 | 1.000 |
| OSCS-md | **0.024** | **0.768** | **0.033** | **0.723** | **0.018** | **0.881** | **0.027** | **0.904** | **0.022** | **0.424** | **0.026** | **0.599** | 0.056 | 0.596 | **0.008** | **0.591** | 0.053 | 0.861 | **0.025** | **0.470** | 0.056 | 0.414 | **0.039** | **0.774** |
| OSCS-badacts | **0.027** | **0.956** | **0.022** | **0.912** | **0.024** | **0.946** | **0.046** | **0.997** | **0.030** | **0.949** | **0.048** | **0.966** | 0.059 | 0.821 | **0.015** | **0.889** | **0.011** | **0.867** | 0.051 | 0.893 | 0.060 | 0.965 | **0.008** | **0.814** |

## 6. Experiments

In this section, we empirically evaluate the effectiveness and generalizability of OSCS across two LLM safety tasks: **backdoor sample detection** and **jailbreak prompt detection**[1]. For both tasks, we assume that the defender has access only to a small set of benign calibration data and must process a sequential test stream containing an unknown proportion of malicious inputs. Unless otherwise specified, we set the target FAR level to $\alpha = 0.05$. Additional experimental settings can be found in Appendix A.1.

### 6.1. Task 1: Backdoor Sample Detection

**Experimental Setup.** We evaluate OSCS on three widely used text classification benchmarks: **Yelp** (Yelp, 2024) for sentiment analysis, **AG News** (Zhang et al., 2015) for topic classification, and **HSOL** (Davidson et al., 2017) for toxic content detection. Experiments are conducted on two victim models, **BERT-base** (Devlin et al., 2019) and **RoBERTa-base** (Liu et al., 2019). We consider four representative backdoor attacks covering diverse trigger mechanisms: **BadNets** (Gu et al., 2019), **AddSent** (Dai et al., 2019), **StyleBkd** (Qi et al., 2021b), and **SynBkd** (Qi et al., 2021c), with 20% of the training data poisoned for all attacks. We compare our method, **OSCS**, against five representative backdoor detection defenses: **STRIP** (Gao et al., 2019), **BKI** (Chen & Dai, 2021), **RAP** (Yang et al., 2021b), **CUBE** (Cui et al., 2022), and **SCM** (Xian et al., 2023). Methods originally designed for training-time detection are adapted to inference-time filtering, and all baselines follow their original threshold selection protocols. Detection scores are computed using either Mahalanobis distance-based scores (**MDS**) or activation-based scores from **BadActs** (Yi et al., 2024). For methods requiring statistical guarantees, including OSCS and SCM, we report results using both scoring functions, denoted by the suffixes "-md" and "-badacts", respectively. We set the time horizon set to $T = 20000$ and repeat each experiment 5 times. More experimental setups can be found in Appendix A.2.

#### 6.1.1. RESULTS

**Overall FAR Control Performance** In this experiment, we randomly poison 20% of sample in the test stream and evaluate OSCS against several baseline defenses across multiple attack scenarios and three datasets. The FAR at the final time step $T$, i.e., $\mathrm{FAR}(\hat{\boldsymbol{\delta}}^T)$, for the BERT-base-uncased and RoBERTa-Base model is presented in Table 1 and Table 2, respectively. From the results, we make the following observations. First, OSCS consistently achieves (or nearly achieves) the target FAR level at the final time step across most scenarios, demonstrating its ability to reliably regulate FAR in an online setting. Second, although some baselines, such as BKI, CUBE, and SCM, perform well under specific attack types or datasets, they fail to maintain consistent FAR control across diverse settings. This inconsistency highlights the limitations of existing detection-based defenses in providing robust and controllable protection against backdoor attacks. Notably, SCM, which is specifically designed to control the false rejection rate of benign samples, also struggles to achieve stable FAR control. This observation highlights the need for methods explicitly designed to provide online FAR guarantees, such as OSCS.

**OSCS's Performance over Time** To better illustrate the ability of OSCS to control FAR in online settings, we track its FAR over time under the same experimental settings and present the results on the Yelp dataset using RoBERTa-base

---

[1]Our experimental code can be found at https://github.com/huzr1999/OSCS.

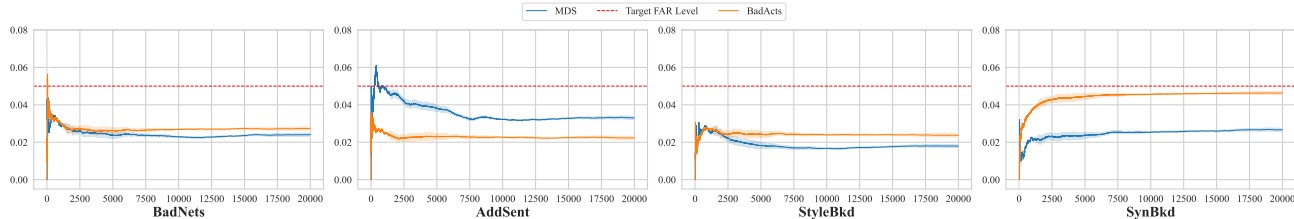

*Figure 1.* The FAR control performance of OSCS over time. X-axis represents the time step and Y-axis represents the empirical FAR up to that time step. The shadow area represents the standard deviation over 5 runs.

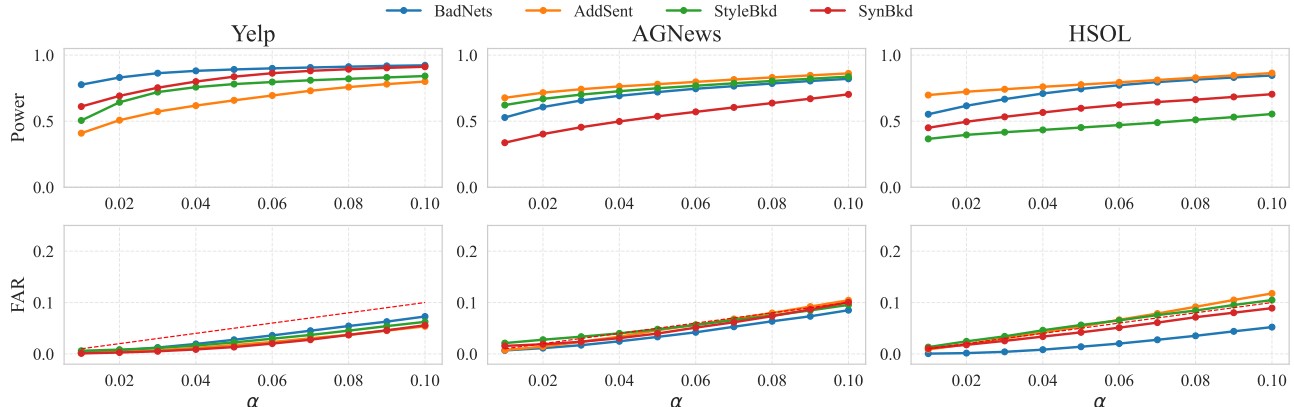

*Figure 2.* The FAR and Power of OSCS under varying FAR threshold $\alpha$. The red dashed line indicates the target FAR level.

in Figure 1. The area below the red dashed line corresponds to successful FAR control. From the figure, we make the following observations. First, although OSCS may exhibit temporary FAR fluctuations in the early stages (e.g., around $t = 2000$) due to limited data for estimating the mixture distribution, its performance gradually stabilizes as more observations become available. Over time, OSCS consistently maintains FAR below the target level, demonstrating its adaptability to streaming data and its ability to regulate FAR reliably in an online setting. Second, while different scoring functions may lead to slight variations in FAR trajectories, OSCS consistently achieves stable FAR control across all scoring functions, highlighting its robustness and versatility under diverse detection scenarios.

**FAR Control under Different Thresholds**  We further evaluate the performance of OSCS under different safety requirements by varying the FAR threshold $\alpha$. Specifically, we vary $\alpha$ from 0.01 to 0.1 and report the corresponding power and FAR of BERT-base in Figure 2. The results show that OSCS consistently maintains FAR below the target level across most scenarios, even under stringent settings such as $\alpha = 0.01$. These results demonstrate the adaptability of OSCS to diverse safety requirements.

### 6.2. Task 2: Jailbreaking Detection

**Experimental Setup**  In this experiment, following prior works (Zheng et al., 2024; Zou et al., 2023; Liu et al.,

2023), we utilize **AdvBench** (Zou et al., 2023), a benchmark dataset for eliciting harmful responses from LLMs. For the victim LLMs, we evaluate our approach on **Mistral-7B-Instruct** (Jiang et al., 2023), **Qwen2-7B-Instruct** (Yang et al., 2024), and **Llama3.1-8B-Instruct** (Grattafiori et al., 2024). For jailbreak attacks, we focus on two state-of-the-art attack methods, **AutoDAN-ga** and **AutoDAN-hga** (Liu et al., 2023). Building on the detection framework, we integrate OSCS with two scoring functions: **PPL** (perplexity score) (Jain et al., 2023) and **MDS** (Mahalanobis distance-based score) (Dong et al., 2025). The resulting OSCS variants are denoted with the suffix "-ppl" and "-md," respectively. Similar to the backdoor detection setting, we include SCM as a baseline since it shares a similar objective of controlling the quality of accepted samples. Here, we set the time horizon to $T = 5000$ and repeat experiments 10 times. More details can be found in Appendix A.3.

#### 6.2.1. RESULTS

**Overall performance**  Similar to the backdoor detection setting, we conduct experiments with a jailbreak ratio of 20% to further demonstrate the ability of OSCS to control FAR across different LLM safety tasks. The results are presented in Table 3. From the table, we observe that OSCS consistently achieves effective FAR control across different attack scenarios and victim models. Note that under certain settings, the baseline SCM can achieve FAR control, but

*Table 3.* The FAR control results for jailbreaking detection on three LLMs under two jailbreak attacks. The numbers presented are the average and standard error of 10 independent experiments. We bold the method that controls the FAR ($FAR \leq 0.05$), or the method with the lowest FAR if no method achieves successful control.

| | Mistral-7B-Instruct-v0.3 | | | | Qwen2-7B-Instruct | | | | Llama-3.1-8B-Instruct | | | |
| | AutoDAN-ga | | AutoDAN-hga | | AutoDAN-ga | | AutoDAN-hga | | AutoDAN-ga | | AutoDAN-hga | |
| | FAR | Power | FAR | Power | FAR | Power | FAR | Power | FAR | Power | FAR | Power |
|---|---|---|---|---|---|---|---|---|---|---|---|---|
| SCM-ppl | 0.136±0.003 | 0.899±0.002 | 0.133±0.003 | 0.899±0.002 | 0.120±0.003 | 0.899±0.002 | 0.129±0.003 | 0.899±0.002 | 0.159±0.003 | 0.899±0.002 | 0.154±0.002 | 0.899±0.002 |
| SCM-md | 0.142±0.003 | 0.996±0.001 | 0.141±0.003 | 0.996±0.001 | 0.000±0.000 | 0.924±0.003 | 0.000±0.000 | 0.924±0.003 | 0.166±0.002 | 0.919±0.003 | 0.168±0.001 | 0.919±0.003 |
| OSCS-ppl | **0.013±0.004** | **0.711±0.043** | **0.014±0.005** | **0.710±0.042** | **0.014±0.006** | **0.717±0.046** | **0.016±0.006** | **0.714±0.046** | **0.011±0.006** | **0.708±0.038** | **0.012±0.006** | **0.710±0.038** |
| OSCS-md | **0.044±0.006** | **0.929±0.014** | **0.044±0.006** | **0.929±0.013** | **0.047±0.001** | **0.990±0.001** | **0.047±0.001** | **0.989±0.001** | **0.044±0.003** | **0.684±0.014** | 0.051±0.003 | 0.681±0.014 |

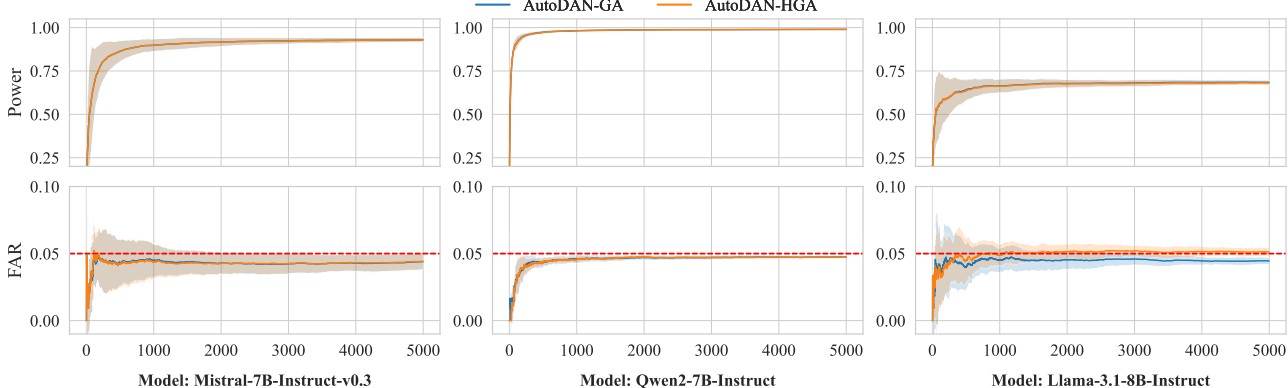

*Figure 3.* The FAR control performance of OSCS over time in jailbreaking detection scenarios. The upper and lower part correspond to Power and FAR, respectively. The shadow area represents the standard deviation over 10 runs.

this only happens coincidentally rather than by design, as SCM is originally designed to control the false rejection rate (FRR) instead of FAR. In contrast, OSCS is explicitly designed to regulate FAR, demonstrating its robustness and reliability in maintaining the desired FAR level across diverse conditions.

**OSCS's performance over time**    To further validate the generalizability of OSCS for online FAR control, we track its FAR over time for jailbreak prompt detection under experimental settings similar to those used in backdoor detection. The results for MDS are presented in Figure 3, where the area below the red dashed line corresponds to successful FAR control. From the figure, we observe that OSCS consistently maintains FAR below the target level throughout most of the testing period across three victim models and two attack methods. This demonstrates the robustness of OSCS in streaming settings and its ability to reliably regulate FAR over time for jailbreak prompt detection.

## 7. Conclusion

In this paper, we studied malicious input detection for LLMs under explicit FAR control in realistic online deployment settings. To this end, we proposed OSCS, an online framework that transforms detection scores into probabilistic benignity estimates and enables real-time accept/reject decisions under user-specified FAR constraints. We theoretically establish FAR control guarantees for OSCS under mild as-

sumptions and empirically validate its effectiveness across both backdoor and jailbreak attack scenarios. These results demonstrate that OSCS provides a practical and principled solution for safety-critical LLM applications.

## Impact Statement

This work aims to enhance the safe deployment of LLMs by providing explicit and provable FAR control over malicious inputs in realistic online settings, without requiring malicious calibration data. However, improper parameter choices in OSCS may lead to overly conservative filtering, potentially reducing usability and raising fairness concerns across different user groups (Pessach & Shmueli, 2022; Zhang et al., 2025b; Wang et al., 2025). Moreover, adversaries aware of the OSCS mechanism may adopt adaptive strategies, such as dynamically varying poisoning rates or malicious sample distributions, which could degrade the framework's FAR control performance. In addition, large-scale real-world deployments may exhibit complex distribution shifts (Arakelyan et al., 2023; Li et al., 2023a; Shu & Yu, 2024) that are not fully captured by the assumptions considered in our analysis. Evaluating OSCS under a broader range of deployment scenarios and attack models therefore remains an important direction for future research. To mitigate these risks, future work should focus on robust parameter selection, fairness auditing, and improving resilience against adaptive attacks through continuous monitoring.

# Acknowledgements

This research / project is supported by the National Research Foundation, Singapore, and Cyber Security Agency of Singapore under its National Cybersecurity R&D Programme and CyberSG R&D Cyber Research Programme Office. Any opinions, findings and conclusions or recommendations expressed in these materials are those of the author(s) and do not reflect the views of National Research Foundation, Singapore, Cyber Security Agency of Singapore as well as CyberSG R&D Programme Office, Singapore.

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

# A. Omitted Details of Experimental Setup

In this section, we provide additional details regarding the experimental setup for both backdoor sample detection and jailbreak prompt detection tasks. First we describe the common hyperparameter settings shared by both tasks, followed by specific implementation details for each task.

## A.1. Common Hyperparameter Settings Shared by two Tasks

We use a Gaussian kernel as the kernel function $K$ for both KDE and RKDE. For both methods, the bandwidth is selected using Silverman's rule of thumb. Although the theoretical analysis adopts the standard shrinking bandwidth schedule $h_t = t^{-\gamma}$ for analytical tractability, in practice we employ Silverman's rule-of-thumb bandwidth computed from the calibration set to improve finite-sample empirical stability. We set the decay parameter to $\tau = 0.95$. The score range is estimated from the calibration data and expanded to five times the observed score range in the calibration set. To ensure sufficient numerical precision, the number of grid points is chosen such that the grid spacing is $0.01$.

## A.2. Experimental Settings for Task1 (Backdoor Sample Detection)

**Implementation Details**   We evaluate backdoor sample detection using four representative backdoor attack techniques: **BadNets** (Gu et al., 2019), which introduces rare-word triggers (e.g., "bf"); **AddSent** (Dai et al., 2019), which appends full-sentence triggers (e.g., "I watched this 3D movie"); **StyleBkd** (Qi et al., 2021b), which applies stylistic transformations (e.g., a "Bible" writing style) as triggers; and **SynBkd** (Qi et al., 2021c), which uses specific syntactic patterns (e.g., "S(SBAR)(,)(NP)(VP)(.)") as triggers. For all attack methods, 20% of the training data is poisoned. The victim models are trained with backdoors for 5 epochs using a learning rate of $2 \times 10^{-5}$. All attack methods and baseline defenses are implemented using the **OpenBackdoor** framework (Cui et al., 2022), with default hyperparameter settings for both attack and defense methods. The experiments are conducted using **PyTorch** (Paszke et al., 2019) and the **Transformers** library (Wolf et al., 2020). We use the AdamW optimizer and run all experiments on a single NVIDIA V100 GPU with 32GB of memory. Each experiment is repeated 5 times, and we report the average results.

**Scoring Function**   In our experiments, we use two scoring functions: Mahalanobis distance-based scores (MDS), which measure the distance of a sample's feature representation from the distribution of clean samples, and activation-based scores (**BadActs**, Yi et al., 2024), which capture deviations in model activations caused by potential backdoor triggers. Both scoring functions require an additional clean calibration set for score construction. To this end, we split the original calibration set evenly into two disjoint subsets. The first subset is used to estimate the necessary statistics—namely, the mean and covariance for MDS, and the clean activation ranges for BadActs. The second subset serves as the calibration set for evaluation, as described in Section 4.

- **Mahalanobis Distance (MDS):** Using the first half of the calibration set, we estimate the mean vector $\mu$ and covariance matrix $\Sigma$ of feature representations extracted from clean samples. For a test input $x_j$, the Mahalanobis distance is computed as

$$W_{\text{Mahalanobis}}(x_j) = \sqrt{(f(x_j) - \mu)^\top \Sigma^{-1}(f(x_j) - \mu)}, \tag{20}$$

   where $f(x_j)$ denotes the feature representation of $x_j$. This score measures the deviation of a test sample from the distribution of clean samples in the feature space. For this scoring function, we set $G(w) = -w$ so that smaller transformed scores correspond to a higher probability of being a backdoor sample.

- **BadActs Score:** We follow the scoring procedure introduced in BadActs (Yi et al., 2024). Specifically, for each test sample $x_j$, the intermediate indicator for the $k$-th activation value is defined as

$$\Phi^k(x_j) = \mathbb{1}_{[\mu^k - a\sigma^k, \, \mu^k + a\sigma^k]}(r_{j,k}),$$

where $\mu^k$ and $\sigma^k$ denote the mean and standard deviation of the $k$-th activation estimated from clean samples, $r_{j,k}$ is the corresponding activation of $x_j$, and $a$ is a hyperparameter set to 3 in our experiments.

The final BadActs score is obtained by averaging over all activation values:

$$W_{\text{BadActs}}(x_j) = \frac{1}{L \cdot d} \sum_{k=1}^{L \cdot d} \Phi^k(x_j),$$

*Table 4.* Glossary of commonly used symbols.

| Symbol | Meaning | Symbol | Meaning |
|---|---|---|---|
| $\hat{f}_{\mathrm{mix}}^{(t)}$ $(\hat{f}_t)$ | Recursive estimator of the mixture density $f_{\mathrm{mix}}$ | $g_j$ | $j$-th grid point |
| $f_c$, $f_{\mathrm{mix}}$ | True density of clean/mix distribution | $L$, $\hat{L}_t$ | True posterior benign probability and its estimate |
| $K$ | Kernel function | $R$ | Radius of the compact interval |
| $w_{t,i}$ | Weight assigned to the $i$-th kernel at time $t$ | $h_i$ | Bandwidth of the $i$-th kernel |
| $\lambda_t$ | Decay factor at time $t$ | $\gamma$ | The power of $h_i = i^{-\gamma}$ |
| $\tau$ | The power of $\lambda_t = 1 - \frac{1}{t^\tau}$ | $\pi_c$, $\hat{\pi}_c^{(t)}$ | True/estimated benign proportion |

where $L \cdot d$ is the total number of activation values across all layers. This score represents the fraction of activations that fall within the expected clean range. For this score, we set $G(w) = w$.

### A.3. Experimental Settings for Task 2 (Jailbreaking Detection)

**Attack Methods: AutoDAN**  AutoDAN is a recently proposed automated jailbreaking method that systematically searches for adversarial prompts capable of bypassing LLM safety mechanisms. Specifically, it uses genetic algorithms to evolve prompts over multiple generations, optimizing for those that successfully elicit harmful responses from the target LLM. AutoDAN-ga employs a standard genetic algorithm approach in paragraph-level population while AutoDAN-hga additionally incorporates sentence-level population searching. Both methods have demonstrated high success rates in generating effective jailbreak prompts across various LLMs (Liu et al., 2023).

**Scoring Function**  In our experiments, we utilize the perplexity (PPL) and Mahalanobis (MDS) score as the scoring function for jailbreak sample detection. The perplexity score measures how well a language model predicts a given input sequence, with higher perplexity indicating that the input is less likely under the model's learned distribution. Specifically, for a given input prompt $X_t$, the perplexity score is computed as:

$$W_{\mathrm{PPL}}(X_t) = \exp\left(-\frac{1}{N}\sum_{i=1}^{N}\log P(X_{t,i}|X_{t,<i})\right),$$

where $N$ is the length of the prompt, $X_{t,j}$ is the $j$-th token in the prompt, and $P(X_{t,i}|X_{t,<i})$ is the conditional probability of token $X_{t,i}$ given the preceding tokens $X_{t,<i}$ as predicted by the language model. The mahalanobis score is computed similarly to the backdoor detection setting.

## B. Proofs

In this section, we provide detailed proofs of the theoretical results presented in Section 5. The proofs are organized as follows.

1. **Useful Lemmas (B.1):** We present several auxiliary lemmas that will be instrumental in establishing the main results.

2. **Proof of the Uniform Convergence of RKDE on Compact Intervals (B.2):** We provide the lemmas and proof of Theorem B.12, which establishes the uniform convergence of the RKDE estimator on compact intervals.

3. **Proof of the Convergence of the Proportion Estimator (B.3):** We present the lemmas and proof of Theorem B.15, which establishes the consistency of the estimator for the proportion of clean samples.

4. **Proof of the Uniform Convergence of the L-estimator (B.4):** Building on Theorems B.12 and B.15, we provide the proof of Theorem B.16, which establishes the uniform convergence of the L-estimator for posterior benign probabilities.

5. **Proof of the Main Theorem (B.5 and B.6):** Finally, we present the necessary lemmas and the proof of Theorem 5.10, which shows that OSCS controls the FAR at the desired level under suitable conditions.

To facilitate the presentation of the proofs, we first summarize the commonly used notations in Table 4.

## B.1. Useful Lemmas

**Lemma B.1** (Uniform Convergence of KDE (Jiang, 2017)). *If Assumption 5.2 holds and we take the bandwidth $h = n_0^{-1/(2\beta+1)}$, then with probability at least $1 - 1/n_0$,*

$$\sup_{w \in \mathbb{R}} |\hat{f}(w) - f(w)| \le C n_0^{-\frac{\beta}{2\beta+1}} \sqrt{\log n_0},$$

*where $C = C(M, c_\beta, \beta, K)$ is a positive constant depending on $M$ and $\beta, c_\beta$ of Hölder continuity and the kernel $K(\cdot)$.*

**Lemma B.2** (Hoeffding's Inequality). *Let $X_1, \ldots, X_n$ be independent random variables with $X_i \in [l_{\text{low}}, l_{\text{low}} + L]$ almost surely. Define the sample mean $\bar{X}_n = \frac{1}{n} \sum_{i=1}^{t} X_i$ and $\mu = \mathbb{E}[\bar{X}_n]$. Then for any $\delta > 0$, with probability at least $1 - \delta/2$,*

$$|\bar{X}_n - \mu| \le L \cdot \sqrt{\frac{\log(4/\delta)}{2n}}. \tag{21}$$

**Lemma B.3** (Bernstein's Inequality). *Let $Z_1, \ldots, Z_n$ be independent random variables such that for all $i \in \{1, \ldots, n\}$, they are almost surely bounded and zero-mean:*

$$|Z_i| \le M_n \qquad \text{and} \qquad \mathbb{E}[Z_i] = 0$$

*Let $S_t = \sum_{i=1}^{t} Z_i$ be the sum, and let $V_n$ be the sum of the variance:*

$$V_n = \sum_{i=1}^{t} \text{Var}[Z_i] = \sum_{i=1}^{t} \mathbb{E}[Z_i^2]$$

*For any confidence level $\delta \in (0, 1)$, with probability at least $1 - \delta$, the deviation of the sum $|S_t|$ is bounded by:*

$$|S_t| \le \sqrt{2V_n \ln\left(\frac{2}{\delta}\right)} + \frac{2}{3} M_n \ln\left(\frac{2}{\delta}\right)$$

**Lemma B.4** (Freedman's Inequality). *Let $(Z_k, \mathcal{F}_k)_{k=0}^{t}$ be a martingale with differences $X_k = Z_k - Z_{k-1}$. Suppose that the differences are uniformly bounded, i.e., $|X_k| \le M$ almost surely for all $k$. Let $V_n = \sum_{k=1}^{t} \mathbb{E}[X_k^2 \mid \mathcal{F}_{k-1}]$ be the predictable quadratic variation of the martingale. Then, for any $t > 0$ and $\sigma^2 > 0$,*

$$\Pr\left(Z_n \ge t \text{ and } V_n \le \sigma^2\right) \le \exp\left(-\frac{t^2/2}{\sigma^2 + Mt/3}\right).$$

## B.2. Lemmas and Theorems for Uniform Convergence of RKDE

We first give the some notations and definitions that will be used in the proofs of this part.

**Notations** Here, to simplify the notation in the proofs of this part, we write the estimator of the mix density $f_{\text{mix}}$ (i.e., $\hat{f}_{\text{mix}}^{(t)}$) as $\hat{f}_t$. Following the RKDE construction in Section 4, the estimator $\hat{f}_t$ is recursively defined

$$\hat{f}_t(g_j) = \lambda_t \hat{f}_{t-1}(g_j) + (1 - \lambda_t) K_{h_t}(g_j - W_t), \quad j = 1, \ldots, B \tag{22}$$

where the decay factor $\lambda_t$ and the bandwidth $h_t$ are defined as

$$\lambda_t = 1 - \frac{1}{t^\tau}, \quad \frac{1}{2} < \tau < 1, \qquad h_t = t^{-\gamma}, \quad 0 < \gamma < \tau. \tag{23}$$

The estimator on the entire interval is constructed via linear interpolation between adjacent grid points:

$$\hat{f}_t(w) = \hat{f}_t(g_j) + \frac{\hat{f}_t(g_{j+1}) - \hat{f}_t(g_j)}{g_{j+1} - g_j}(w - g_j), \quad w \in [g_j, g_{j+1}]. \tag{24}$$

We define the weights $w_{t,i}$ based on the sequence $\{\lambda_k\}_{k\geq 1}$ as

$$w_{t,i} := \begin{cases} (1-\lambda_i)\prod_{k=i+1}^{t}\lambda_k, & 1\leq i\leq t, \\ \prod_{k=1}^{t}\lambda_k, & i=0. \end{cases} \tag{25}$$

**Lemma B.5.** *Let $p > 0$ and define*

$$S_t := \sum_{i=1}^{t}i^{-p}. \tag{26}$$

*Then the sum of the series is bounded as follows:*

$$S_t = \begin{cases} O(1), & p>1, \\ O(\log t), & p=1, \\ O(t^{1-p}), & p<1. \end{cases}$$

The implicit constants in the $O(\cdot)$ notation depend only on $p$.

*Proof.* We analyze $S_t$ using standard integral comparisons, considering three cases according to the value of $p$.

**Case 1: $p > 1$**    The function $x^{-p}$ is positive and integrable on $[1,\infty)$. Therefore,

$$S_t = 1 + \sum_{i=2}^{t}i^{-p} \leq 1 + \int_1^{\infty}x^{-p}\,dx = 1 + \frac{1}{p-1}, \tag{27}$$

which shows that $S_t = O(1)$.

**Case 2: $p = 1$**    In this case, $S_t$ is the harmonic series. Using integral comparison,

$$S_t = 1 + \sum_{i=2}^{t}\frac{1}{i} \leq 1 + \int_1^{t}\frac{1}{x}\,dx = 1 + \ln t, \tag{28}$$

and hence $S_t = O(\log t)$.

**Case 3: $0 < p < 1$**    The function $x^{-p}$ is positive and decreasing for $x > 0$, and

$$S_t \leq 1 + \int_1^{t}x^{-p}\,dx = 1 + \frac{t^{1-p}-1}{1-p} \leq \frac{t^{1-p}}{1-p}. \tag{29}$$

Thus $S_t = O(t^{1-p})$.

Combining the three cases yields the stated bound. $\square$

**Lemma B.6.** *Given the weights $\{w_{t,i}\}_{i=0}^{t}$ defined as in Equation 25 and $h_i = i^{-\gamma}$ with $\gamma \in (0,1)$, for any $\beta \in (0,1]$, there exists a constant $C > 0$ for all sufficiently large $t$,*

$$\sum_{i=1}^{t}w_{t,i}h_i^{\beta} \leq Ct^{-\gamma\beta}. \tag{30}$$

*Proof.* Define

$$S_t := \sum_{i=1}^{t}w_{t,i}h_i^{\beta}, \tag{31}$$

and split the sum at $m := \lfloor t/2 \rfloor - 2$, resulting in

$$S_t = \sum_{i=1}^{m} w_{t,i} h_i^{\beta} + \sum_{i=m+1}^{t} w_{t,i} h_i^{\beta} = S_t^{(1)} + S_t^{(2)}. \tag{32}$$

Then, we bound two terms separately.

**Step 1: Bound on $S_t^{(1)}$** For $1 \le i \le m$, using $1 - x \le e^{-x}$ for all $x \in \mathbb{R}$, we obtain

$$w_{t,i} = i^{-\tau} \prod_{k=i+1}^{t} \left(1 - k^{-\tau}\right) \le i^{-\tau} \exp\left(-\sum_{k=i+1}^{t} k^{-\tau}\right). \tag{33}$$

Moreover, since $i \le m = \lfloor t/2 \rfloor - 1$, we have

$$\sum_{k=i+1}^{t} k^{-\tau} \ge \sum_{k=\lfloor t/2 \rfloor - 1}^{t-1} k^{-\tau} \ge \int_{t/2}^{t} x^{-\tau} \, dx = C_1 t^{1-\tau}, \tag{34}$$

where $C_1 = \frac{1 - (1/2)^{1-\tau}}{1-\tau} > 0$. Substituting the above bound into the expression for $w_{t,i}$ and using $h_i^{\beta} = i^{-\gamma\beta}$, we have

$$S_t^{(1)} \le \exp(-C_1 t^{1-\tau}) \sum_{i=1}^{m} i^{-\tau+\gamma\beta}. \tag{35}$$

By Lemma B.5, no matter whether $\tau + \gamma\beta$ is greater than, equal to, or less than 1, the polynomial sum grows at most polynomially in $t$, that is, there exists constants $C_2, C_3 > 0$ such that

$$S_t^{(1)} \le C_2 \exp\left(-C_3 t^{1-\tau}\right). \tag{36}$$

**Step 2: Bound on $S_t^{(2)}$** Since $h_i$ is decreasing in $i$ and the weights satisfy $\sum_{i=1}^{t} w_{t,i} = 1$, we have

$$S_t^{(2)} \le h_{m+1}^{\beta} \sum_{i=m+1}^{t} w_{t,i} \le h_{m+1}^{\beta} = (m+1)^{-\gamma\beta} \le C_4 t^{-\gamma\beta} \tag{37}$$

for some constant $C_4 > 0$.

**Conclusion** Combining Equation 36 and the bound on $S_t^{(2)}$ and noting that the exponentially small term is negligible compared to any polynomial decay, we prove the lemma by choosing $C = C_4 + 1$ for sufficiently large $t$. □

**Lemma B.7.** *Under the same conditions as in Lemma B.6, there exists a constant $C > 0$ for all sufficiently large $t$,*

$$\sum_{i=1}^{t} w_{t,i}^2 h_i^{-1} \le C t^{\gamma-\tau}. \tag{38}$$

*Proof.* Define

$$S_t := \sum_{i=1}^{t} w_{t,i}^2 h_i^{-1}. \tag{39}$$

We split the sum at $m = \lfloor t/2 \rfloor - 2$, resulting in

$$S_t = \sum_{i=1}^{m} w_{t,i}^2 h_i^{-1} + \sum_{i=m+1}^{t} w_{t,i}^2 h_i^{-1} = S_t^{(1)} + S_t^{(2)}. \tag{40}$$

Then, we bound two terms separately.

**Step 1: Bounding $S_t^{(1)}$** Using the same argument for bounding the $S_t^{(1)}$ term in lemma B.6, we have

$$w_{t,i}^2 \leq \frac{1}{i^{2\tau}} \exp(-2C_1 t^{1-\tau}). \tag{41}$$

and hence

$$S_t^{(1)} \leq \exp(-2C_1 t^{1-\tau}) \sum_{i=1}^m i^{\gamma-2\tau} \tag{42}$$

Since the sum $\sum_{i=1}^m i^{\gamma-2\tau} \leq t^{|\gamma-2\tau|+1}$ grows at most polynomially in $t$ and is dominated by the exponentially decaying term, for sufficiently large $t$, there exist constants $C_2, C_3 > 0$ such that

$$S_t^{(1)} \leq C_2 \exp(-C_3 t^{1-\tau}). \tag{43}$$

**Step 2: Bounding $S_t^{(2)}$** For $m < i \leq t$, we bound the squared weights as

$$w_{t,i}^2 = i^{-2\tau} \left[ \prod_{k=i+1}^t \left(1 - k^{-\tau}\right) \right]^2 \leq i^{-2\tau} \exp\left(-2 \sum_{k=i+1}^t k^{-\tau}\right). \tag{44}$$

Furthermore, since $k^{-\tau}$ is a positive and decreasing function of $k$, we have

$$\sum_{k=i+1}^t k^{-\tau} \geq \int_{i+1}^{t+1} x^{-\tau} \, dx \geq (t+1)^{-\tau}(t-i) \geq 2^{-\tau} t^{-\tau}(t-i). \tag{45}$$

Substituting the above bound into the expression for $w_{t,i}^2$, we obtain

$$w_{t,i}^2 \leq i^{-2\tau} \exp\left(-2^{-\tau+1} t^{-\tau}(t-i)\right). \tag{46}$$

Multiplying the Equation 46 by $h_i^{-1} = i^\gamma$, we have

$$w_{t,i}^2 h_i^{-1} \leq i^{\gamma-2\tau} \exp\left(-2^{-\tau+1} t^{-\tau}(t-i)\right).$$

Since $m+1 \leq i \leq t$ with $m = \lfloor t/2 \rfloor - 2$, $i$ and $t$ are of the same order. Therefore, there exists a constant $C_p$ such that

$$i^{\gamma-2\tau} \leq C_p t^{\gamma-2\tau}.$$

for all sufficiently large $t$. Consequently,

$$w_{t,i}^2 h_i^{-1} \leq C_p t^{\gamma-2\tau} \exp\left(-2^{-\tau+1} t^{-\tau}(t-i)\right).$$

Summing over $i = m+1, \ldots, t$ and bounding the sum by an integral,

$$\begin{aligned} S_t^{(2)} &\leq C_p t^{\gamma-2\tau} \left(1 + \int_0^\infty e^{-2^{-\tau+1} t^{-\tau} x} \, dx\right) \\ &\leq C_p t^{\gamma-2\tau} \left(1 + C t^\tau\right) \\ &\leq C_4 t^{\gamma-\tau}. \end{aligned} \tag{47}$$

**Conclusion** Combining the bounds for $S_t^{(1)}$ and $S_t^{(2)}$, and absorbing the exponentially vanishing terms into the constant $C$, we conclude that for sufficiently large $t$,

$$S_t \leq C t^{\gamma-\tau}, \tag{48}$$

which proves the lemma.

$\square$

**Lemma B.8.** *Assume that $0 < \gamma < \tau < 1$, under the same conditions as in Lemma B.6, there exists a constant $C > 0$ such that for all $t \geq 1$,*

$$\max_{1 \leq i \leq t} w_{t,i} \, h_i^{-1} \leq C \, t^{\gamma - \tau}.$$

*Proof.* Consider the sequence

$$a_i := w_{t,i} i^{\gamma}, \qquad i = 1, \ldots, t.$$

We analyze the ratio of consecutive terms:

$$R_i := \frac{a_{i+1}}{a_i} = \left( \frac{i}{i+1} \right)^{\tau - \gamma} \frac{1}{1 - (i+1)^{-\tau}}.$$

We will show that there exists $i_0$ such that $R_i \geq 1$ for all $i \geq i_0$. This implies that the sequence $i \mapsto a_i$ is nondecreasing for $i \geq i_0$, so the maximum of $a_i$ over $1 \leq i \leq t$ is attained either at some $i \leq i_0$ or at $i = t$.

**Step 1: Lower bound $R_i$** Take logarithms:

$$\log R_i = (\tau - \gamma) \log \frac{i}{i+1} - \log \left( 1 - (i+1)^{-\tau} \right).$$

Using the inequalities $\log(1 + x) \leq x$ for $x > 0$ and $-\log(1 - x) \geq x$ for $0 < x < 1$ to bound the $\log \frac{i}{i+1}$ and $-\log \left( 1 - (i+1)^{-\tau} \right)$ terms respectively, we have

$$\log R_i \geq -(\tau - \gamma) \left( \frac{1}{i} \right) + (i+1)^{-\tau}.$$

Therefore, a sufficient condition for $R_i \geq 1$ is that the lower bound above is nonnegative, namely

$$(i+1)^{-\tau} \geq \frac{(\tau - \gamma)}{i}.$$

**Step 2: Find $i_0$ to satisfy the inequality** For $i \geq 1$ and $\tau < 1$, observe that $(i+1)^{-\tau} \geq (2i)^{-\tau} = 2^{-\tau} i^{-\tau} \geq \frac{1}{2} i^{-\tau}$. Therefore, a sufficient condition is

$$\frac{1}{2} i^{-\tau} \geq \frac{(\tau - \gamma)}{i} \quad \Longleftrightarrow \quad i^{1 - \tau} \geq 2(\tau - \gamma).$$

This motivates the choice

$$i_0 := \left\lceil \left( 2(\tau - \gamma) \right)^{\frac{1}{1 - \tau}} \right\rceil \vee 1,$$

which guarantees $R_i \geq 1$ for all $i \geq i_0$.

**Step 3: Bound the maximum** Since $i \mapsto a_i$ is nondecreasing for $i \geq i_0$, the maximum over $1 \leq i \leq t$ satisfies

$$\max_{1 \leq i \leq t} a_i \leq \max \left\{ \max_{1 \leq i \leq i_0} a_i, \, a_t \right\}.$$

For $1 \leq i \leq i_0$, use the crude bound $w_{t,i} \leq i^{-\tau}$:

$$\max_{1 \leq i \leq i_0} a_i \leq \max_{1 \leq i \leq i_0} i^{\gamma - \tau} =: C_1.$$

For $i = t$, we have the exact expression

$$a_t = w_{t,t} t^{\gamma} = t^{\gamma - \tau}.$$

Thus,

$$\max_{1 \le i \le t} a_i \le \max\{C_1,\, t^{\gamma - \tau}\}.$$

Setting $C := \max\{C_1, 1\}$ gives the desired inequality:

$$\max_{1 \le i \le t} w_{t,i} i^{\gamma} \le C\, t^{\gamma - \tau}, \quad \forall t \ge 1.$$

$\square$

**Lemma B.9** (Bias Bound of RKDE). *Let $f$ be $\beta$-Hölder continuous. Under assumption 5.2 and 5.4, for any fixed grid point $g_j \in G$, there exist constants $C > 0$ and $t_0 \in \mathbb{N}$, depending only on $c_\beta, \beta, K, \tau, \gamma$, such that for all $t \ge t_0$, the estimation $\hat{f}_t$ defined in Equation 9 satisfies*

$$\left| \mathbb{E}[\hat{f}_t(g_j)] - f(g_j) \right| \le C t^{-\gamma \beta}$$

*Proof.* Fix an arbitrary grid point $g_j \in G$, and for notation simplicity, we omit the subscript of $g_j$ in the following proof.

By unrolling the recursion, the expectation of $\hat{f}_t(g)$ can be expressed as a weighted sum:

$$\mathbb{E}\left[\hat{f}_t(g)\right] = \sum_{i=1}^{t} w_{t,i} \mathbb{E}[K_{h_i}(g - W_i)] + w_{t,0} \hat{f}_0(g) \tag{49}$$

where the weights are given by Equation 25.

Subtracting $f(g)$ and taking absolute values, we have the bias term:

$$\left| \mathbb{E}[\hat{f}_t(g)] - f(g) \right| \le \sum_{i=1}^{t} w_{t,i} \left| \mathbb{E}[K_{h_i}(g - W_i)] - f(g) \right| + w_{t,0} |\hat{f}_0(g) - f(g)|. \tag{50}$$

We first bound the bias of each individual KDE term $\mathbb{E}[K_{h_i}(g - W_i)]$ and then combine the bounds using the weighted summation. For any fixed $i$, we have

$$\begin{aligned}
\left| \mathbb{E}[K_{h_i}(g - W_i)] - f(g) \right| &= \left| \int K_{h_i}(g - x) f(x),\, dx - f(g) \right| \\
&= \left| \int K(u) f(g - h_i u),\, du - f(g) \right| \\
&\le \int |K(u)||f(g - h_i u) - f(g)| du \\
&\le c_\beta h_i^{\beta} \int |K(u)||u|^{\beta},\, du \\
&= C_1 h_i^{\beta},
\end{aligned}$$

where the second line follows from the change of variables $u = (g - x)/h_i$, and the last inequality uses the Hölder continuity of $f$ from Assumption 5.2, and the finiteness of the kernel moment $\int |K(u)||u|^{\beta} du$ from Assumption 5.4.

Next, substituting the above bound back into Equation 50, we obtain:

$$\left| \mathbb{E}[\hat{f}_t(g)] - f(g) \right| \le C_1 \sum_{i=1}^{t} w_{t,i} h_i^{\beta} + w_{t,0} |\hat{f}_0(g) - f(g)| \tag{51}$$

Bounding the first term by Lemma B.6, we obtain

$$\sum_{i=1}^{t} w_{t,i} h_i^{\beta} \le C_2 t^{-\gamma \beta} \tag{52}$$

Bounding the second term involves analyzing the decay of $w_{t,0}$. Since

$$w_{t,0} = \prod_{k=1}^{t} \left(1 - \frac{1}{k^\tau}\right) \leq \exp\left(-\sum_{k=1}^{t} \frac{1}{k^\tau}\right), \tag{53}$$

and $\sum_{k=1}^{t} k^{-\tau} \asymp t^{1-\tau}$ for $\tau \in (0,1)$, the term $w_{t,0}$ decays exponentially fast. Given that we initialize the estimator $\hat{f}_0$ as a uniform distribution over the finite grid $G$, the quantity $|\hat{f}_0(g) - f(g)|$ is bounded over all grid points $g \in G$. Consequently,

$$w_{t,0}|\hat{f}_0(g) - f(g)| = O\left(e^{-C_3 t^{1-\tau}}\right) \tag{54}$$

Combining the above bounds yields

$$\left|\mathbb{E}[\hat{f}_t(g)] - f(g)\right| \leq C t^{-\gamma\beta} \tag{55}$$

for some constant $C > 0$ and sufficiently large $t$, uniformly over all grid points $g_j$. $\qquad\square$

**Lemma B.10** (Variance Bound of RKDE). *Assume that $0 < \gamma < \tau < 1$, and suppose Assumptions 5.1, 5.2 and 5.4 hold. Then, for any fixed grid point $g_j \in G$, there exist constants $C > 0$ and $t_0 \in \mathbb{N}$, depending only on $\|f\|_\infty$, $\|K\|_2$, $\|K\|_\infty$, $\tau$, $\gamma$, such that for all $t \geq t_0$, the estimation $\hat{f}_t$ defined in Equation 22 satisfies*

$$\left|\hat{f}_t(g_j) - \mathbb{E}[\hat{f}_t(g_j)]\right| \leq C t^{\frac{\gamma-\tau}{2}} \sqrt{\log(1/\delta)},$$

*with probability at least $1 - \delta$.*

*Proof.* Like before, we fix a grid point $g_j \in G$ and suppress the subscript $j$ for notational simplicity. To bound the $|\hat{f}_t(g) - \mathbb{E}[\hat{f}_t(g)]|$ term, we first write it as a sum of independent random variables and then apply Bernstein's inequality to obtain the desired concentration bound. Specifically, using the recursive expansion, we may write

$$\hat{f}_t(g) - \mathbb{E}[\hat{f}_t(g)] = \sum_{i=1}^{t} Y_i + Y_0,$$

where

$$Y_i := w_{t,i}\left(K_{h_i}(g - W_i) - \mathbb{E}[K_{h_i}(g - W_i)]\right), \ i = 1, \ldots, t \quad \text{and} \quad Y_0 := w_{t,0}\left(\hat{f}_0(g) - \mathbb{E}[\hat{f}_0(g)]\right).$$

Since the initialization $\hat{f}_0(g)$ is deterministic, we have $Y_0 = 0$, and therefore

$$\hat{f}_t(g) - \mathbb{E}[\hat{f}_t(g)] = \sum_{i=1}^{t} Y_i. \tag{56}$$

To apply Bernstein's inequality, we need to bound the variance of each summand $Y_i$ and also obtain a uniform bound on $|Y_i|$. Next, we first derive the variance bound and then the uniform bound.

**Bound on variance** To bound the variance of $\hat{f}_t(g) - \mathbb{E}[\hat{f}_t(g)]$, we first bound the variance of each summand $Y_i$ and then sum over $i$. Since the random variables $\{Y_i\}_{i=1}^{t}$ are independent and centered, we have

$$\mathrm{Var}(Y_i) \leq \mathbb{E}[Y_i^2] = w_{t,i}^2 \, \mathbb{E}[K_{h_i}^2(g - W_i)], \quad i = 1, \ldots, t.$$

Using the scaling $K_{h_i}(w) = h_i^{-1} K(w/h_i)$ and $u = (g - W_i)/h_i$, we obtain

$$\mathbb{E}[K_{h_i}^2(g - W_i)] = \int h_i^{-2} K^2\left(\frac{g-w}{h_i}\right) f(w)dw = \int h_i^{-1} K^2(u) f(g - h_i u)\, du \leq h_i^{-1} \|f\|_\infty \|K\|_2^2, \quad i = 1, \ldots, t.$$

Multiplying the above bound by $w_{t,i}^2$ and summing over $i$, we have

$$\mathrm{Var}\left[\hat{f}_t(g) - \mathbb{E}[\hat{f}_t(g)]\right] \leq \sum_{i=1}^{t} \mathrm{Var}(Y_i) \leq \|f\|_\infty \|K\|_2^2 \sum_{i=1}^{t} w_{t,i}^2 h_i^{-1}. \tag{57}$$

Then, by Lemma B.7, we obtain

$$\text{Var}\left[\hat{f}_t(g) - \mathbb{E}[\hat{f}_t(g)]\right] \leq C_V t^{\gamma-\tau} \tag{58}$$

for some constant $C_V > 0$ depending only on $\|f\|_\infty, \|K\|_2, \tau, \gamma$.

**Uniform bound on summands**   Since $|K_{h_i}(u)| \leq h_i^{-1}\|K\|_\infty$, we have

$$|Y_i| \leq w_{t,i}\big(|K_{h_i}(g-W_i)| + \mathbb{E}|K_{h_i}(g-W_i)|\big) \leq 2w_{t,i}h_i^{-1}\|K\|_\infty.$$

Hence,

$$U := \max_{1 \leq i \leq t} |Y_i| \leq 2\|K\|_\infty \max_{1 \leq i \leq t} w_{t,i}h_i^{-1}.$$

By Lemma B.8,

$$U \leq C_U t^{\gamma-\tau}$$

for some constant $C_U > 0$ depending only on $\|K\|_\infty, \tau, \gamma$.

**Application of Bernstein's inequality**   Applying Bernstein's inequality (Lemma B.3) to $\sum_{i=1}^t Y_i$, we obtain that for any $\delta \in (0,1)$, with probability at least $1-\delta$,

$$\left|\sum_{i=1}^t Y_i\right| \leq \sqrt{2C_V t^{\gamma-\tau}\log(2/\delta)} + \frac{2}{3}C_U t^{\gamma-\tau}\log(2/\delta).$$

The second term is of lower order and can be absorbed into the first for sufficiently large $t$. Therefore, there exists a constant $C > 0$ such that

$$\left|\hat{f}_t(g) - \mathbb{E}[\hat{f}_t(g)]\right| \leq C t^{\frac{\gamma-\tau}{2}} \sqrt{\log(1/\delta)}$$

with probability at least $1-\delta$.

$\square$

**Theorem B.11** (Uniform Convergence of Recursive KDE on Fixed Grids). *Suppose the assumptions stated in Lemma B.9 and B.10 hold, then with probability at least $1-\delta$,*

$$\max_{1 \leq j \leq B} |\hat{f}_t(g_j) - f(g_j)| \leq C t^{\frac{-\tau\beta}{2\beta+1}} \sqrt{\log(B/\delta)},$$

*where $C$ is a positive constant.*

*Proof.* Combining the bias and variance bounds given by Lemma B.9 and B.10, we have for any fixed $g_j$, with probability at least $1 - \delta/B$:

$$|\hat{f}_t(g_j) - f(g_j)| \leq C_1 t^{-\gamma\beta} + C_2 t^{\frac{\gamma-\tau}{2}} \sqrt{\log(B/\delta)}.$$

By setting $\gamma = \frac{\tau}{2\beta+1} < \tau$, we balance the bias and variance terms to achieve the optimal convergence rate, resulting in

$$|\hat{f}_t(g_j) - f(g_j)| \leq C t^{\frac{-\tau\beta}{2\beta+1}} \sqrt{\log(B/\delta)}.$$

where $C$ is a constant depending on $C_1$ and $C_2$.

Finally, applying a union bound over all grid points $g_j$ for $j = 1, \ldots, B$, we have that with probability at least $1 - \delta$,

$$\max_{1 \leq j \leq B} |\hat{f}_t(g_j) - f(g_j)| \leq C t^{\frac{-\tau\beta}{2\beta+1}} \sqrt{\log(B/\delta)}.$$

$\square$

**Theorem B.12** (Uniform Convergence of RKDE on a Compact Interval). *Suppose the same assumptions as in Theorem B.11 hold, and $\hat{f}_t(x)$ is defined by the piecewise linear interpolation of $\hat{f}_t(g_j)$ on the grid $G$. Let $G = \{g_1, \ldots, g_B\}$ be a uniform grid on $[-R, R]$ with mesh size*

$$\Delta = \max_j(g_{j+1} - g_j) \asymp B^{-1}.$$

*Then for any $\delta \in (0, 1)$, with probability at least $1 - \delta$,*

$$\sup_{w \in [-R,R]} |\hat{f}_t(w) - f(w)| \leq C_1\, t^{-\frac{\tau\beta}{2\beta+1}}\sqrt{\log(B/\delta)} + C_2 B^{-\beta},$$

*where $C_1, C_2 > 0$ are constants independent of $t$.*

*Proof.* We bound the uniform error on $[-R, R]$ by combining the estimation error at grid points with the interpolation error between grid points. Specifically, fix any $w \in [-R, R]$, and let $g_j, g_{j+1} \in G$ be the adjacent grid points such that $g_j \leq w \leq g_{j+1}$. By the triangle inequality,

$$|\hat{f}_t(w) - f(w)| \leq |\hat{f}_t(w) - \hat{f}_t(g_j)| + |\hat{f}_t(g_j) - f(g_j)| + |f(g_j) - f(w)|.$$

Next, we further decompose the first term into a form similar to the second and third terms by using the linear interpolation structure of $\hat{f}_t$. By the definition of linear interpolation, we have

$$\begin{aligned}
|\hat{f}_t(w) - \hat{f}_t(g_j)| &= \frac{w - g_j}{g_{j+1} - g_j}|\hat{f}_t(g_{j+1}) - \hat{f}_t(g_j)| \\
&\leq |\hat{f}_t(g_{j+1}) - f(g_{j+1})| + |\hat{f}_t(g_j) - f(g_j)| + |f(g_{j+1}) - f(g_j)|.
\end{aligned}$$

Substituting this bound into the previous display yields

$$|\hat{f}_t(w) - f(w)| \leq 3\max_{1 \leq k \leq B}|\hat{f}_t(g_k) - f(g_k)| + 2|f(g_{j+1}) - f(g_j)|.$$

Then, we bound the two terms on the right-hand side separately.

**Interpolation error of $f$**   Since $f$ is $\beta$-Hölder continuous on $[-R, R]$ and the grid is uniform, we have

$$|f(g_{j+1}) - f(g_j)| \leq L|g_{j+1} - g_j|^\beta \leq C_L B^{-\beta}.$$

**Estimation error at grid points**   By Theorem B.11, with probability at least $1 - \delta$,

$$\max_{1 \leq k \leq B}|\hat{f}_t(g_k) - f(g_k)| \leq C_0 t^{-\frac{\tau\beta}{2\beta+1}}\sqrt{\log(B/\delta)}.$$

**Conclusion.**   Combining the above bounds, we obtain that with probability at least $1 - \delta$,

$$\sup_{w \in [-R,R]} |\hat{f}_t(w) - f(w)| \leq C_1 t^{-\frac{\tau\beta}{2\beta+1}}\sqrt{\log(B/\delta)} + C_2 B^{-\beta}.$$

where $C_1 = 3C_0$ and $C_2 = 2C_L$ are constants independent of $t$. $\qquad\square$

### B.3. Lemmas and Theorems for Consistency of Null Proportion Estimator

**Lemma B.13.** *Let $p \in [0, 1]$ denote a ground-truth quantity and $\hat{p} \in [0, 1]$ its estimator. Fix a threshold $\lambda \in (0, 1)$, and define the events*

$$A := \{p > \lambda\}, \qquad \hat{A} := \{\hat{p} > \lambda\}.$$

*Then for any $\epsilon > 0$, the symmetric difference between $A$ and $\hat{A}$ satisfies*

$$\begin{aligned}
A \triangle \hat{A} &= (A \setminus \hat{A}) \cup (\hat{A} \setminus A) \\
&\subseteq \{|p - \hat{p}| > \epsilon\} \cup \{|p - \lambda| \leq \epsilon\}.
\end{aligned} \tag{59}$$

*As a consequence,*

$$\left|\Pr(A) - \Pr(\hat{A})\right| \leq \Pr(|p - \hat{p}| > \epsilon) + \Pr(|p - \lambda| \leq \epsilon). \tag{60}$$

*Proof.* Suppose that both

$$|p - \hat{p}| \leq \epsilon \quad \text{and} \quad |p - \lambda| > \epsilon$$

hold. We show that under these conditions, $A$ and $\hat{A}$ must coincide. First, if $p > \lambda + \epsilon$, then we have

$$\hat{p} \geq p - |p - \hat{p}| > p - \epsilon > \lambda,$$

which implies $\hat{A} = A$. Second, if $p < \lambda - \epsilon$, then

$$\hat{p} \leq p + |p - \hat{p}| < p + \epsilon < \lambda,$$

which also implies $\hat{A} = A$. In both cases, the events $A$ and $\hat{A}$ agree. Therefore, a disagreement between $A$ and $\hat{A}$ can occur only if at least one of the conditions

$$|p - \hat{p}| > \epsilon \quad \text{or} \quad |p - \lambda| \leq \epsilon$$

is satisfied, yielding

$$A \triangle \hat{A} \subseteq \{|p - \hat{p}| > \epsilon\} \cup \{|p - \lambda| \leq \epsilon\}.$$

Taking probabilities on both sides and applying

$$|\Pr(A) - \Pr(\hat{A})| \leq \Pr(A \triangle \hat{A}) \tag{61}$$

complete the proof. □

**Lemma B.14** (Small-ball probability with tail control). *Under Assumptions 5.1 and 5.2, let $p_t$ be the true p-value defined as*

$$p_t := F_c(W_t),$$

*where $F_c$ is the cumulative distribution function of $W_t \mid \delta_t = 0$ with density $f_c$, and $W_t$ has marginal density $f_{mix}$.*

*Then for any $R > 0$, there exists a constant $C_R > 0$ such that for any $\lambda \in [0, 1]$ and any $\epsilon > 0$,*

$$\Pr\big(|p_t - \lambda| \leq \epsilon\big) \leq C_R \epsilon + \frac{\mathbb{E}|W_t|^K}{R^K}.$$

*Proof.* By definition, $p_t = F_c(W_t)$. We decompose the probability as

$$\Pr(|p_t - \lambda| \leq \epsilon) \leq \Pr(|p_t - \lambda| \leq \epsilon, \ |W_t| \leq R) + \Pr(|W_t| > R).$$

We first bound the localized term. On the compact set $[-R, R]$, Assumption 5.2 ensures that there exist constants $0 < l_R \leq M < \infty$ such that

$$l_R \leq f_c(w) \leq M, \qquad l_R \leq f_{mix}(w) \leq M, \qquad \forall |w| \leq R.$$

In particular, $F_c$ is strictly increasing and invertible on $[-R, R]$. Let $f_{p_t}$ denote the density of $p_t$ restricted to the event $\{|W_t| \leq R\}$. By the change-of-variables formula, for any $u$ in the image of $[-R, R]$ under $F_c$,

$$f_{p_t}(u) = \frac{f_{mix}(F_c^{-1}(u))}{f_c(F_c^{-1}(u))} \leq \frac{M}{l_R} =: C_R'.$$

Therefore, we have

$$\Pr(|p_t - \lambda| \leq \epsilon, \ |W_t| \leq R) \leq \int_{\lambda - \epsilon}^{\lambda + \epsilon} f_{p_t}(u) \, du \leq 2C_R' \epsilon.$$

For the tail term, by Markov's inequality,

$$\Pr(|W_t| > R) \leq \frac{\mathbb{E}|W_t|^K}{R^K}.$$

Combining the two parts yields

$$\Pr(|p_t - \lambda| \leq \epsilon) \leq 2C_R' \epsilon + \frac{\mathbb{E}|W_t|^K}{R^K}.$$

which completes the proof. □

**Theorem B.15** (Consistency of the null proportion estimator). *Under Assumption 5.2 and 5.6, there exist positive constants* $C_1, C_2, C_3, C_4 > 0$ *such that, with probability at least* $1 - \frac{1}{t}$,

$$\left| \pi_c - \hat{\pi}_c^{(t)} \right| \leq C_1 \sqrt{\frac{\log n_0}{n_0}} + C_2 \sqrt{\frac{\log t}{t}} + C_3 R^{-k} + C_4 \epsilon_\lambda, \tag{62}$$

*Proof.* The proof here is very similar to that of Lemma 5 in Hu et al. (2026), but we include it here for completeness. Recall that the empirical $p$-value estimator is defined as

$$\hat{p}_t = \frac{1 + \sum_{i \in [n_0]} \mathbf{1}(G(w_t) > G(w_i))}{1 + n_0}, \tag{63}$$

where $G$ is a monotone function. Define $A_t := \{p_t > \lambda\}$ and $\hat{A}_t := \{\hat{p}_t > \lambda\}$. By the law of total probability:

$$\mathbf{Pr}(A_t) = \mathbf{Pr}(p_t > \lambda \mid \delta_t = 0) \cdot \pi_c + \mathbf{Pr}(p_t > \lambda \mid \delta_t = 1) \cdot (1 - \pi_c). \tag{64}$$

For clean samples, $p_t \sim \text{Uniform}[0, 1]$, which implies

$$\mathbf{Pr}(p_t > \lambda \mid \delta_t = 0) = 1 - \lambda. \tag{65}$$

For notation simplicity, let $\epsilon'$ denote $\mathbf{Pr}(p_t > \lambda \mid \delta_t = 1)$. Therefore,

$$\begin{aligned} \mathbf{Pr}(A_t) &= \pi_c(1 - \lambda) + (1 - \pi_c) \cdot \mathbf{Pr}(p_t > \lambda \mid \delta_t = 1) \\ &= \pi_c \cdot (1 - \lambda) + (1 - \pi_c)\epsilon' \end{aligned} \tag{66}$$

Then the error of the null proportion estimator can thus be written as

$$\begin{aligned} \left| \pi_c - \hat{\pi}_c^{(t)} \right| &= \frac{1}{1 - \lambda} \left| \mathbf{Pr}(A_t) - \frac{1}{t} \sum_{t'=1}^{t} \mathbf{1}(\hat{A}_{t'}) - (1 - \pi_c)\epsilon' \right| \\ &\leq \frac{1}{1 - \lambda} \left( \underbrace{\left| \mathbf{Pr}(A_t) - \mathbf{Pr}(\hat{A}_t) \right|}_{(1)} + \underbrace{\left| \mathbf{Pr}(\hat{A}_t) - \frac{1}{t} \sum_{t'=1}^{t} \mathbf{1}(\hat{A}_{t'}) \right|}_{(2)} + \epsilon' \right). \end{aligned} \tag{67}$$

We now bound each term individually.

**Term (1):** By Lemma B.13, for any $\epsilon > 0$,

$$|\mathbf{Pr}(A_t) - \mathbf{Pr}(\hat{A}_t)| \leq \mathbf{Pr}(|\hat{p}_t - p_t| > \epsilon) + \mathbf{Pr}(|p_t - \lambda| \leq \epsilon). \tag{68}$$

For the first term on the right-hand side, conditioned on $w_t$, the $V_i = \mathbf{1}(G(w_t) > G(w_i))$ are i.i.d. Bernoulli with mean $p_t$. By Hoeffding's inequality,

$$\mathbf{Pr}(|\hat{p}_t - p_t| > \epsilon \mid w_t) \leq 2\exp(-2n_0\epsilon^2).$$

Since the bound is uniform over $w_t$, it holds unconditionally:

$$\mathbf{Pr}(|\hat{p}_t - p_t| > \epsilon) \leq 2\exp(-2n_0\epsilon^2).$$

For the second term, by Lemma B.14, there exists a constant $C > 0$ such that

$$\mathbf{Pr}(|p_t - \lambda| \leq \epsilon) \leq C \cdot \epsilon + \frac{\mathbb{E}|W|^K}{R^K}.$$

Combining the two bounds, we have

$$|\mathbf{Pr}(A_t) - \mathbf{Pr}(\hat{A}_t)| \leq 2\exp(-2n_0\epsilon^2) + C \cdot \epsilon + \frac{\mathbb{E}|W|^K}{R^K}. \tag{69}$$

**Term (2):** Define $U_t = \mathbf{1}(\hat{p}_t > \lambda)$. Conditioned on $\mathcal{D}_{\text{cal}}$, the $U_t$ are i.i.d. random variables. By applying Hoeffding's inequality:

$$\mathbf{Pr}\left(\left|\mathbb{E}[U_t \mid \mathcal{D}_{\text{cal}}] - \frac{1}{t}\sum_{t'=1}^{t} U_{t'}\right| < \epsilon \mid \mathcal{D}_{\text{cal}}\right) \geq 1 - 2\exp(-2t\epsilon^2). \tag{70}$$

Again, since the bound does not depend on $\mathcal{D}_{\text{cal}}$, it holds unconditionally. Thus, with probability at least $1 - \delta$, the second term is bounded as

$$(2) \leq \sqrt{\frac{\log\left(\frac{2}{\delta}\right)}{2t}} \tag{71}$$

**Final Bound:** Combining the two terms and using Assumption 5.6, we obtain, with probability at least $1 - \delta$,

$$|\pi_c - \hat{\pi}_c^{(t)}| \leq \frac{1}{1-\lambda}\left(2\exp(-2n_0\epsilon^2) + C\cdot\epsilon + \sqrt{\frac{\log\left(\frac{2}{\delta}\right)}{2t}} + \frac{\mathbb{E}|W|^K}{R^K} + \epsilon_\lambda\right) \tag{72}$$

Choosing

$$\epsilon = \sqrt{\frac{\log n_0}{n_0}}, \quad \delta = \frac{1}{t} \tag{73}$$

yields

$$|\pi_c - \hat{\pi}_c^{(t)}| \leq \frac{1}{1-\lambda}\left(2\cdot\frac{1}{n_0^2} + C\cdot\sqrt{\frac{\log n_0}{n_0}} + \sqrt{\frac{\log(2t)}{2t}} + \frac{\mathbb{E}|W|^K}{R^K} + (1-\pi_c)\epsilon_\lambda\right) \tag{74}$$

with probability at least $1 - \frac{1}{t}$.

For sufficiently large $n_0$ and $t$, the first term will be absorbed by the second term and the factor of $\sqrt{\log(2t)}$ can be bounded by a constant multiple of $\sqrt{\log t}$. Therefore, there exist constants $C_1, C_2, C_3, C_4 > 0$ such that, with probability at least $1 - \frac{1}{t}$,

$$|\pi_c - \hat{\pi}_c^{(t)}| \leq \frac{1}{1-\lambda}\left(C'\cdot\sqrt{\frac{\log n_0}{n_0}} + C''\sqrt{\frac{\log t}{t}} + \frac{\mathbb{E}|W|^K}{R^K} + (1-\pi_c)\epsilon_\lambda\right)$$

$$= C_1\cdot\sqrt{\frac{\log n_0}{n_0}} + C_2\cdot\sqrt{\frac{\log t}{t}} + C_3 R^{-K} + C_4\epsilon_\lambda, \tag{75}$$

where $C_1 = \frac{C'}{1-\lambda}$, $C_2 = \frac{C''}{1-\lambda}$, $C_3 = \frac{\mathbb{E}|W|^K}{1-\lambda}$, and $C_4 = \frac{1-\pi_c}{1-\lambda}$. $\qquad\square$

### B.4. Theorem of Uniform Convergence of L(w) Estimator

**Theorem B.16** (Uniform convergence of $\hat{L}$). *Under Assumption 5.1, 5.4, 5.2, 5.6 hold, there exist constants* $C_1, C_2, C_3, C_4, C_5 > 0$ *such that, with probability at least* $1 - \frac{1}{n_0} - \frac{2}{t}$,

$$\sup_{w\in[-R,R]}\left|L(w) - \hat{L}_t(w)\right| \leq C_1 n_0^{-\frac{\beta_1}{2\beta_1+1}}\sqrt{\log n_0} + C_2 t^{-\frac{\tau\beta_2}{2\beta_2+1}}\sqrt{\log Bt} + C_3 B^{-\beta_2} + C_4 R^{-K} + C_5\epsilon_\lambda. \tag{76}$$

*Proof.* Recall that

$$L(w) = \frac{f_c(w)\,\pi_c}{f_{\text{mix}}(w)}, \qquad \tilde{L}_t(w) = \frac{\hat{f}_c(w)\,\hat{\pi}_c^{(t)}}{\hat{f}_{\text{mix}}^{(t)}(w)} \qquad \hat{L}_t(w) = \Pi_{[0,1]}\left(\tilde{L}_t(w)\right).$$

where $\Pi_{[0,1]}$ denotes the projection operator onto the interval $[0,1]$. Since $L(w) \in [0,1]$, we have

$$|\hat{L}_t(w) - L(w)| = |\Pi_{[0,1]}(\tilde{L}_t(w)) - \Pi_{[0,1]}(L(w))| \leq |\tilde{L}_t(w) - L(w)|.$$

Therefore, to prove the uniform convergence of $\hat{L}_t$ to $L$, it suffices to show the uniform convergence of $\tilde{L}_t$ to $L$. Specifically, for any $w \in [-R, R]$, we have

$$\sup_{w\in[-R,R]}\left|L(w) - \tilde{L}_t(w)\right| = \sup_{w\in[-R,R]}\left|\frac{\hat{f}_c(w)\,\hat{\pi}_c^{(t)}}{\hat{f}_{\text{mix}}^{(t)}(w)} - \frac{f_c(w)\,\pi_c}{f_{\text{mix}}(w)}\right| \tag{77}$$

Rearranging terms and applying the triangle inequality yields

$$
\sup_{w\in[-R,R]} \left|L(w)-\tilde{L}_t(w)\right| \le \sup_{w\in[-R,R]} \frac{1}{f_{\text{mix}}(w)\hat{f}_{\text{mix}}^{(t)}(w)} \left(\hat{f}_c(w)f_{\text{mix}}(w)\left|\hat{\pi}_c^{(t)}-\pi_c\right|\right.
$$
$$
\left.+ f_{\text{mix}}(w)\pi_c\left|\hat{f}_c(w)-f_c(w)\right| + \pi_c\,f_c(w)\left|f_{\text{mix}}(w)-\hat{f}_{\text{mix}}^{(t)}(w)\right|\right)
$$
$$
\le \sup_{w\in[-R,R]} \left[\frac{\hat{f}_c(w)}{\hat{f}_{\text{mix}}^{(t)}(w)}\left|\hat{\pi}_c^{(t)}-\pi_c\right| + \frac{1}{\hat{f}_{\text{mix}}^{(t)}(w)}\left|\hat{f}_c(w)-f_c(w)\right| + \frac{1}{\hat{f}_{\text{mix}}^{(t)}(w)}\left|f_{\text{mix}}(w)-\hat{f}_{\text{mix}}^{(t)}(w)\right|\right]
\tag{78}
$$

By Assumption 5.2, there exists $l > 0$ such that $\inf_{w\in[-R,R]} f_{\text{mix}}(w) \ge l$. Moreover, Theorem B.12 implies that, with probability at least $1 - \frac{1}{t}$,

$$
\sup_{w\in[-R,R]} \left|\hat{f}_{\text{mix}}^{(t)}(w) - f_{\text{mix}}(w)\right| \le C_{f_{\text{mix}}}\, t^{-\frac{\tau\beta_2}{2\beta_2+1}}\sqrt{\log Bt} + C_B B^{-\beta_2}.
\tag{79}
$$

Hence, for sufficiently large $t$ and $B$,

$$
\inf_{w\in[-R,R]} \hat{f}_{\text{mix}}^{(t)}(w) \ge l - C_{f_{\text{mix}}}\, t^{-\frac{\tau\beta_2}{2\beta_2+1}}\sqrt{\log Bt} - C_B B^{-\beta_2} \ge \frac{l}{2}.
\tag{80}
$$

Similarly, we have $\sup_{w\in[-R,R]} \hat{f}_c(w) \le 3/2M$ for sufficiently large $n_0$. Combining these bounds yields

$$
\sup_{w\in[-R,R]} \left|L(w)-\tilde{L}_t(w)\right| \le \frac{2}{l}\left(\sup_{w\in[-R,R]}\left|\hat{f}_c(w)-f_c(w)\right| + \frac{3}{2}M\left|\hat{\pi}_c^{(t)}-\pi_c\right| + \sup_{w\in[-R,R]}\left|\hat{f}_{\text{mix}}^{(t)}(w)-f_{\text{mix}}(w)\right|\right).
\tag{81}
$$

Applying Lemma B.1 to $\hat{f}_c$, Theorem B.12 to $\hat{f}_{\text{mix}}^{(t)}$, and Theorem B.15 to $\hat{\pi}_c^{(t)}$, we obtain that, with probability at least $1 - \frac{1}{n_0} - \frac{2}{t}$,

$$
\sup_{w\in[-R,R]} \left|L(w)-\tilde{L}_t(w)\right|
$$
$$
\le \frac{2}{l}\left(C_{n_0}\, n_0^{-\frac{\beta_1}{2\beta_1+1}}\sqrt{\log n_0} + C_{f_{\text{mix}}}\, t^{-\frac{\tau\beta_2}{2\beta_2+1}}\sqrt{\log Bt} + C_B B^{-\beta_2} + C_{\pi_c}\sqrt{\frac{\log n_0}{n_0}} + C'_{\pi_c}\sqrt{\frac{\log t}{t}} + C''_{\pi_c}R^{-K} + C'''_{\pi_c}\epsilon_\lambda\right).
\tag{82}
$$

For sufficiently large $n_0$ and $t$, the terms $\sqrt{\frac{\log n_0}{n_0}}$ and $\sqrt{\frac{\log t}{t}}$ are dominated by $n_0^{-\frac{\beta_1}{2\beta_1+1}}\sqrt{\log n_0}$ and $t^{-\frac{\tau\beta_2}{2\beta_2+1}}\sqrt{\log t}$, respectively. Absorbing constants into $C_{n_0}$ and $C_{f_{\text{mix}}}$ completes the proof. $\qquad\square$

## B.5. Lemmas for the Main Theorem

**Lemma B.17.** *Let $0 \le a_t \le 1$ be nonnegative random variables. Define*

$$
R_T := \sum_{t=1}^{T}(1-\hat{\delta}_t), \qquad Q_T := \frac{1}{R_T}\sum_{t=1}^{T}a_t(1-\hat{\delta}_t)
$$

*with the convention $Q_T = 0$ when $R_T = 0$. Under Assumption 5.8, fix any $\eta \in (0,1)$. Then for any $T \ge t_0$,*

$$
\mathbb{E}[Q_T] \le \frac{1}{(1-\eta)cT}\sum_{t=1}^{T}\mathbb{E}[a_t] + \exp\left(-\frac{\eta^2 cT}{2(1+\eta/3)}\right).
$$

*Proof.* To control the expectation of $Q_T$, we decompose the event space into a "good event" where $R_T$ is close to its mean $\mu_T := \sum_{t=1}^{T} q_t$, and its complement. Specifically, we define the "good event"

$$
G_T := \{R_T \ge (1-\eta)\mu_T\}.
\tag{83}
$$

Then we can write

$$
\mathbb{E}[Q_T] = \mathbb{E}[Q_T\mathbf{1}_{G_T}] + \mathbb{E}[Q_T\mathbf{1}_{G_T^c}].
$$

Next, we control each term separately.

**Expectation bound on $G_T$**   For the expectation on the good event $G_T$, we have $R_T \geq (1-\eta)\mu_T$ pathwise, hence

$$Q_T \mathbf{1}_{G_T} \leq \frac{\sum_{t=1}^{T} a_t(1 - \hat{\delta}_t)}{(1-\eta)\mu_T} \mathbf{1}_{G_T}.$$

Taking expectation and using $0 \leq \hat{\delta}_t \leq 1$,

$$\mathbb{E}[Q_T \mathbf{1}_{G_T}] \leq \frac{1}{(1-\eta)\mu_T} \sum_{t=1}^{T} \mathbb{E}[a_t(1 - \hat{\delta}_t)] \leq \frac{1}{(1-\eta)\mu_T} \sum_{t=1}^{T} \mathbb{E}[a_t].$$

**Expectation bound on $G_T^c$**   To control the expectation on the complement event $G_T^c$, we first use the trivial bound $Q_T \leq 1$ to get

$$\mathbb{E}[Q_T \mathbf{1}_{G_T^c}] \leq \mathbf{Pr}(G_T^c) = \mathbf{Pr}(R_T < (1-\eta)\mu_T).$$

To bound the above probability, define

$$U_t := (1 - \hat{\delta}_t) - q_t, \qquad M_t := \sum_{t'=1}^{t} U_{t'}. \tag{84}$$

Since

$$\mathbb{E}[U_t \mid \mathcal{F}_{t-1}] = \mathbb{E}[(1 - \hat{\delta}_t) - q_t \mid \mathcal{F}_{t-1}] = q_t - q_t = 0,$$

the process $(M_t)_{t \geq 1}$ is a martingale. Moreover, since $\hat{\delta}_t \in \{0, 1\}$, $1 - \hat{\delta}_t$ is Bernoulli with mean $q_t$ conditional on $\mathcal{F}_{t-1}$. Therefore, we have

$$\mathrm{Var}(U_t \mid \mathcal{F}_{t-1}) = q_t(1 - q_t) \leq q_t.$$

Then, the predictable quadratic variation satisfies

$$V_T := \sum_{t=1}^{T} \mathrm{Var}(U_t \mid \mathcal{F}_{t-1}) \leq \sum_{t=1}^{T} q_t = \mu_T.$$

Moreover, $|U_t| \leq 1$ almost surely. Applying Freedman's inequality (one-sided form),

$$\mathbf{Pr}(M_T \leq -\eta\mu_T) \leq \exp\left(-\frac{(\eta\mu_T)^2}{2(\mu_T + \eta\mu_T/3)}\right) \tag{85}$$

Since $R_T = \mu_T + M_T$, the left-hand side equals $\mathbf{Pr}(R_T < (1-\eta)\mu_T)$.

Combining the above bounds yields

$$\mathbb{E}[Q_T] \leq \frac{1}{(1-\eta)\mu_T} \sum_{t=1}^{T} \mathbb{E}[a_t] + \exp\left(-\frac{\eta^2 \mu_T}{2(1 + \eta/3)}\right). \tag{86}$$

Using the assumption $\mu_T \geq cT$ completes the proof.

$\qquad\square$

**Lemma B.18.** *Under the same assumptions as Theorem B.16, let $L_t := L(w_t)$ and define*

$$\hat{L}_t := \hat{L}_t(w_t), \tag{87}$$

*where $\hat{L}_t(\cdot)$ denotes the estimator of $L(\cdot)$ constructed at time t. There exist constants $C_1, C_2, C_3, C_4, C_5 > 0$ such that*

$$\frac{1}{T} \sum_{t=1}^{T} \mathbb{E}[|L_t - \hat{L}_t|] \leq C_1 n_0^{-\frac{\beta_1}{2\beta_1 + 1}} \sqrt{\log n_0} + C_2 T^{-\frac{\tau\beta_2}{2\beta_2 + 1}} \sqrt{\log BT} + C_3 B^{-\beta_2} + C_4 R^{-K} + C_5 \epsilon_\lambda \tag{88}$$

*Proof.* To control the average error $\frac{1}{T} \sum_{t=1}^{T} \mathbb{E}[|L_t - \hat{L}_t|]$, we first bound the error at each iteration $t$ by decomposing it into a "good event" where the uniform convergence bound holds and its complement. Then we average over iterations to obtain the final bound.

**Bounding the expectation on each time step**   Let $t_0$ be the smallest index such that the uniform bounds in Theorem B.16 hold for all $t > t_0$. Fix any $t > t_0$. By theorem B.16, we define the "good event" $\mathcal{E}_t$ as the event where $w_t$ falls in range $[-R, R]$ and the uniform convergence bound holds at iteration $t$:

$$\mathcal{E}_t = \left\{ w_t \in [-R, R] \quad \text{and} \quad \sup_{w \in [-R,R]} |L(w) - \hat{L}_t(w)| \leq C_n n_0^{-\frac{\beta_1}{2\beta_1+1}} \sqrt{\log n_0} + C_t t^{-\frac{\tau\beta_2}{2\beta_2+1}} \sqrt{\log Bt} + C_B B^{-\beta_2} + C_R R^{-K} + C_\epsilon \epsilon_\lambda \right\}, \tag{89}$$

Therefore, the probability of the complement event satisfies

$$\mathbf{Pr}(\mathcal{E}_t^c) \leq \frac{1}{n_0} + \frac{2}{t} + \mathbf{Pr}(|w_t| > R) \leq \frac{1}{n_0} + \frac{2}{t} + \frac{\mathbb{E}|W|^K}{R^K}. \tag{90}$$

Based on this definition, we can decompose the expectation of the error at iteration $t$ as

$$\mathbb{E}[|L_t - \hat{L}_t|] = \mathbb{E}[|L_t - \hat{L}_t| \mid \mathcal{E}_t] \cdot \mathbf{Pr}(\mathcal{E}_t) + \mathbb{E}[|L_t - \hat{L}_t| \mid \mathcal{E}_t^c] \cdot \mathbf{Pr}(\mathcal{E}_t^c) \tag{91}$$

On the event $\mathcal{E}_t$, the deterministic bound holds and on the complement event we use the trivial bound $|L_t - \hat{L}_t| \leq 1$. Thus, we have

$$\mathbb{E}[|L_t - \hat{L}_t|] \leq C_n n_0^{-\frac{\beta_1}{2\beta_1+1}} \sqrt{\log n_0} + C_t t^{-\frac{\tau\beta_2}{2\beta_2+1}} \sqrt{\log Bt} + C_B B^{-\beta_2} + C_R R^{-K} + C_\epsilon \epsilon_\lambda + \frac{1}{n_0} + \frac{2}{t} + \frac{\mathbb{E}|W|^K}{R^K} \tag{92}$$

Absorbing lower-order terms into constants yields, for all sufficiently large $n_0$ and $t$,

$$\mathbb{E}[|L_t - \hat{L}_t|] \leq C_n' n_0^{-\frac{\beta_1}{2\beta_1+1}} \sqrt{\log n_0} + C_t' t^{-\frac{\tau\beta_2}{2\beta_2+1}} \sqrt{\log Bt} + C_B B^{-\beta_2} + C_R R^{-K} + C_\epsilon \epsilon_\lambda \tag{93}$$

**Average over iterations**   For $t \leq t_0$, we use the trivial bound $|L_t - \hat{L}_t| \leq 1$, thus

$$\frac{1}{T} \sum_{t=1}^{T} \mathbb{E}[|L_t - \hat{L}_t|] \leq \frac{t_0}{T} + \frac{1}{T} \sum_{t=t_0}^{T} \left( C_n' n_0^{-\frac{\beta_1}{2\beta_1+1}} \sqrt{\log n_0} + C_t' t^{-\frac{\tau\beta_2}{2\beta_2+1}} \sqrt{\log Bt} + C_B' B^{-\beta_2} + C_R R^{-K} + C_\epsilon \epsilon_\lambda \right)$$

$$\leq \frac{t_0}{T} + C_n'' n_0^{-\frac{\beta_1}{2\beta_1+1}} \sqrt{\log n_0} + \frac{C_t''}{T} \sum_{t=t_0}^{T} t^{-\frac{\tau\beta_2}{2\beta_2+1}} \sqrt{\log Bt} + C_B'' B^{-\beta_2} + C_R R^{-K} + C_\epsilon \epsilon_\lambda \tag{94}$$

To bound the third term, we have

$$\frac{1}{T} \sum_{t=t_0}^{T} t^{-\frac{\tau\beta_2}{2\beta_2+1}} \sqrt{\log Bt} \leq \frac{\sqrt{\log BT}}{T} \sum_{t=t_0}^{T} t^{-\frac{\tau\beta_2}{2\beta_2+1}} \leq C_4 T^{-\frac{\tau\beta_2}{2\beta_2+1}} \sqrt{\log BT} \tag{95}$$

where the last inequality follows from Lemma B.5.

Combining the above bounds yields

$$\frac{1}{T} \sum_{t=1}^{T} \mathbb{E}[|L_t - \hat{L}_t|] \leq \frac{t_0}{T} + C_1' n_0^{-\frac{\beta_1}{2\beta_1+1}} \sqrt{\log n_0} + C_2' T^{-\frac{\tau\beta_2}{2\beta_2+1}} \sqrt{\log BT} + C_3' B^{-\beta_2} + C_4' R^{-K} + C_5' \epsilon_\lambda \tag{96}$$

For sufficiently large $T$, the first term is negligible, completing the proof. $\qquad \square$

### B.6. The Proof of the Main Theorem

**Proposition B.19** (The equivalence of FAR definitions). *The definition of FAR in equation* (1) *is equivalent to*

$$\text{FAR}(\hat{\boldsymbol{\delta}}^T) = \mathbb{E}\left[ \frac{\sum_{t \leq T} (1 - L_t)(1 - \hat{\delta}_t)}{\sum_{t \leq T} (1 - \hat{\delta}_t)} \right]. \tag{6}$$

*Proof.* By definition,

$$\text{FAR}(\hat{\boldsymbol{\delta}}^T) = \mathbb{E}\left[\frac{\sum_{t\leq T}\delta_t(1-\hat{\delta}_t)}{\sum_{t\leq T}(1-\hat{\delta}_t)}\right] \tag{97}$$

Define the filtration as

$$\mathcal{F}_T := \sigma(\mathcal{D}_{\text{cal}}, \{w_t\}_{t\leq T}) \tag{98}$$

Using the law of total expectation, we have

$$\text{FAR}(\hat{\boldsymbol{\delta}}^T) = \mathbb{E}\left[\mathbb{E}\left[\frac{\sum_{t\leq T}\delta_t(1-\hat{\delta}_t)}{\sum_{t\leq T}(1-\hat{\delta}_t)} \mid \mathcal{F}_T\right]\right]$$

$$= \mathbb{E}\left[\frac{1}{\sum_{t\leq T}(1-\hat{\delta}_t)}\sum_{t\leq T}\mathbb{E}\left[\delta_t(1-\hat{\delta}_t) \mid \mathcal{F}_T\right]\right]$$

$$= \mathbb{E}\left[\frac{1}{\sum_{t\leq T}(1-\hat{\delta}_t)}\sum_{t\leq T}(1-\hat{\delta}_t)\mathbf{Pr}(\delta_t = 1 \mid \mathcal{F}_T)\right]$$

$$= \mathbb{E}\left[\frac{\sum_{t\leq T}(1-\hat{\delta}_t)(1-L_t)}{\sum_{t\leq T}(1-\hat{\delta}_t)}\right]$$

where the second equality follows from the linearity of expectation and the third equality follows the fact that $\hat{\delta}_t$ is measurable with respect to $\mathcal{F}_T$. $\qquad\square$

**Theorem 5.10** (FAR control). *Under the stated assumptions, the FAR of the decision sequence $\hat{\boldsymbol{\delta}}^T$ produced by Algorithm 1 satisfies*

$$\text{FAR}(\hat{\boldsymbol{\delta}}^T) \leq \alpha + \Delta_{n_0,T,B,R},$$

*where the excess term $\Delta_{n_0,T,B,R}$ is given by*

$$\Delta_{n_0,T,B,R} = C_1 n_0^{-\frac{\beta_1}{2\beta_1+1}}\sqrt{\log n_0} + C_2 T^{-\frac{\tau\beta_2}{2\beta_2+1}}\sqrt{\log(TB)} \tag{19}$$
$$+ C_3 B^{-\beta_2} + C_4 R^{-K} + C_5\epsilon_\lambda,$$

*for constants $C_1, C_2, C_3, C_4, C_5 > 0$.*

*Proof.* We write $R_T = \sum_{t\leq T}(1-\hat{\delta}_t)$ as the total number of acceptance up to time $T$. By proposition 4.1,

$$\text{FAR}(\hat{\boldsymbol{\delta}}^T) = \mathbb{E}\left[\frac{1}{R_T}\sum_{t\leq T}(1-\hat{\delta}_t)(1-L_t)\right] \tag{99}$$

Using the decomposition $1-L_t = (1-\hat{L}_t) + (\hat{L}_t - L_t)$, we have

$$\text{FAR}(\hat{\boldsymbol{\delta}}^T) = \mathbb{E}\left[\frac{1}{R_T}\sum_{t\leq T}(1-\hat{L}_t)(1-\hat{\delta}_t)\right] + \mathbb{E}\left[\frac{1}{R_T}\sum_{t\leq T}(\hat{L}_t - L_t)(1-\hat{\delta}_t)\right]$$

$$\leq \alpha + \left|\mathbb{E}\left[\frac{1}{R_T}\sum_{t\leq T}(\hat{L}_t - L_t)(1-\hat{\delta}_t)\right]\right|$$

by construction of the decision rule. It remains to bound the excess term

$$\Delta_{n_0,T,B,R} = \left|\mathbb{E}\left[\frac{1}{R_T}\sum_{t\leq T}(L_t - \hat{L}_t)(1-\hat{\delta}_t)\right]\right| \leq \mathbb{E}\left[\frac{1}{R_T}\sum_{t\leq T}|L_t - \hat{L}_t|(1-\hat{\delta}_t)\right] \tag{100}$$

Let $a_t := |L_t - \hat{L}_t|$. By Lemma B.17, which controls averages over accepted indices, we obtain for any $\eta \in (0,1)$,

$$\mathbb{E}\left[\frac{1}{R_T}\sum_{t\le T}a_t(1-\hat{\delta}_t)\right] \le \frac{1}{(1-\eta)cT}\sum_{t=1}^{T}\mathbb{E}[a_t] + \exp\left(-\frac{\eta^2 cT}{2(1+\eta/3)}\right) \quad (101)$$

By lemma B.18, for sufficiently large $T$,

$$\frac{1}{T}\sum_{t=1}^{T}\mathbb{E}[a_t] \le C_1 n_0^{-\frac{\beta_1}{2\beta_1+1}}\sqrt{\log n_0} + C_2 T^{-\frac{\tau\beta_2}{2\beta_2+1}}\sqrt{\log BT} + C_3 B^{-\beta_2} + C_4 R^{-K} + C_5\epsilon_\lambda \quad (102)$$

Combining the above bounds yields

$$\Delta_{n_0,T,B,R} \le \frac{1}{(1-\eta)c}\left(C_1 n_0^{-\frac{\beta_1}{2\beta_1+1}}\sqrt{\log n_0} + C_2 T^{-\frac{\tau\beta_2}{2\beta_2+1}}\sqrt{\log BT} + C_3 B^{-\beta_2} + C_4 R^{-K} + C_5\epsilon_\lambda\right) + \exp\left(-\frac{\eta^2 cT}{2(1+\eta/3)}\right) \quad (103)$$

For sufficiently large $T$, the exponential term is negligible compared to the polynomial rates and can be absorbed into the constants. Renaming constants completes the proof.

$\square$

## C. Additional Experiments

In this section, we present additional experimental results. First, we evaluate the temporal performance of OSCS on two additional datasets, namely AGNews and HSOL. Then, we provide a detailed analysis of the power performance of OSCS under certain settings where it may exhibit relatively low power. Finally, we empirically validate the separation assumption (Assumption 5.6).

### C.1. OSCS's Temporal Performance on Additional Datasets

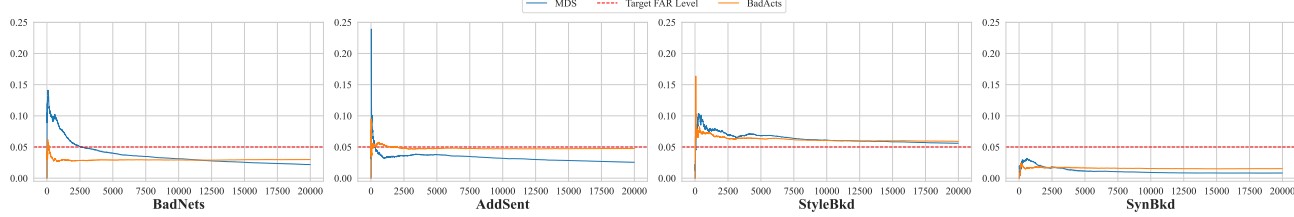

*Figure 4.* The FAR control performance of OSCS on AGNews over time. Other settings are the same as in the Figure 1.

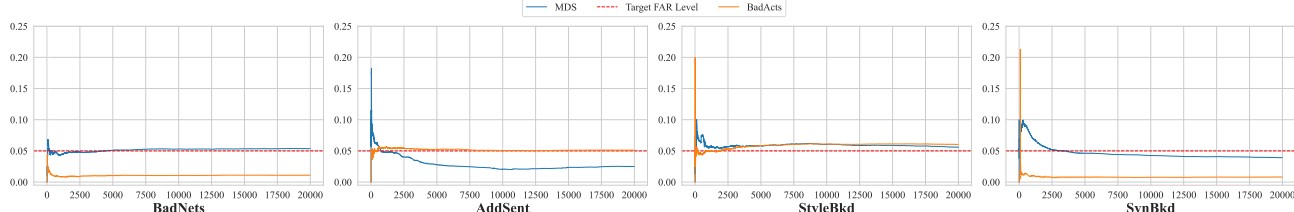

*Figure 5.* The FAR control performance of OSCS on HSOL over time. Other settings are the same as in the Figure 1.

For backdoor attacks, we further evaluate the performance of OSCS over time on two additional datasets, namely HSOL and AGNews. The results are presented in Figure 4 and Figure 5, respectively. Similar trends to those observed in the main manuscript can be seen across both datasets. Although OSCS may exhibit a relatively higher FAR during the early stages due to limited observations, it rapidly adapts to the attack patterns as more data becomes available, ultimately achieving controlled FAR in the long run. These results further demonstrate the effectiveness and robustness of OSCS in handling backdoor attacks across different datasets.

*Table 5.* Comparison between OSCS and the oracle method using BERT-base-uncased.

| | | BadNets | | AddSent | | StyleBkd | | SynBkd | |
|---|---|---|---|---|---|---|---|---|---|
| | | FAR | Power | FAR | Power | FAR | Power | FAR | Power |
| Yelp | OSCS | 0.028±0.008 | 0.891±0.017 | 0.017±0.010 | 0.658±0.136 | 0.022±0.005 | 0.781±0.025 | 0.013±0.008 | 0.837±0.058 |
| | Oracle | 0.050±0.000 | 0.914±0.001 | 0.050±0.000 | 0.841±0.002 | 0.050±0.000 | 0.854±0.002 | 0.050±0.000 | 0.949±0.001 |
| AGNews | OSCS | 0.033±0.008 | 0.720±0.034 | 0.046±0.009 | 0.780±0.026 | 0.048±0.002 | 0.749±0.005 | 0.040±0.006 | 0.537±0.021 |
| | Oracle | 0.050±0.000 | 0.841±0.003 | 0.050±0.000 | 0.864±0.002 | 0.050±0.000 | 0.797±0.007 | 0.050±0.000 | 0.731±0.004 |
| HSOL | OSCS | 0.014±0.007 | 0.744±0.086 | 0.053±0.010 | 0.777±0.040 | 0.056±0.004 | 0.452±0.007 | 0.042±0.013 | 0.598±0.052 |
| | Oracle | 0.050±0.000 | 0.889±0.004 | 0.050±0.000 | 0.846±0.004 | 0.051±0.000 | 0.459±0.005 | 0.050±0.000 | 0.631±0.004 |

*Table 6.* Comparison between OSCS and the oracle method using RoBERTa-base.

| | | BadNets | | AddSent | | StyleBkd | | SynBkd | |
|---|---|---|---|---|---|---|---|---|---|
| | | FAR | Power | FAR | Power | FAR | Power | FAR | Power |
| Yelp | OSCS | 0.024±0.006 | 0.768±0.023 | 0.033±0.011 | 0.723±0.097 | 0.018±0.008 | 0.881±0.042 | 0.027±0.002 | 0.904±0.007 |
| | Oracle | 0.050±0.000 | 0.796±0.004 | 0.050±0.000 | 0.844±0.004 | 0.050±0.000 | 0.962±0.002 | 0.050±0.000 | 0.967±0.001 |
| AGNews | OSCS | 0.022±0.011 | 0.424±0.186 | 0.026±0.014 | 0.599±0.251 | 0.056±0.016 | 0.596±0.115 | 0.008±0.002 | 0.591±0.084 |
| | Oracle | 0.050±0.000 | 0.520±0.001 | 0.050±0.000 | 0.746±0.004 | 0.051±0.000 | 0.702±0.010 | 0.050±0.000 | 0.765±0.003 |
| HSOL | OSCS | 0.053±0.009 | 0.861±0.017 | 0.025±0.016 | 0.470±0.180 | 0.056±0.032 | 0.414±0.214 | 0.039±0.003 | 0.774±0.010 |
| | Oracle | 0.050±0.000 | 0.871±0.004 | 0.050±0.000 | 0.738±0.006 | 0.050±0.000 | 0.542±0.007 | 0.050±0.000 | 0.854±0.002 |

## C.2. Power Analysis

Under certain settings, OSCS may exhibit relatively low power, such as OSCS-MD on the SynBkd in the HSOL dataset. In this section, we provide a detailed analysis of the underlying reasons for this phenomenon and discuss how more effective scoring functions can improve power while still maintaining rigorous FAR control.

Fundamentally, power and FAR are conflicting objectives, and the trade-off between them is largely determined by the discriminative capability of the scoring function. When the score distributions of benign and malicious samples substantially overlap, methods that enforce FAR control must adopt more conservative decision rules to avoid accepting ambiguous samples, which inevitably reduces power. This explains the results in Table 1 and Table 2: the MD score provides relatively weak discriminative signals, leading to substantial overlap between benign and malicious samples and consequently lower power. In contrast, BadActs leverages internal model activations that are more directly associated with malicious behavior, resulting in better separation and significantly higher power under the same FAR constraint.

To further validate this explanation, we additionally consider a strong oracle baseline. This oracle has access to the ground-truth test labels and selects the optimal score threshold offline to maximize power while satisfying the same FAR constraint. As such, it nearly represents the best achievable performance attainable for a given scoring function. The results are presented in Table 5 and Table 6. We observe that even with the oracle threshold selection, the MD score still achieves relatively limited power and only marginally outperforms our method. This suggests that the low power is inherent to the scoring function itself rather than the threshold selection procedure.

Overall, these results suggest that low power mainly arises when the scoring function fails to adequately separate benign and malicious samples. Consequently, designing more task-specific scoring functions that leverage richer and more informative signals is essential for achieving higher power while preserving strict FAR guarantees.

## C.3. Validity of the Separation Assumption

In Assumption 5.6, we assume that the score distribution of poisoned samples is sufficiently separated from that of benign samples, in the sense that poisoned samples tend to yield relatively small p-values. To empirically validate this assumption, we explicitly compute the quantity $\epsilon = \Pr(p_j > \lambda \mid \delta_j = 1)$ for all experimental settings. The results are summarized in the Table 7. We observe that $\epsilon$ remains consistently very small, and is often close to zero across all settings, indicating that the separation condition required by Assumption 5.6 is well satisfied in practice.

Intuitively, $\epsilon$ measures the proportion of poisoned samples whose scores exceed the $\lambda$-quantile of the benign calibration distribution, i.e., poisoned samples that are difficult to distinguish from benign ones. In our experiments, we set $\lambda = 0.95$, corresponding to a high quantile of the benign score distribution. Under this choice, as long as the scoring function provides

*Table 7.* The value of $\epsilon = \Pr(p_j > \lambda \mid \delta_j = 1)$ across different settings.

| | | BERT-base-uncased | | | | RoBERTa-base | | | |
|---|---|---|---|---|---|---|---|---|---|
| | | BadNets | AddSent | StyleBkd | SynBkd | BadNets | AddSent | StyleBkd | SynBkd |
| Yelp | md | 0.000±0.000 | 0.000±0.000 | 0.000±0.000 | 0.000±0.000 | 0.000±0.000 | 0.000±0.000 | 0.000±0.000 | 0.000±0.000 |
| | badacts | 0.000±0.000 | 0.000±0.000 | 0.000±0.000 | 0.000±0.000 | 0.000±0.000 | 0.000±0.000 | 0.000±0.000 | 0.000±0.000 |
| AGNews | md | 0.000±0.000 | 0.000±0.000 | 0.000±0.000 | 0.000±0.000 | 0.000±0.000 | 0.000±0.000 | 0.000±0.000 | 0.000±0.000 |
| | badacts | 0.000±0.000 | 0.000±0.000 | 0.003±0.000 | 0.000±0.000 | 0.000±0.000 | 0.000±0.000 | 0.003±0.000 | 0.000±0.000 |
| HSOL | md | 0.000±0.000 | 0.000±0.000 | 0.000±0.000 | 0.000±0.000 | 0.000±0.000 | 0.000±0.000 | 0.000±0.000 | 0.000±0.000 |
| | badacts | 0.000±0.000 | 0.000±0.000 | 0.002±0.000 | 0.000±0.000 | 0.000±0.000 | 0.000±0.000 | 0.002±0.000 | 0.000±0.000 |

reasonable discrimination—namely, most poisoned samples receive lower scores than benign samples—the resulting value of $\epsilon$ will be small enough, making the error term $C_5\epsilon_\lambda$ in Theorem 5.10 negligible.

# D. Adapt OSCS to Time-Varying Poisoning Rates

In this section, we consider a more challenging and realistic setting where the poisoning rate is not fixed but evolves over time. Such a scenario naturally arises in practical deployments, where the attack intensity may vary in a real-time manner. Next, we first formally define the problem setting, then present a methodological extension of OSCS to accommodate time-varying mixture distributions, and finally provide theoretical guarantees and empirical results demonstrating the effectiveness of the proposed method under various temporal drift patterns.

## D.1. Problem Formulation

We consider a non-stationary setting where the poison proportion $1 - \pi_c^{(t)}$ can vary over time. Note that the mixture distribution is also time-varying, given by

$$f_{\text{mix}}^{(t)} = \pi_c^{(t)} f_c + (1 - \pi_c^{(t)}) f_p, \tag{104}$$

where $f_c$ and $f_p$ are the clean and poisoned score distributions, respectively, as in Equation 3.

## D.2. Methodology

To accommodate this non-stationary setting, we extend OSCS to incorporate **exponential weighting** into both the clean proportion estimation and the density estimation procedures, thereby enabling the framework to adapt more rapidly to recent observations while gradually forgetting outdated information. Specifically, the clean proportion is estimated as

$$\hat{\pi}_c^{(t)} = \frac{\sum_{j=t-q}^{t-1} \kappa_b(j-t) \, \mathbf{1}(\hat{p}_j > \lambda)}{(1-\lambda) \sum_{j=t-q}^{t-1} \kappa_b(j-t)},$$

where $\kappa_b(\cdot)$ denotes an exponential weighting kernel. In parallel, we construct an exponentially weighted estimator for the mixture density through the recursive update

$$\hat{f}_{\text{mix}}^{(t)}(g) = \rho \hat{f}_{\text{mix}}^{(t-1)}(g) + (1 - \rho) K_h(g - W_t),$$

where $\rho = \exp(-1/b)$ and $K_h(\cdot)$ is a kernel function with bandwidth $h$.

This recursive formulation is equivalent to applying the exponential kernel $\kappa_b(s) = \exp(-|s|/b)$ over historical samples, while allowing efficient online updates without storing the entire observation history.

The effective size of this temporal window is jointly controlled by the parameters $b$ and $q$. Here, $b$ determines the decay rate of the exponential weights, larger values of $b$ produce smoother but less adaptive estimates, whereas smaller values enable faster adaptation to distributional shifts. On the other hand, the parameter $q$ specifies the maximum historical horizon considered during estimation. This design allows the estimator to focus on the most relevant recent observations while still maintaining sufficient statistical stability.

Compared with the original framework, which treats historical observations approximately uniformly, the revised formulation introduces a time-aware averaging mechanism that assigns larger weights to recent observations and progressively downweights older samples. As a result, distant historical samples have negligible influence on the current estimate.

## D.3. Thoretical Results

In this part, we present theoretical results for the FAR control performance of the proposed method under time-varying poisoning rates. Here, we first give two additional assumptions that characterize the temporal smoothness of the clean proportion and the mixture density, respectively.

**Assumption D.1** (Temporal smoothness of benign proportion). There exists a constant $\zeta > 0$ such that for all $t, s \geq 1$,

$$|\pi_c^{(t)} - \pi_c^{(s)}| \leq \zeta|t - s|.$$

**Assumption D.2** (Temporal smoothness of mixture densities). There exists a constant $\zeta_f > 0$ such that for all $t, s \geq 0$,

$$\sup_{w \in \mathbb{R}} |f_t(w) - f_s(w)| \leq \zeta_f|t - s|.$$

Then, we present a modified version of Theorem 5.10 that incorporates additional error terms to capture the impact of temporal distribution shift.

**Theorem D.3** (FAR control under temporal distribution shift). *Suppose Assumptions 5.1, 5.2, 5.4, 5.6, 5.8, D.1 and D.2 hold. Choose*

$$h = (1 - \rho)^{\frac{1}{2\beta_2 + 1}}.$$

*Then there exist constants $C_1, \ldots, C_6 > 0$ such that, for sufficiently large $T$,*

$$\mathrm{FAR}(\hat{\boldsymbol{\delta}}^T) \leq \alpha + \Delta_{\mathrm{adaptive}},$$

*where*

$$\Delta_{\mathrm{adaptive}} = C_1 \, n_0^{-\frac{\beta_1}{2\beta_1 + 1}} \sqrt{\log n_0} + C_2 \zeta_f^{\frac{\beta_2}{3\beta_2 + 1}} (\log(BT))^{\frac{2\beta_2 + 1}{2(3\beta_2 + 1)}} + C_3 B^{-\beta_2} + C_4 R^{-K} + C_5 \epsilon_\lambda + C_6 \zeta^{1/3} (\log T)^{1/3}$$

The proof can be found at Appendix D.5.

*Remark* D.4. Compared with Theorem 5.10, the excess FAR term here contains two additional components characterizing the effect of temporal distribution shift. The $\zeta_f$ term captures the impact of the temporal variation of the mixture density $f_t$, while the $\zeta$ term captures the effect of the temporal variation of the benign proportion $\pi_c^{(t)}$. In practice, $\zeta$ and $\zeta_f$ quantify the average local drift between consecutive time steps. Therefore, when the underlying distribution evolves smoothly over time, these quantities are typically small, implying that the additional excess FAR introduced by temporal distribution shift remains controlled.

*Remark* D.5. Unlike the stationary setting in Theorem 5.10, the excess FAR term in Theorem D.3 does not vanish as $T \to \infty$. This is because the adaptive estimators employ exponential forgetting and temporally localized estimation in order to track nonstationary distributions. More specifically, the effective sample size of the adaptive RKDE is of order $(1 - \rho)^{-1}$, while the effective sample size of the temporally weighted benign proportion estimator is of order $\min\{b, q\}$. Since these quantities remain finite under fixed hyperparameter choices, the estimation error does not converge to zero even as $T \to \infty$. This non-vanishing error reflects the fundamental tradeoff between adaptivity and statistical efficiency: using a shorter memory improves responsiveness to temporal distribution shift, but increases estimation variance.

## D.4. Experiments

### D.4.1. EXPERIMENTAL SETUP

To conduct a comprehensive evaluation of the proposed method under time-varying poisoning rates, we simulate various temporal drift patterns in the poisoning intensity. Specifically, at each time step $t$, the poisoning rate is given by

$$1 - \pi_c^{(t)} = 0.2 + \mathrm{drift}(t).$$

Here, we consider three representative drift patterns to capture different types of temporal dynamics:

- **Periodic linear** (amplitude = 0.2, period = 2000): triangle-wave pattern with linear increase and decrease, introducing structured but non-smooth changes. The amplitude determines the range of variation (here the poisoning rate $1 - \hat{\pi}_c^{(t)}$ varies within ±0.2 around the base level), while the period controls how long one full cycle lasts; larger periods lead to slower oscillations.

- **Sinusoidal** (amplitude = 0.2, period = 2000): smooth periodic variation, a standard non-stationary benchmark. Amplitude and period have the same meaning as Periodic linear.

- **Bounded random walk** (scale = 0.01, amplitude = 0.2): stochastic drift with reflective boundaries, capturing unpredictable but bounded changes. The scale controls the step size of random fluctuations (i.e., how rapidly the drift changes), while the amplitude limits the overall range via reflection.

Under all settings, the poisoning ratio remains within the interval $[0, 0.4]$, covering a broad spectrum of smooth, abrupt, and stochastic non-stationary scenarios.

### D.4.2. RESULTS

*Table 8.* Performance of OSCS under time-varying poisoning rates on the Yelp dataset with BERT-base-uncased.

| | BadNets | | AddSent | | StyleBkd | | SynBkd | |
|---|---|---|---|---|---|---|---|---|
| | FAR | Power | FAR | Power | FAR | Power | FAR | Power |
| Periodic Linear | 0.010±0.001 | 0.892±0.004 | 0.025±0.001 | 0.929±0.003 | 0.038±0.001 | 0.885±0.002 | 0.028±0.001 | 0.929±0.002 |
| Sinusoidal | 0.009±0.001 | 0.892±0.004 | 0.024±0.001 | 0.933±0.005 | 0.035±0.001 | 0.887±0.002 | 0.025±0.000 | 0.932±0.002 |
| Bounded Random Walk | 0.009±0.001 | 0.891±0.004 | 0.025±0.001 | 0.931±0.005 | 0.038±0.003 | 0.884±0.002 | 0.027±0.001 | 0.929±0.002 |

*Table 9.* Performance of OSCS under time-varying poisoning rates on the Yelp dataset with RoBERTa-base.

| | BadNets | | AddSent | | StyleBkd | | SynBkd | |
|---|---|---|---|---|---|---|---|---|
| | FAR | Power | FAR | Power | FAR | Power | FAR | Power |
| Periodic Linear | 0.010±0.001 | 0.887±0.004 | 0.017±0.000 | 0.900±0.005 | 0.015±0.001 | 0.916±0.004 | 0.019±0.001 | 0.970±0.002 |
| Sinusoidal | 0.009±0.001 | 0.886±0.005 | 0.015±0.001 | 0.900±0.003 | 0.014±0.001 | 0.918±0.005 | 0.018±0.001 | 0.973±0.002 |
| Bounded Random Walk | 0.011±0.001 | 0.886±0.004 | 0.016±0.001 | 0.899±0.003 | 0.015±0.002 | 0.913±0.004 | 0.019±0.000 | 0.970±0.002 |

We conduct experiments on the Yelp dataset with two models. The results in the above tables show that even under dynamically changing poisoning rates, our method continues to maintain effective FAR control, demonstrating robustness to non-stationarity and adaptive adversaries.

### D.5. Proof of Lemma and Theorems for Time-Varying Setting

In this part, we provide detailed proofs for the lemmas and theorems presented in the previous section regarding the time-varying setting. The proofs are consists of 3 parts:

1. Lemmas and proofs of Theorem D.8, which gives the error bound for the clean proportion estimator $\hat{\pi}_c^{(t)}$

2. Lemmas and proofs of Theorem D.15, which gives the error bound for the mixture density estimator $\hat{f}_{\mathrm{mix}}^{(t)}$

3. Lemmas and proofs of Theorem D.3, which gives the overall FAR control guarantee for the OSCS procedure under time-varying poisoning rates

### D.5.1. THE ERROR BOUND FOR THE CLEAN PROPORTION ESTIMATOR

Here, we use $w_j$ to denote the exponential weights $\kappa_b(j - t)$ for brevity, i.e.

$$w_j = \kappa_b(j - t) = e^{(j-t)/b}, \quad j = t - q, \dots, t - 1. \tag{105}$$

**Lemma D.6** (Effective sample size of exponential weights)**.** *Assume $b \geq 1$ and $q \geq 1$. Define the effective sample size of the exponential weights as*

$$n_{\mathrm{eff}} := \frac{\left(\sum_{j=t-q}^{t-1} w_j\right)^2}{\sum_{j=t-q}^{t-1} w_j^2}.$$

*Then there exist constants $C_1, C_2 > 0$ independent of $b$ and $q$ such that*

$$c_1 \min\{b, q\} \leq n_{\mathrm{eff}} \leq c_2 \min\{b, q\}.$$

*Proof.* We discuss two cases: $q \geq b$ and $q < b$ separately.

$q \geq b$ **case**   Let $k = t - j$, so that $k = 1, \ldots, q$. Then

$$\sum_{j=t-q}^{t-1} w_j = \sum_{k=1}^{q} e^{-k/b} = e^{-1/b} \frac{1 - e^{-q/b}}{1 - e^{-1/b}}, \quad \text{and} \quad \sum_{j=t-q}^{t-1} w_j^2 = e^{-2/b} \frac{1 - e^{-2q/b}}{1 - e^{-2/b}}.$$

Substituting into the definition of $n_{\text{eff}}$ gives

$$n_{\text{eff}} = \frac{1 - e^{-q/b}}{1 + e^{-q/b}} \cdot \frac{1 + e^{-1/b}}{1 - e^{-1/b}}.$$

Next, we derive bounds for the two factors in the above expression. For $b \geq 1$, using $\frac{x}{2} \leq 1 - e^{-x} \leq x$ for $0 < x \leq 1$ with $x = 1/b$, we obtain

$$\frac{1}{2b} \leq 1 - e^{-1/b} \leq \frac{1}{b}.$$

Since $1 \leq 1 + e^{-1/b} \leq 2$, it follows that

$$b \leq \frac{1 + e^{-1/b}}{1 - e^{-1/b}} \leq 4b.$$

Moreover,

$$0 < \frac{1 - e^{-q/b}}{1 + e^{-q/b}} \leq 1.$$

Since $q \geq b$, we have $e^{-q/b} \leq e^{-1}$, so

$$\frac{1 - e^{-q/b}}{1 + e^{-q/b}} \geq \frac{1 - e^{-1}}{1 + e^{-1}} =: c_0 > 0.$$

Combining these bounds yields

$$c_1 b \leq n_{\text{eff}} \leq c_2 b.$$

$q < b$ **case**   On the other hand, if $q < b$, then $0 \leq (t - j)/b \leq 1$, implying

$$e^{-1} \leq w_j \leq 1.$$

Hence

$$e^{-1} q \leq \sum_{j=t-q}^{t-1} w_j \leq q, \qquad e^{-2} q \leq \sum_{j=t-q}^{t-1} w_j^2 \leq q.$$

Substituting into the definition of $n_{\text{eff}}$ gives

$$e^{-2} q \leq n_{\text{eff}} \leq e^2 q.$$

Combining both regimes completes the proof.   □

**Lemma D.7** (First moment of exponentially weighted sums). *Let*

$$S_0 := \sum_{k=1}^{q} e^{-k/b}, \qquad S_1 := \sum_{k=1}^{q} k e^{-k/b},$$

*where $b \geq 1$ and $q \geq 1$. Then we have*

$$\frac{S_1}{S_0} \leq 2b.$$

*Proof.* Denote $r = e^{-1/b} \in (0,1)$, then using the standard geometric-series identity, we have

$$S_0 = \sum_{k=1}^{q} e^{-k/b} = e^{-1/b} \frac{1 - e^{-q/b}}{1 - e^{-1/b}} = r \cdot \frac{1 - r^q}{1 - r}.$$

Similarly, using the finite weighted geometric-series formula, we have

$$S_1 = \sum_{k=1}^{q} k r^k = \frac{r\left(1 - (q+1)r^q + qr^{q+1}\right)}{(1-r)^2}.$$

Therefore,

$$\frac{S_1}{S_0} = \frac{1 - (q+1)r^q + qr^{q+1}}{(1-r)(1-r^q)}.$$

Since $r \in (0,1)$, we have

$$0 \le 1 - (q+1)r^q + qr^{q+1} \le 1 - r^q,$$

Therefore,

$$\frac{S_1}{S_0} \le \frac{1}{1-r} = \frac{1}{1 - e^{-1/b}}.$$

Using the inequality

$$1 - e^{-x} \ge \frac{x}{2}, \qquad 0 < x \le 1,$$

with $x = 1/b$, we have

$$1 - e^{-1/b} \ge \frac{1}{2b},$$

and hence

$$\frac{S_1}{S_0} \le 2b.$$

This completes the proof. $\qquad\square$

**Theorem D.8** (Error bound of temporally weighted benign proportion estimator). *Suppose Assumptions 5.2, 5.6 and D.1 hold, $b \ge 1$ and $q \ge 1$. Then for any $\delta \in (0,1)$, with probability at least $1 - \delta$,*

$$\left| \hat{\pi}_c^{(t)} - \pi_c^{(t)} \right| \le C_1 \sqrt{\frac{\log n_0}{n_0}} + C_2 R^{-K} + C_3 \sqrt{\frac{\log(1/\delta)}{\min\{b, q\}}} + C_4 \zeta b + C_5 \epsilon_\lambda,$$

*where $C_1, \ldots, C_5 > 0$ are constants independent of $t$ and $b$.*

*Proof.* Introduce the intermediate quantity

$$\tilde{\pi}_c^{(t)} = \frac{\sum_{j=t-q}^{t-1} w_j \Pr(p_j > \lambda)}{(1-\lambda) \sum_{j=t-q}^{t-1} w_j}.$$

By triangle inequality,

$$\left| \hat{\pi}_c^{(t)} - \pi_c^{(t)} \right| \le \underbrace{\left| \hat{\pi}_c^{(t)} - \tilde{\pi}_c^{(t)} \right|}_{(I)} + \underbrace{\left| \tilde{\pi}_c^{(t)} - \pi_c^{(t)} \right|}_{(II)}.$$

**Step 1: Control of** $(I)$  We further decompose

$$(I) \leq (I_1) + (I_2),$$

where

$$(I_1) = \left| \frac{\sum_j w_j \left( \Pr(\hat{p}_j > \lambda) - \Pr(p_j > \lambda) \right)}{(1 - \lambda) \sum_j w_j} \right| \quad \text{and} \quad (I_2) = \left| \frac{\sum_j w_j \left( \mathbf{1}[\hat{p}_j > \lambda] - \Pr(\hat{p}_j > \lambda) \right)}{(1 - \lambda) \sum_j w_j} \right|.$$

By the same argument used to derive the Equation 69, letting $\epsilon = \sqrt{\log n_0 / n_0}$ and absorb the lower-order terms into constants, we have

$$(I_1) \leq C_1 \sqrt{\frac{\log n_0}{n_0}} + C_2 R^{-K}.$$

Next, apply the weighted Hoeffding inequality to $(I_2)$. Conditioned on the calibration dataset $\mathcal{D}_{\mathrm{cal}}$, the random variables

$$\mathbf{1}[\hat{p}_j > \lambda], \qquad j = t - q, \ldots, t - 1,$$

are independent Bernoulli random variables. Therefore, applying the weighted Hoeffding inequality conditional on $\mathcal{D}_{\mathrm{cal}}$ yields that, with probability at least $1 - \delta$,

$$(I_2) \leq C \sqrt{\frac{\log(1/\delta) \sum_j w_j^2}{(\sum_j w_j)^2}}.$$

Marginalizing over $\mathcal{D}_{\mathrm{cal}}$ gives the unconditional bound. Then, using Lemma D.6, we have

$$\frac{\sum_j w_j^2}{(\sum_j w_j)^2} = \frac{1}{n_{\mathrm{eff}}} \leq \frac{C}{\min\{b, q\}}.$$

Hence,

$$(I_2) \leq C_3 \sqrt{\frac{\log(1/\delta)}{\min\{b, q\}}}.$$

**Step 2: Control of** $(II)$  Using the definition of $\tilde{\pi}_c^{(t)}$,

$$(II) = \frac{1}{1 - \lambda} \left| \frac{\sum_j w_j \Pr(p_j > \lambda)}{\sum_j w_j} - (1 - \lambda) \pi_c^{(t)} \right|.$$

Using the decomposition

$$\Pr(p_j > \lambda) = (1 - \lambda) \pi_c^{(j)} + (1 - \pi_c^{(j)}) \Pr(p_j > \lambda | \delta_j = 1),$$

where $|(1 - \pi_c^{(j)}) \Pr(p_j > \lambda | \delta_j = 1)| \leq \epsilon_\lambda$, we obtain

$$(II) \leq \frac{\sum_j w_j |\pi_c^{(j)} - \pi_c^{(t)}|}{\sum_j w_j} + \frac{1}{1 - \lambda} \epsilon_\lambda.$$

By Assumption D.1, we have $|\pi_c^{(j)} - \pi_c^{(t)}| \leq \zeta |t - j|$. Therefore,

$$(II) \leq \zeta \frac{\sum_j w_j |t - j|}{\sum_j w_j} + \frac{1}{1 - \lambda} \epsilon_\lambda.$$

Substituting $k = t - j$ gives

$$\frac{\sum_j w_j |t - j|}{\sum_j w_j} = \frac{\sum_{k=1}^q k e^{-k/b}}{\sum_{k=1}^q e^{-k/b}}.$$

By the result of Lemma D.7, we have

$$\frac{\sum_{k=1}^{q} ke^{-k/b}}{\sum_{k=1}^{q} e^{-k/b}} \leq 2b.$$

Hence,

$$(II) \leq C_4 \zeta b + \frac{1}{1-\lambda}\epsilon_\lambda.$$

Combining the bounds for $(I)$ and $(II)$ completes the proof. $\square$

### D.5.2. THE ERROR BOUND FOR THE MIXTURE DENSITY ESTIMATOR

Like before, we use $\hat{f}_t$ to denote $\hat{f}_{\mathrm{mix}}^{(t)}$ for simplicity. Under the scenario where the true mixture density is time-varying, we denote the mixture density at time $t$ as $f_{\mathrm{mix}}^{(t)}$. Also, we define the exponential weights as

$$w_{t,i} = \begin{cases} (1-\rho)\rho^{t-i}, & 1 \leq i \leq t \\ \rho^t, & i = 0 \end{cases}$$

for $0 < \rho < 1$, then the exponentially weighted kernel density estimator can be expressed as

$$\hat{f}_t(g) = \sum_{i=1}^{t} w_{t,i} K_h(g - W_i) + w_{t,0}\hat{f}_0(g),$$

**Lemma D.9** (Weighted kernel bias bound). *There exists a constant $C > 0$ such that*

$$\sum_{i=1}^{t} w_{t,i} h^\beta \leq Ch^\beta.$$

*Proof.* Since

$$\sum_{i=1}^{t} w_{t,i} = (1-\rho)\sum_{i=1}^{t} \rho^{t-i} = 1 - \rho^t \leq 1,$$

we immediately obtain

$$\sum_{i=1}^{t} w_{t,i} h^\beta = h^\beta \sum_{i=1}^{t} w_{t,i} \leq h^\beta.$$

This completes the proof. $\square$

**Lemma D.10** (Quadratic-weight bound). *For any $h > 0$,*

$$\sum_{i=1}^{t} w_{t,i}^2 h^{-1} \leq (1-\rho)h^{-1}.$$

*Proof.* We compute

$$\sum_{i=1}^{t} w_{t,i}^2 = (1-\rho)^2 \sum_{i=1}^{t} \rho^{2(t-i)} = (1-\rho)^2 \sum_{k=0}^{t-1} \rho^{2k}.$$

Using the finite geometric-series formula,

$$\sum_{k=0}^{t-1} \rho^{2k} = \frac{1-\rho^{2t}}{1-\rho^2} \leq \frac{1}{1-\rho^2}.$$

Therefore,

$$\sum_{i=1}^{t} w_{t,i}^2 \leq \frac{(1-\rho)^2}{1-\rho^2} = \frac{1-\rho}{1+\rho} \leq 1 - \rho.$$

Multiplying both sides by $h^{-1}$ completes the proof. $\square$

**Lemma D.11** (Maximum-weight bound). *For any $h > 0$,*

$$\max_{1 \le i \le t} w_{t,i} h^{-1} \le (1 - \rho) h^{-1}.$$

*Proof.* Since $0 < \rho < 1$, the sequence

$$i \mapsto w_{t,i} = (1 - \rho) \rho^{t-i}$$

is increasing in $i$. Hence the maximum is attained at $i = t$, yielding

$$\max_{1 \le i \le t} w_{t,i} = w_{t,t} = 1 - \rho.$$

Multiplying by $h^{-1}$ completes the proof. □

**Lemma D.12** (Temporal drift bound). *Suppose Assumption D.2 holds. For any $g \in \mathbb{R}$,*

$$\sum_{i=1}^{t} w_{t,i} |f_i(g) - f_t(g)| \le \frac{\zeta_f \rho}{1 - \rho}.$$

*Proof.* By Assumption D.2,

$$|f_i(g) - f_t(g)| \le \zeta_f |t - i|.$$

Therefore,

$$\sum_{i=1}^{t} w_{t,i} |f_i(g) - f_t(g)| \le \zeta_f \sum_{i=1}^{t} w_{t,i} (t - i).$$

Substituting

$$w_{t,i} = (1 - \rho) \rho^{t-i},$$

and letting $k = t - i$, we obtain

$$\sum_{i=1}^{t} w_{t,i} (t - i) = (1 - \rho) \sum_{k=0}^{t-1} k \rho^k.$$

Using the finite weighted geometric-series formula,

$$\sum_{k=0}^{t-1} k \rho^k = \frac{\rho - t \rho^t + (t-1) \rho^{t+1}}{(1 - \rho)^2} \le \frac{\rho}{(1 - \rho)^2}.$$

The second inequality follows from the fact that $-t\rho^t + (t-1)\rho^{t+1} = \rho^t(t(\rho - 1) - \rho) \le 0$. Hence,

$$\sum_{i=1}^{t} w_{t,i} (t - i) \le \frac{\rho}{1 - \rho}.$$

Combining the above inequalities yields

$$\sum_{i=1}^{t} w_{t,i} |f_i(g) - f_t(g)| \le \frac{\zeta_f \rho}{1 - \rho}.$$

This completes the proof. □

**Lemma D.13** (Bias bound for temporally adaptive RKDE). *Suppose Assumptions 5.2, 5.4, and D.2 hold. Then for any fixed grid point $g_j \in G$,*

$$\left| \mathbb{E}[\hat{f}_t(g_j)] - f_t(g_j) \right| \le C_1 h^\beta + C_2 \frac{\zeta_f \rho}{1 - \rho} + C_3 \rho^t,$$

*where $C_1, C_2, C_3 > 0$ are constants independent of $t$.*

*Proof.* Fix a grid point $g_j \in G$, and suppress the subscript $j$ for notational simplicity.

Introduce the intermediate quantity

$$\tilde{f}_t(g) = \sum_{i=1}^{t} w_{t,i} f_i(g) + w_{t,0} f_0(g),$$

Using triangle inequality,

$$\left| \mathbb{E}[\hat{f}_t(g)] - f_t(g) \right| \leq \underbrace{\left| \mathbb{E}[\hat{f}_t(g)] - \tilde{f}_t(g) \right|}_{(I)} + \underbrace{\left| \tilde{f}_t(g) - f_t(g) \right|}_{(II)}.$$

For term $(I)$, unrolling the recursion gives

$$\mathbb{E}[\hat{f}_t(g)] = \sum_{i=1}^{t} w_{t,i} \mathbb{E}[K_h(g - W_i)] + w_{t,0} \hat{f}_0(g).$$

Hence,

$$(I) \leq \sum_{i=1}^{t} w_{t,i} \left| \mathbb{E}[K_h(g - W_i)] - f_i(g) \right| + w_{t,0} |\hat{f}_0(g) - f_0(g)|.$$

Using the same kernel-bias argument as in the proof of Lemma B.9,

$$|\mathbb{E}[K_h(g - W_i)] - f_i(g)| \leq Ch^{\beta}.$$

Therefore,

$$(I) \leq Ch^{\beta} \sum_{i=1}^{t} w_{t,i} + \rho^t |\hat{f}_0(g) - f_0(g)|.$$

Applying Lemma D.9 yields

$$(I) \leq C_1 h^{\beta} + C_3 \rho^t.$$

Next, for term $(II)$,

$$(II) = \left| \sum_{i=1}^{t} w_{t,i}(f_i(g) - f_t(g)) + w_{t,0}(f_0(g) - f_t(g)) \right|.$$

Using Assumption D.2,

$$|f_i(g) - f_t(g)| \leq \zeta_f |t - i|.$$

Hence,

$$(II) \leq \zeta_f \sum_{i=1}^{t} w_{t,i} |t - i| + \rho^t |f_0(g) - f_t(g)|.$$

Applying Lemma D.12 gives

$$(II) \leq C_2 \frac{\zeta_f \rho}{1 - \rho} + C_3 \rho^t.$$

Combining the bounds completes the proof. $\qquad\square$

**Lemma D.14** (Variance bound for temporally adaptive RKDE). *Suppose Assumptions 5.1, 5.2, and 5.4 hold. Then for any fixed grid point $g_j \in G$, with probability at least $1 - \delta$,*

$$\left| \hat{f}_t(g_j) - \mathbb{E}[\hat{f}_t(g_j)] \right| \leq C \sqrt{\frac{(1 - \rho) \log(1/\delta)}{h}},$$

*where $C > 0$ is independent of $t$.*

*Proof.* The proof follows the same decomposition as in Lemma B.10. Define

$$Y_i = w_{t,i}\Big(K_h(g_j - W_i) - \mathbb{E}[K_h(g_j - W_i)]\Big).$$

Then

$$\hat{f}_t(g_j) - \mathbb{E}[\hat{f}_t(g_j)] = \sum_{i=1}^t Y_i.$$

Using the same variance calculation as in Lemma B.10,

$$\mathrm{Var}(Y_i) \le C w_{t,i}^2 h^{-1}.$$

Therefore,

$$\sum_{i=1}^t \mathrm{Var}(Y_i) \le C \sum_{i=1}^t w_{t,i}^2 h^{-1}.$$

Applying Lemma D.10,

$$\sum_{i=1}^t w_{t,i}^2 h^{-1} \le (1-\rho) h^{-1}.$$

Moreover,

$$|Y_i| \le C w_{t,i} h^{-1}.$$

Applying Lemma D.11,

$$|Y_i| \le C(1-\rho) h^{-1}.$$

Applying Bernstein's inequality completes the proof. $\qquad\square$

**Theorem D.15** (Error bound on fixed grids for adaptive RKDE). *Suppose the assumptions of Lemma D.13 and Lemma D.14 hold. Choose the bandwidth*

$$h = (1-\rho)^{\frac{1}{2\beta+1}}.$$

*Then there exist constants $C_1, C_2 > 0$ such that for sufficiently large $t$, with probability at least $1 - \delta$,*

$$\max_{1 \le j \le B} |\hat{f}_t(g_j) - f_t(g_j)| \le C_1 (1-\rho)^{\frac{\beta}{2\beta+1}} \sqrt{\log(B/\delta)} + C_2 \frac{\zeta_f \rho}{1-\rho}.$$

*Proof.* By Lemma D.13 and Lemma D.14, for any fixed grid point $g_j$, with probability at least $1 - \delta/B$,

$$|\hat{f}_t(g_j) - f_t(g_j)| \le C_1 h^\beta + C_2 \sqrt{\frac{(1-\rho)\log(B/\delta)}{h}} + C_3 \frac{\zeta_f \rho}{1-\rho} + C_4 \rho^t.$$

Since $0 < \rho < 1$, the initialization term $\rho^t$ decays exponentially fast. Therefore, for sufficiently large $t$, the term $C_4 \rho^t$ can be absorbed into the constants.

Next, we optimize the choice of bandwidth $h$ to balance the bias and variance terms. Setting

$$h = (1-\rho)^{\frac{1}{2\beta+1}},$$

both terms have the same order and can be combined into

$$C(1-\rho)^{\frac{\beta}{2\beta+1}} \cdot \sqrt{\log(B/\delta)}.$$

Substituting this choice of $h$ into the previous bound yields

$$|\hat{f}_t(g_j) - f_t(g_j)| \le C_1 \left((1-\rho)^{\frac{\beta}{2\beta+1}} \cdot \sqrt{\log(B/\delta)}\right) + C_2 \frac{\zeta_f \rho}{1-\rho}.$$

Finally, applying the union bound over all grid points completes the proof. $\qquad\square$

**Theorem D.16** (Error bound on a compact interval for adaptive RKDE). *Suppose the assumptions of Lemma D.13 and Lemma D.14 hold, and let $\hat{f}_t(w)$ be defined by the piecewise linear interpolation of $\hat{f}_t(g_j)$ on the grid $G$.*

*Let $G = \{g_1, \ldots, g_B\}$ be a uniform grid on $[-R, R]$ with mesh size*

$$\Delta = \max_j (g_{j+1} - g_j) \asymp B^{-1}.$$

*Choose the bandwidth*

$$h = (1 - \rho)^{\frac{1}{2\beta+1}}.$$

*Then there exist constants $C_1, C_2, C_3 > 0$ such that for sufficiently large $t$, with probability at least $1 - \delta$,*

$$\sup_{w \in [-R,R]} |\hat{f}_t(w) - f_t(w)| \leq C_1 (1-\rho)^{\frac{\beta}{2\beta+1}} \sqrt{\log(B/\delta)} + C_2 \frac{\zeta_f \rho}{1 - \rho} + C_3 B^{-\beta}.$$

*Proof.* The proof follows the same interpolation argument as in Theorem B.12.

Fix any $w \in [-R, R]$, and let $g_j, g_{j+1} \in G$ be the adjacent grid points such that $g_j \leq w \leq g_{j+1}$. By the same decomposition as in Theorem B.12, we have

$$|\hat{f}_t(w) - f_t(w)| \leq 3 \max_{1 \leq k \leq B} |\hat{f}_t(g_k) - f_t(g_k)| + 2|f_t(g_{j+1}) - f_t(g_j)|.$$

Since $f_t$ is uniformly $\beta$-Hölder continuous on $[-R, R]$,

$$|f_t(g_{j+1}) - f_t(g_j)| \leq C|g_{j+1} - g_j|^\beta \leq CB^{-\beta}.$$

Moreover, by Theorem D.15, with probability at least $1 - \delta$,

$$\max_{1 \leq k \leq B} |\hat{f}_t(g_k) - f_t(g_k)| \leq C_1 (1-\rho)^{\frac{\beta}{2\beta+1}} \sqrt{\log(B/\delta)} + C_2 \frac{\zeta_f \rho}{1 - \rho}.$$

Substituting the above bounds into the previous display yields

$$\sup_{w \in [-R,R]} |\hat{f}_t(w) - f_t(w)| \leq C_1 (1-\rho)^{\frac{\beta}{2\beta+1}} \sqrt{\log(B/\delta)} + C_2 \frac{\zeta_f \rho}{1 - \rho} + C_3 B^{-\beta}.$$

Absorbing constants completes the proof. $\qquad\square$

### D.5.3. OVERALL FAR CONTROL GUARANTEE

**Theorem D.3** (FAR control under temporal distribution shift). *Suppose Assumptions 5.1, 5.2, 5.4, 5.6, 5.8, D.1 and D.2 hold. Choose*

$$h = (1 - \rho)^{\frac{1}{2\beta_2+1}}.$$

*Then there exist constants $C_1, \ldots, C_6 > 0$ such that, for sufficiently large $T$,*

$$\mathrm{FAR}(\hat{\boldsymbol{\delta}}^T) \leq \alpha + \Delta_{\text{adaptive}},$$

*where*

$$\Delta_{\text{adaptive}} = C_1 n_0^{-\frac{\beta_1}{2\beta_1+1}} \sqrt{\log n_0} + C_2 \zeta_f^{\frac{\beta_2}{3\beta_2+1}} (\log(BT))^{\frac{2\beta_2+1}{2(3\beta_2+1)}} + C_3 B^{-\beta_2} + C_4 R^{-K} + C_5 \epsilon_\lambda + C_6 \zeta^{1/3} (\log T)^{1/3}$$

*Proof.* The proof follows the same decomposition as in Theorem 5.10. By the same argument as in the proof of Theorem B.16, we have, with probability at least $1 - \frac{1}{n_0} - 2\delta$,

$$\sup_{w \in [-R,R]} \left| L_t(w) - \hat{L}_t(w) \right| \leq \frac{2}{l} \left( \sup_{w \in [-R,R]} \left| \hat{f}_c(w) - f_c(w) \right| + \frac{3}{2} M \left| \hat{\pi}_c^{(t)} - \pi_c^{(t)} \right| + \sup_{w \in [-R,R]} \left| \hat{f}_t(w) - f_t(w) \right| \right). \quad (106)$$

Applying Lemma B.1 to $\hat{f}_c$, Theorem D.16 to $\hat{f}_t$, and Theorem D.8 to $\hat{\pi}_c^{(t)}$, we obtain that, with probability at least $1 - \frac{1}{n_0} - 2\delta$,

$$\sup_{w \in [-R,R]} \left| L_t(w) - \hat{L}_t(w) \right| \leq C_{n_0} \, n_0^{-\frac{\beta_1}{2\beta_1+1}} \sqrt{\log n_0} + C_{f_{\mathrm{mix}}} (1-\rho)^{\frac{\beta_2}{2\beta_2+1}} \sqrt{\log(B/\delta)} + C_\zeta \frac{\zeta_f \rho}{1-\rho}$$
$$+ C_B B^{-\beta_2} + C_{\pi_c} \sqrt{\frac{\log n_0}{n_0}} + C'_{\pi_c} \sqrt{\frac{\log(1/\delta)}{\min\{b,q\}}} + C''_{\pi_c} R^{-K} + C'''_{\pi_c} \epsilon_\lambda + C'_\zeta \zeta b \quad (107)$$

Next, similar to the argument we obtain Equation 96 in Lemma B.18, we have

$$\frac{1}{T} \sum_{t=1}^{T} \mathbb{E}[|L_t - \hat{L}_t|] \leq C_{n_0} \, n_0^{-\frac{\beta_1}{2\beta_1+1}} \sqrt{\log n_0} + C_{f_{\mathrm{mix}}} (1-\rho)^{\frac{\beta_2}{2\beta_2+1}} \sqrt{\log(B/\delta)} + C_{\zeta_f} \frac{\zeta_f \rho}{1-\rho}$$
$$+ C_B B^{-\beta_2} + C_{\pi_c} \sqrt{\frac{\log n_0}{n_0}} + C'_{\pi_c} \sqrt{\frac{\log(1/\delta)}{\min\{b,q\}}} + C''_{\pi_c} R^{-K} + C'''_{\pi_c} \epsilon_\lambda + C_\zeta \zeta b \quad (108)$$
$$+ \frac{1}{n_0} + 2\delta$$

For sufficiently large $n_0$, the terms $\sqrt{\frac{\log n_0}{n_0}}$ and $\frac{1}{n_0}$ are dominated by $n_0^{-\frac{\beta_1}{2\beta_1+1}} \sqrt{\log n_0}$. Therefore, we can absorb the terms $\sqrt{\frac{\log n_0}{n_0}}$ into the first term by adjusting the constant $C_{n_0}$.

Then, we optimize the parameters to balance the terms. Setting

$$\delta = \frac{1}{T}, \quad b = q = \left( \frac{\log T}{\zeta^2} \right)^{1/3}, \quad \text{and} \quad (1-\rho) = \left( \frac{\zeta_f}{\sqrt{\log(BT)}} \right)^{\frac{2\beta_2+1}{3\beta_2+1}}, \quad (109)$$

and absorbing constant-order factors into the constants (note that $\rho \in (0,1)$, so the multiplicative factor $\rho$ in $\frac{\zeta_f \rho}{1-\rho}$ can be absorbed into the constant), we obtain

$$\frac{1}{T} \sum_{t=1}^{T} \mathbb{E}[|L_t - \hat{L}_t|] \leq C_{n_0} \, n_0^{-\frac{\beta_1}{2\beta_1+1}} \sqrt{\log n_0} + C'_{\zeta_f} \zeta_f^{\frac{\beta_2}{3\beta_2+1}} \left( \log(BT) \right)^{\frac{2\beta_2+1}{2(3\beta_2+1)}}$$
$$+ C_B B^{-\beta_2} + C''_{\pi_c} R^{-K} + C'''_{\pi_c} \epsilon_\lambda + C'_\zeta \zeta^{1/3} \left( \log T \right)^{1/3} \quad (110)$$

Finally, following the same argument as in the proof of Theorem 5.10, we complete the proof.

$\square$

