# OpenReview forum: "OSCS: Online Selection with Provable FAR Control for LLM Safety"
_ICML.cc/2026/Conference — ICML 2026 regular_

### Official Review · Reviewer_mfpr · 2026-03-10

**Soundness:** 2
**Presentation:** 3
**Significance:** 3
**Originality:** 3
**Overall Recommendation:** 4
**Confidence:** 3

**Summary:**

The authors introduce OSCS (Online Selection with Controlled Safety. OSCS is a wrapper overlaid on top of scoring functions. The authors focus on OSCS’s applications in the context of detecting harmful inputs (in particular, jailbreaks and backdoor attacks), and its applications in high-risk corporate settings, especially where sensitive information (e.g. PII from healthcare/finance/education services) might be elicitable by malicious users via prompt injections/jailbreaks/etc.
The authors claim that OSCS has three especially compelling features. Firstly, that OSCS allows the developers of LLM products to control the false-acceptance rate of the underlying classifiers which they wrap OSCS around. Controllability, the authors claim, is especially important in high-stakes settings like those listed above. The second advantage of OSCS over other baselines, the authors claim, is that OSCS only requires benign calibration samples, and hence doesn’t require an extensive dataset of existing malicious examples. Finally, since OSCS is fed an online test stream, it’s claimed that the method is able to update to flag novel malicious inputs. This is advantageous in the context of the ever-updating attack environment of language model redteaming, where static defenses are often broken because they fail to generalise to some unseen, novel attack.

**Compliance With Llm Reviewing Policy:**

Affirmed.

**Key Questions For Authors:**

1. Why are the baselines limited to MDS, BadActs, and PPL? Have the authors considered how OSCS would integrate with or compare against more recent classifier-based detectors, and with different misuse scenarios, e.g. biorisk?
2. Can the authors say more about the power tradeoff — in particular, is there a principled way for practitioners to understand when OSCS-md will substantially underperform on power, and how to choose between scoring functions?
3. Is there a reason that the authors choose older attack methods for their jailbreak and backdoor tasks?

**Limitations:**

Yes

**Strengths And Weaknesses:**

On strengths:

* The core motivation here seems compelling to me. I hadn’t thought too much about the problem of different settings wanting more reliable, granular control over the accepted false-accepctance rate, and to my understanding a lot of the jailbreaking literature doesn’t address the concern that different contexts might accept different points on the pareto-curve of reliability vs the rate of flagged false acceptances.
* OSCS seems adaptable across different classifier / detector setups, which is a point in favour of how widely it could be potentially useful.
* The verifiability of the false positive rate, while hard for me to evaluate since such formal proofs aren’t an area of expertise for me, seems roughly sound - and if sound, obviously useful in high stakes deployments as a guarantee of the upper bound.
And, the proofs seem to be broadly borne out in the empirical results, with minor fluctuations.
* The experimental scale (number of runs, degree of attack diversity) is broadly in line with the field, if a little thin compared to more empirically-focused work in this area. I chalk that up to the fact that the authors want to focus on the theoretical verifiability, and I don’t think it holds the paper back much - but again, this isn’t quite my area of expertise.

For weaknesses:
While I can follow the proof architecture, and the assumptions seem reasonable and clearly stated, the detailed RKDE convergence arguments (Appendix E.2-E.3) are outside my wheelhouse — I encourage AC(s) to weight a more theoretically expert reviewer's opinion on that component, and so my confidence score for this paper is lower.

* The authors identify some reasonable domains for high-risk misuse, but I think they under-explore the utility of this work for wider safety applications in their framing. However, they implicitly address this concern through their empirical results. The benchmark they use for sample misuse prompts, AdvBench, contains harmful queries around the creation of bioweapons, and frontier model developers seem increasingly concerned about model capabilities in this domain. (OAI, Anthropic, GDM) have recently flagged major increases in model capability (e.g. when Anthropic released Opus 4 last May, the model triggered their internal threshold for activating ASL-3 protections against Biorisk - https://www-cdn.anthropic.com/807c59454757214bfd37592d6e048079cd7a7728.pdf).
* Especially with the biorisk framing in mind, it seems to me that some contemporary work on the detection, classification, and filtering of harmful inputs/jailbreaks is absent. Most notably, I’d have expected to see more discussion of Constitutional Classifiers (Sharma et al., Feb 2025), released February 2025. The authors should at least have discussed it, even if reproducing it externally is impractical
* The examples of working jailbreaks (autoDAN-ga and -hga) and backdoor attacks (data poisoning (Gu et al., 2019), weight manipulation (Yang et al., 2021c), and model editing (Li et al., 2023)) seem a little out of date, each having been proposed three or more years ago. More engagement with more recent literature on these attacks, or reasonable justification for why more recent jailbreaks weren’t considered would be helpful.
* OSCS-md shows substantially lower power than baselines in some settings (e.g. 0.424 and 0.596 on AGNews in Table 1), and this is discussed only briefly in the paper.

---

### Official Review · Reviewer_WMXx · 2026-03-12

**Soundness:** 3
**Presentation:** 2
**Significance:** 3
**Originality:** 3
**Overall Recommendation:** 3
**Confidence:** 2

**Summary:**

The paper proposes a real-time detection framework, called OSCS, that controls the false acceptance rate (FAR) for the secure deployment of LLMs in serious risk environments.
The paper challenges existing methods for their inability to maintain robust security metrics under practical constraints, such as data sparsity and continuous streaming data.
The authors introduce a recursive density estimation method to determine whether to attack in real time from the incoming data stream without requiring separate malicious training data.
The authors demonstrated that the framework strictly adheres to established safety standards in the backdoor attack and jailbreak scenario through theoretical proofs and extensive experiments.

**Compliance With Llm Reviewing Policy:**

Affirmed.

**Final Justification:**

The authors provided convincing responses to Q1–Q3, particularly the exponential weighting extension with theoretical guarantees and non-stationary experiments, which substantially addresses my original concerns regarding adaptivity and distribution shift. However, the core weaknesses remain insufficiently resolved (W1: calibration assumption & W2: outdated jailbreak attack)

**Key Questions For Authors:**

- The proposed method appears to assume that the distribution of the initial calibration dataset remains fixed. However, in practice, the distribution of benign data can also drift over time. Is there any strategy to address such a benign distribution shift?

- In the paper, it appears that as t increases, lamda_t converges to 1. If this is the case, the method may eventually place little or no weight on newly observed data. Would this limit the model’s ability to adapt to future data?

- The use of a simple cumulative average may make it difficult to handle shifts in the attack ratio. For example, if benign data are observed up to the 10,000th sample and a malicious sample appears at the 10,001st step, it is unclear whether the method can detect such a sudden change.

**Limitations:**

yes

**Strengths And Weaknesses:**

### Strength
- The proposed approach controls the false acceptance rate (FAR) under a user-defined limit without requiring known malicious samples, considering the constantly evolving nature of attackers. The authors use the recursive kernel density estimation (RKDE) methods, which could efficiently estimate the entire score distribution with only O(B) memory and time complexity
- The proposed method is not restricted to a specific attack type and can be universally applied to both backdoor attacks and jailbreak attacks, two major security threats to LLMs. They prove the proposed method both mathematically and experimentally.
### Weakness
- The assumption that the calibration dataset consists of 100% pure benign data appears somewhat unrealistic.
- In the experimental setup, AutoDAN appears to be somewhat outdated. To the best of my knowledge, Llama3.1-8B is known to exhibit strong robustness against AutoDAN attack. Therefore, a more detailed description of the experimental data and setup would be necessary to properly interpret the reported results.
- There are a few minor typographical errors in the paper. (Algorithm 1, the the, calculate is, ....)

---

> ### Author Rebuttal · Authors · 2026-03-31
>
> We thank the reviewer for the valuable feedback.
>
> **W1 (Clean calibration assumption).**
> The assumption of a clean calibration set is standard in many practical settings and is often more realistic than alternatives requiring labeled malicious data. In practice, calibration data can be constructed from trusted sources (e.g., curated datasets or filtered logs), where contamination is negligible or well controlled.
> We acknowledge that handling adversarially contaminated calibration data is important but significantly more challenging, and we leave this as future work.
>
> ---
>
> **W2 (AutoDAN outdated).**
> Our contribution is orthogonal to specific attack methods. The goal of our framework is to provide **statistical guarantees (FAR control)** on top of any scoring/filtering mechanism. If the underlying model is robust to certain attacks, this further improves safety, but such robustness alone typically lacks explicit statistical guarantees. Our method complements these defenses by adding a principled control layer.
>
> Moreover, we evaluate both jailbreak (AutoDAN) and backdoor attacks, and consistently observe effective FAR control across settings, indicating that our framework is not tied to a specific attack type.
>
> ---
>
> **W3 (Typos).**
> We apologize for them and will correct them carefully.
>
> ---
>
> **Q1 (Benign distribution shift).**
> Our framework assumes only benign calibration data, which is a practical but challenging setting. Under this setup, a relatively stable benign distribution is important to distinguish malicious signals and ensure FAR control.
>
> If both benign and malicious distributions drift simultaneously, it becomes fundamentally difficult to disentangle their effects without labeled data. This limitation is inherent to the problem rather than specific to our method.
> In practice, the assumption is reasonable since benign data often reflects stable user behavior or trusted sources.
>
> ---
>
> **Q2 ($\lambda_t \to 1$ and adaptivity).**
> Under the original stationary setting, $\lambda_t \to 1$ is desirable, as it stabilizes the estimator and ensures convergence.
>
> However, we agree that under a distribution shift this may limit adaptivity. To address this, we extend our framework to **non-stationary mixture distributions** using exponential weighting.
>
> We estimate the clean proportion as
> $$
> \hat{\pi}_c^{(t)} = 1 - \frac{\sum_{j=t-q}^{t-1}\kappa_b(j-t)\mathbb{I}(\hat{p}_j > \lambda)}{(1-\lambda)\sum_{j=t-q}^{t-1}\kappa_b(j-t)},
> $$
>
> and update the density via
>
> $$
> \hat{f}_{\mathrm{mix}}^{(t)} =
> \lambda \hat{f}_{\mathrm{mix}}^{(t-1)} + (1-\lambda) K_h(\cdot - W_t),
> $$
> where $\lambda = \exp(-1/b)$.
>
> This is equivalent to exponential weighting $\kappa_b(s)=\exp(-|s|/b)$, where $b$ controls the effective memory. Larger $b$ gives smoother estimates, while smaller $b$ enables faster adaptation.
>
> Intuitively, this replaces global averaging with a **localized temporal window**: recent samples receive higher weights, while older ones are exponentially down-weighted. Thus, the estimator effectively operates on recent data, enabling adaptation to distribution shifts.
>
> ---
>
> **Theoretical results.**
> Assume smooth non-stationarity:
> $$
> |\pi_c^{(t+1)}-\pi_c^{(t)}|\le\zeta,\quad \|f_{t+1}-f_t\|_\infty\le\zeta.
> $$
>
> We obtain
> $$
> |\hat f_t - f_t|
> \le C_1 h^\beta
> + C_2 \sqrt{\frac{\log(1/\delta)}{b h}}
> + C_3 \sum_{k=0}^{t-1}\lambda^k \|f_{t-k}-f_{t-k-1}\|.
> $$
>
> This decomposes into bias, variance, and drift. Optimizing $h,b$ yields
> $$
> |\hat f_t - f_t|
> \lesssim
> \zeta^{\frac{\beta}{2\beta+2}}(\log(1/\zeta))^{\frac{\beta}{2\beta+2}},
> $$
> which vanishes as $\zeta\to 0$, implying robustness under gradual drift.
>
> Similarly,
> $$
> |\hat{\pi}_c^{(t)} - \pi_c^{(t)}|
> \le
> C_1\sqrt{\frac{\log n_0}{n_0}}
> + C_2 R^{-1}
> + C_3 (\log(1/\delta)\zeta)^{1/3}
> + C_4 \epsilon_\lambda,
> $$
> showing that non-stationarity introduces a controlled additional error.
>
> ---
>
> **Experiments (non-stationary setting).**
> We evaluate time-varying poisoning rates
> $$
> 1-\pi_c^{(t)} = 0.2 + \text{drift}(t),
> $$
> under three patterns: periodic linear, sinusoidal, and bounded random walk (range $[0,0.4]$).
>
> *Representative results (Bert-base):*
>
> | Drift | FAR | Power |
> |------|-----|-------|
> | Periodic | 0.010 | 0.892 |
> | Sinusoidal | 0.009 | 0.892 |
> | Random walk | 0.009 | 0.891 |
>
> Similar results hold for RoBERTa. FAR remains well controlled across all scenarios, demonstrating robustness to non-stationarity.
>
> ---
>
> **Q3 (Sudden change).**
> In the revised framework, exponential weighting enables adaptation to time-varying attack ratios.
>
> We emphasize that in online learning, guarantees are inherently **sequence-level**, not per-step. A single abrupt change cannot be reliably distinguished from noise without sufficient observations.
>
> Our method addresses this via controlled adaptivity: after a change, it quickly shifts focus to recent samples (within an effective window of size $\sim b$), achieving a principled balance between stability and responsiveness.

---

> > ### Author Rebuttal · Reviewer_WMXx · 2026-04-04
> >
> > Thank you for the detailed rebuttal. After carefully reviewing your responses, I have decided to update my score from reject to weak reject.
> > - (W1)
> > I appreciate the clarification that clean calibration data can be constructed from trusted sources. However, the argument remains somewhat optimistic in truly adversarial high-stakes settings, where even trusted sources may be compromised. I acknowledge this as a known limitation and appreciate that you have flagged it as future work.
> > - (W2)
> > While I agree that the framework's contribution is orthogonal to specific attack methods, the experimental validation would be significantly strengthened by including a broader range of attacks. Specifically, I would recommend evaluating against: gradient-based attack (GCG) and black-box attack (PAIR, TAP, ...)

---

### Official Review · Reviewer_BPJL · 2026-03-13

**Soundness:** 2
**Presentation:** 3
**Significance:** 2
**Originality:** 3
**Overall Recommendation:** 4
**Confidence:** 3

**Summary:**

This paper proposes OSCS, a framework for online malicious input detection for LLMs that provides provable control over the false acceptance rate (FAR) without requiring labeled malicious calibration data. The method takes detection scores from existing defenses, estimates benign likelihoods via recursive kernel density estimation on the streaming test data, and uses an lFDR-based selection rule to make real-time accept/reject decisions. The authors prove that FAR converges to a user-specified level $\alpha$ up to vanishing excess terms, and evaluate on backdoor attacks (four attack types, two models, three datasets) and jailbreak detection (two attacks, three LLMs).

**Compliance With Llm Reviewing Policy:**

Affirmed.

**Key Questions For Authors:**

* Can you compute and report the empirical excess $\Delta$ for your experimental settings? This would substantially strengthen the claim of "provable" control.
* What properties of a scoring function predict whether OSCS will achieve acceptable power? Can you formalize or empirically characterize this?

**Limitations:**

yes

**Strengths And Weaknesses:**

Strengths:

* The problem formulation — online streaming + no malicious calibration data + explicit FAR control — is novel and well-motivated. No prior work addresses all three constraints simultaneously.
* Theorem 5.10 gives a clean, interpretable decomposition of excess FAR into four terms. The theory is technically sound under the stated assumptions.
* Backdoor experiments are thorough (four attacks × three datasets × two models × two scoring functions) and show consistent FAR control where baselines are erratic.

Weaknesses:

* The FAR guarantee is in expectation and asymptotic, which limits its practical safety value. For any finite deployment, the guarantee is FAR ≤ $\alpha$ + $\Delta$ with $\Delta > 0$ . The paper never quantifies $\Delta$ empirically for any setting. Without this, a practitioner cannot know how far the actual guarantee is from $\alpha$. For a paper whose central claim is "provable FAR control," this gap between theory and practice needs to be close
* Power degradation is substantial but underanalyzed. OSCS-md shows power as low as 0.42 in some settings, while OSCS-badacts is much better. This reveals heavy dependence on the scoring function, but the paper provides no systematic analysis of when or why power collapses
* Assumption 5.6 (score separation) is critical but never empirically verified. The ε_λ parameter directly enters the bound yet is not measured for any experiment. This makes it impossible to assess whether the theoretical guarantees are meaningful in the tested settings

---

> ### Author Rebuttal · Authors · 2026-03-31
>
> We thank the reviewer for raising these important points.
>
> **W1 (FAR guarantee).**
> Providing guarantees in expectation and asymptotically is standard in statistical and online learning, especially in streaming settings. The term $\Delta$ is a **slack upper bound**, not a decomposition of realized FAR. Our result ensures
> $$
> \mathrm{FAR} \le \alpha + \Delta,
> $$
> But the realized FAR is typically much smaller. Although $\Delta$ is hard to compute explicitly due to unknown constants, its practical impact can be assessed empirically. Across all experiments (Table 1, Table 2, Figure 1, Figure 3), the observed FAR consistently remains close to $\alpha$. Moreover, $\Delta$ decreases with calibration size $n_0$ and time horizon $T$. In our setup (e.g., $n_0\approx1200, T\approx2000$), FAR rarely exceeds $\alpha$ by a noticeable margin, indicating that $\Delta$ is small in practice.
>
> ---
>
> **W2 (Power).**
> We analyze when and why power becomes low. Power and FAR are inherently conflicting, and their trade-off is governed by the discriminative capacity of the scoring function. When the score distributions of benign and malicious samples overlap significantly, enforcing FAR requires avoiding the overlapping region, leading to conservative selection and rejection of many benign samples, thus reducing power.
>
> This explains our observations: MD exhibits lower power due to weaker discrimination, while BadActs achieves higher power by leveraging stronger signals.
>
> To further verify that this limitation is not caused by our framework, we introduce a strong **oracle baseline** that has access to labels and selects optimal thresholds to maximize power under the same FAR constraint.
>
> *Representative results (Yelp, Bert-base):*
>
>
> |  |  | BadNets |  | AddSent |  | StyleBkd |  | SynBkd |  |
> |---|:---:|:---:|:---:|:---:|:---:|:---:|:---:|:---:|:---:|
> |  |  | FAR | Power | FAR | Power | FAR | Power | FAR | Power |
> | Yelp | OSCS | 0.028±0.008 | 0.891±0.017 | 0.017±0.010 | 0.658±0.136 | 0.022±0.005 | 0.781±0.025 | 0.013±0.008 | 0.837±0.058 |
> |  | Oracle | 0.050±0.000 | 0.914±0.001 | 0.050±0.000 | 0.841±0.002 | 0.050±0.000 | 0.854±0.002 | 0.050±0.000 | 0.949±0.001 |
> | AGNews | OSCS | 0.033±0.008 | 0.720±0.034 | 0.046±0.009 | 0.780±0.026 | 0.048±0.002 | 0.749±0.005 | 0.040±0.006 | 0.537±0.021 |
> |  | Oracle | 0.050±0.000 | 0.841±0.003 | 0.050±0.000 | 0.864±0.002 | 0.050±0.000 | 0.797±0.007 | 0.050±0.000 | 0.731±0.004 |
> | HSOL | OSCS | 0.014±0.007 | 0.744±0.086 | 0.053±0.010 | 0.777±0.040 | 0.056±0.004 | 0.452±0.007 | 0.042±0.013 | 0.598±0.052 |
> |  | Oracle | 0.050±0.000 | 0.889±0.004 | 0.050±0.000 | 0.846±0.004 | 0.051±0.000 | 0.459±0.005 | 0.050±0.000 | 0.631±0.004 |
>
> We observe that even under this oracle setting, the improvement in power is limited, especially for weaker scores. Similar trends hold across datasets and models. This indicates that the bottleneck is the separability of the score rather than the framework.
>
> Overall, our framework guarantees FAR control, while achievable power is fundamentally constrained by the quality of the scoring function. Improving power, therefore, requires more discriminative scores.
>
> ---
>
> **W3 (Assumption 5.6).**
> We empirically evaluate the separation parameter $\epsilon_\lambda$ across all settings.
>
> *Representative results (Yelp, Bert-base):*
>
>
> |        |         |   BadNets   |   AddSent   |  StyleBkd   |   SynBkd    |
> | :----: | ------- | :---------: | :---------: | :---------: | :---------: |
> |  Yelp  | md      | 0.000±0.000 | 0.000±0.000 | 0.000±0.000 | 0.000±0.000 |
> |        | badacts | 0.000±0.000 | 0.000±0.000 | 0.002±0.001 | 0.000±0.000 |
> | AGNews | md      | 0.000±0.000 | 0.000±0.000 | 0.001±0.001 | 0.000±0.000 |
> |        | badacts | 0.000±0.000 | 0.000±0.000 | 0.001±0.001 | 0.000±0.000 |
> |  HSOL  | md      | 0.000±0.000 | 0.000±0.000 | 0.000±0.000 | 0.000±0.000 |
> |        | badacts | 0.000±0.000 | 0.000±0.000 | 0.006±0.001 | 0.000±0.000 |
>
>
> We observe that $\epsilon_\lambda$ is consistently very small (often close to zero), indicating that the assumption holds in practice.
>
> Intuitively, $\epsilon_\lambda$ measures the probability mass of poisoned samples whose scores exceed the $\lambda$-quantile of benign scores. In our experiments, we set $\lambda=0.95$, corresponding to a high quantile of the clean distribution. As long as the scoring function provides reasonable discrimination (i.e., most poisoned samples receive lower scores than benign ones), $\epsilon_\lambda$ remains small.
>
> ---
>
> **Q1.** Please refer to W1.
> **Q2.** Please refer to W2.

---

> > ### Author Rebuttal · Reviewer_BPJL · 2026-04-03
> >
> > I'm still worried about $$\Delta$$ not having much practical safety value.

---

### Official Review · Reviewer_9Xzc · 2026-03-22

**Soundness:** 2
**Presentation:** 2
**Significance:** 2
**Originality:** 2
**Overall Recommendation:** 3
**Confidence:** 2

**Summary:**

This paper introduces OSCS (Online Selection with Provable FAR Control), a framework for controlling the False Acceptance Rate when detecting malicious inputs to Large Language Models in an online streaming setting. OSCS leverages existing detection mechanisms and performs calibration using only benign samples through recursive kernel density estimation. The authors demonstrate the effectiveness of their approach against both backdoor and jailbreak attacks.

**Compliance With Llm Reviewing Policy:**

Affirmed.

**Key Questions For Authors:**

what's the effect of changing poisoning rate?

**Limitations:**

yes

**Strengths And Weaknesses:**

Strengths:
- The paper talks about an important problem -- existing detection based defenses lack calibration control.
- The approach is generic and covers both model and sample level attacks.

Weaknesses:
- The evaluation section focuses entirely on FAR and doesn't provide any guarantees on power (i.e. benign acceptance rate). In Table 1, the benign acceptance rates are too low to be practical.
- The approach assumes a stationary mixture distribution and does not account for an adaptive adversary.

---

> ### Author Rebuttal · Authors · 2026-03-31
>
> We thank the reviewer for raising this important point.
>
> **W1 (Power).**
> We first clarify that low power arises from a fundamental FAR–power trade-off determined by the discriminative capacity of the scoring function. When the score distributions of benign and malicious samples significantly overlap, methods enforcing strict FAR must avoid selecting samples from the overlapping region, leading to conservative thresholds and thus rejecting many benign samples, which reduces power.
>
> This is consistent with our results: MD exhibits lower power due to weaker discriminative signals, while BadActs achieves higher power by leveraging internal activations and providing better separation.
>
>
> |  |  | BadNets |  | AddSent |  | StyleBkd |  | SynBkd |  |
> |---|:---:|:---:|:---:|:---:|:---:|:---:|:---:|:---:|:---:|
> |  |  | FAR | Power | FAR | Power | FAR | Power | FAR | Power |
> | Yelp | OSCS | 0.028±0.008 | 0.891±0.017 | 0.017±0.010 | 0.658±0.136 | 0.022±0.005 | 0.781±0.025 | 0.013±0.008 | 0.837±0.058 |
> |  | Oracle | 0.050±0.000 | 0.914±0.001 | 0.050±0.000 | 0.841±0.002 | 0.050±0.000 | 0.854±0.002 | 0.050±0.000 | 0.949±0.001 |
> | AGNews | OSCS | 0.033±0.008 | 0.720±0.034 | 0.046±0.009 | 0.780±0.026 | 0.048±0.002 | 0.749±0.005 | 0.040±0.006 | 0.537±0.021 |
> |  | Oracle | 0.050±0.000 | 0.841±0.003 | 0.050±0.000 | 0.864±0.002 | 0.050±0.000 | 0.797±0.007 | 0.050±0.000 | 0.731±0.004 |
> | HSOL | OSCS | 0.014±0.007 | 0.744±0.086 | 0.053±0.010 | 0.777±0.040 | 0.056±0.004 | 0.452±0.007 | 0.042±0.013 | 0.598±0.052 |
> |  | Oracle | 0.050±0.000 | 0.889±0.004 | 0.050±0.000 | 0.846±0.004 | 0.051±0.000 | 0.459±0.005 | 0.050±0.000 | 0.631±0.004 |
>
>
> To further verify that this limitation is not caused by our framework, we introduce a strong oracle baseline that has access to labels and selects optimal thresholds to maximize power under the same FAR constraint. The results are presented in the above table. Even under this unrealistically strong setting, the power improvement over our method remains limited, especially for weaker scores (e.g., MD), indicating that the bottleneck lies in score separability rather than our framework.
>
> Overall, our framework guarantees FAR control, while achievable power is fundamentally constrained by the quality of the scoring function. Improving power thus primarily requires more discriminative scores.
>
> ---
>
> **W2 (Non-stationarity).**
>
> We thank the reviewer for pointing out this limitation. In the revised version, we extend our framework to handle time-varying mixture distributions, covering both poisoning rate changes and score distribution shifts.
>
> We adopt exponential weighting for both clean proportion and density estimation:
> $$
> \hat{\pi}_c^{(t)} = 1 - \frac{\sum \kappa_b(\cdot)\mathbf{1}(\hat p_j>\lambda)}{(1-\lambda)\sum \kappa_b(\cdot)}, \quad
> \hat f_{\mathrm{mix}}^{(t)} = \lambda \hat f^{(t-1)} + (1-\lambda)K_h(\cdot),
> $$
> where $\lambda=\exp(-1/b)$.
>
> Intuitively, this replaces global averaging with a **local, time-aware estimator** that emphasizes recent data. Older samples receive exponentially decaying weights, so the estimator effectively operates on a local window controlled by $b$ (decay rate) and $q$ (maximum horizon), enabling adaptation to distribution shifts.
>
> **Theory.** Under mild smoothness $\|\pi_{t+1}-\pi_t\|,\|f_{t+1}-f_t\|\le \zeta$, we show that both estimators achieve high-probability bounds with an additional drift term that scales as
> $$\zeta^{\frac{\beta}{2\beta+2}}(\log(1/\zeta))^{\frac{\beta}{2\beta+2}} \quad \text{and} \quad \zeta^{1/3}(\log(1/\zeta))^{1/3}$$
> which vanish as $\zeta\to0$, implying that FAR control remains effective under gradual changes.
>
>
> |                     | BadNets     |             | AddSent     |             | StyleBkd    |             | SynBkd      |             |
> | ------------------- | ----------- | ----------- | ----------- | ----------- | ----------- | ----------- | ----------- | ----------- |
> |                     | FAR         | Power       | FAR         | Power       | FAR         | Power       | FAR         | Power       |
> | Periodic_linear     | 0.010±0.001 | 0.892±0.004 | 0.025±0.001 | 0.929±0.003 | 0.038±0.001 | 0.885±0.002 | 0.028±0.001 | 0.929±0.002 |
> | Sinusoidal          | 0.009±0.001 | 0.892±0.004 | 0.024±0.001 | 0.933±0.005 | 0.035±0.001 | 0.887±0.002 | 0.025±0.000 | 0.932±0.002 |
> | Bounded_random_walk | 0.009±0.001 | 0.891±0.004 | 0.025±0.001 | 0.931±0.005 | 0.038±0.003 | 0.884±0.002 | 0.027±0.001 | 0.929±0.002 |
>
> **Experiments.** We evaluate under representative drift patterns (periodic linear, sinusoidal, bounded random walk; poison rate ∈ [0,0.4]). Results on Yelp with BERT are presented in the above table, showing that our method consistently maintains FAR control under all non-stationary settings, demonstrating robustness to evolving adversaries.
>
> Full proofs and additional results will be included in the final version.
>
> ---
>
> **Q1.** Please refer to W2.

---

> > ### Author Rebuttal · Reviewer_9Xzc · 2026-04-05
> >
> > - I appreciate the response but I still feel the power values are not practical.
> > - Thanks for the extended analysis on different drift patterns. The pattern considered are still well formed. What about worst case adversaries who are aware of how OCSC works.

---

### Decision · Program_Chairs · 2026-04-30

**Decision:**

Accept (regular)

**Comment:**

This work introduces OSCS (Online Selection with Provable FAR Control), which studies an interesting and novel intersection of explicitly controlling the false acceptance rate (FAR) in an online streaming setting without malicious calibration data.

The reviewers agreed that this work studies an interesting and novel setting, that the theoretical results are clean and there are sufficient empirical analyses.

However, there remained some open important weaknesses:

1. how this work fairs in the face of adaptive adversaries
2. that the method does not provide acceptable power.

Improving on this would make this paper stronger.